# A mean-field analysis of two-player zero-sum games

**Carles Domingo-Enrich**
Courant Institute of Mathematical Sciences
New York University
New York, NY
cd2754@nyu.edu

**Samy Jelassi**
Princeton University
Princeton, NJ
sjelassi@princeton.edu

**Arthur Mensch**
École Normale Supérieure
Paris, France
arthur.mensch@m4x.org

**Grant Rotskoff**
Courant Institute of Mathematical Sciences
New York University
New York, NY
rotskoff@cims.nyu.edu

**Joan Bruna**
Courant Institute of Mathematical Sciences
& Center for Data Science
New York University
New York, NY
bruna@cims.nyu.edu

## Abstract

Finding Nash equilibria in two-player zero-sum continuous games is a central problem in machine learning, e.g. for training both GANs and robust models. The existence of pure Nash equilibria requires strong conditions which are not typically met in practice. Mixed Nash equilibria exist in greater generality and may be found using mirror descent. Yet this approach does not scale to high dimensions. To address this limitation, we parametrize mixed strategies as mixtures of particles, whose positions and weights are updated using gradient descent-ascent. We study this dynamics as an interacting gradient flow over measure spaces endowed with the Wasserstein-Fisher-Rao metric. We establish global convergence to an approximate equilibrium for the related Langevin gradient-ascent dynamic. We prove a law of large numbers that relates particle dynamics to mean-field dynamics. Our method identifies mixed equilibria in high dimensions and is demonstrably effective for training mixtures of GANs.

## 1 Introduction

Multi-objective optimization problems arise in many fields, from economics to civil engineering. Tasks that require optimizing multiple objectives have also become a routine part of many agent-based machine learning algorithms including generative adversarial networks (Goodfellow et al., 2014), imaginative agents (Racanière et al., 2017), hierarchical reinforcement learning (Wayne and Abbott, 2014) and multi-agent reinforcement learning (Bu et al., 2008). It not only remains difficult to carry out the necessary optimization, but also to assess the optimality of a given solution.

Multi-agent optimization is generally cast as finding equilibria in the space of strategies. The classic notion of equilibrium is due to Nash (Nash, 1951): a Nash equilibrium is a set of agent strategies for which no agent can unilaterally improve its loss value. Pure Nash equilibria, in which each agent

adopts a single strategy, provide a limited notion of optimality because they exist only under restrictive conditions. On the other hand, mixed Nash equilibria (MNE), where agents adopt a strategy from a probability distribution over the set of all strategies, exist in much greater generality (Glicksberg, 1952). Importantly, MNE exist for games with infinite-dimensional compact strategy spaces, in which each player observes a loss function that is continuous in its strategy. We encounter this setting in different game formulations of machine learning problems, like GANs (Goodfellow et al., 2014).

Although MNE are guaranteed to exist, it is difficult to identify them. Indeed, worst-case complexity analyses have shown that without additional assumptions on the losses there is no efficient algorithm for finding a MNE, even in the case of two-player finite games (Daskalakis et al., 2009). Some recent progress has been made; (Hsieh et al., 2019) proposed a mirror-descent algorithm with convergence guarantees, which is approximately realizable in high-dimension.

**Contributions.** Following Hsieh et al. (2019), we formulate continuous two-player zero-sum games as a multi-agent optimization problem over the space of probability measures on strategies. We describe two gradient descent-ascent dynamics in this space, both involving a transport term.

- We show that the stationary points of a gradient ascent-descent flow with Langevin diffusion over the space of mixed strategies are approximate MNE.

- We analyse a gradient ascent-descent dynamics that jointly updates the positions and weights of two mixed strategies to converge to an *exact* MNE. This dynamics corresponds to a gradient descent-ascent flow over the space of measures endowed with a Wasserstein-Fisher-Rao (WFR) metric (Chizat, Peyré, et al., 2018).

- We discretize both dynamics in space and time to obtain implementable training algorithms. We provide mean-field type consistency results on the discretization. We demonstrate numerically how both dynamics overcome the curse of dimensionality for finding MNE on synthetic games. On real data, we use WFR flows to train mixtures of GANs, that explicitly discover data clusters while maintaining good performance.

## 2    Related work

**Equilibria in continuous games.**    Most of the works that study convergence to equilibria in continuous games or GANs do not frame the problem in the infinite-dimensional space of measures, but on finite-dimensional spaces. That is because they either (i) restrict their attention to games with convexity-concavity assumptions in which pure equilibria exist (Mertikopoulos et al., 2019; Lin et al., 2018; Nouiehed et al., 2019), or (ii) provide algorithms with convergence guarantees to local notions of equilibrium such as stable fixed points, local Nash equilibria and local minimax points (Heusel et al., 2017; Adolphs et al., 2018; Mazumdar et al., 2019; Jin et al., 2019; Fiez et al., 2019; Balduzzi et al., 2018). Both approaches differ from ours, which is to give global convergence guarantees without convexity assumptions. Some works have studied approximate MNE in infinite-dimensional measure spaces. Arora et al. (2017) proved the existence of approximate MNE and studied the generalization properties of this approximate solution; their analysis, however, does not provide a constructive method to identify such a solution. In a more explicit setting, Grnarova et al. (2017) designed an online-learning algorithm for finding a MNE in GANs under the assumption that the discriminator is a single hidden layer neural network. Balandat et al. (2016) apply the dual averaging algorithm to the minimax problem and show that it recovers a MNE, but they do not provide any convergence rate nor a practical algorithm for learning mixed NE. Our framework holds without making any assumption on the architectures of the discriminator and generator and provides explicit algorithms with some convergence guarantees.

**Mean-field view of nonlinear gradient descent.**    Our approach is closely related to the mean-field perspective on wide neural networks (Mei et al., 2018; Rotskoff and Vanden-Eijnden, 2018; Chizat and Bach, 2018; Sirignano and Spiliopoulos, 2019; Rotskoff, Jelassi, et al., 2019). These methods view training algorithms as approximations of Wasserstein gradient flows, which are dynamics on measures over the space of neurons. In our setting, a mixed strategy corresponds to a measure over the space of strategies.

**Particle approaches for two-player games.** Our theoretical work sheds a new light on the results of Hsieh et al. (2019), and rigorously justifies important algorithmic modifications the authors introduced. Specifically, they give rates of convergence for infinite-dimensional mirror descent on measures (i.e. updating strategy weights but not their positions). The straightforward implementation of this algorithm performs poorly unless the dimension is low (Fig. 1), which is why they proposed an 'implementable' two-timescale version, in which the inner loop is a transport-based sampling procedure closely related to our Algorithm 1. This implementable version is not studied theoretically, as the two-timescale structure hinders a thorough analysis. Our analysis includes transport on equal footing with mirror descent updates.

## 3 Problem setup and mean-field dynamics

**Notation.** For a topological space $\mathcal{X}$ we denote by $\mathcal{P}(\mathcal{X})$ the space of Borel probability measures on $\mathcal{X}$, and $\mathcal{M}_+(\mathcal{X})$ the space of Borel (positive) measures. For a given measure $\mu \in \mathcal{P}(\mathcal{X})$ that is absolutely continuous with respect to the canonical Borel measure $dx$ of $\mathcal{X}$ and has Radon-Nikodym derivative $\frac{d\mu}{dx} \in \mathcal{C}(\mathcal{X})$, we define its differential entropy $H(\mu) = -\int \log(\frac{d\mu}{dx})d\mu$. For measures $\mu, \nu \in \mathcal{P}(\mathcal{X})$, $\mathcal{W}_2$ is the 2-Wasserstein distance.

### 3.1 Lifting differentiable games to spaces of strategy distributions

**Differentiable two-player zero-sum games.** We recall the definition of a differentiable zero-sum game, and show how finding a mixed Nash equilibrium to such a game is equivalent to solving a bi-linear game in the infinite dimensional space of distributions on strategies. We will use gradient flow approaches for solving the lifted problem.

**Definition 1.** *A two-player zero-sum game consists of a set of two players with parameters $z = (x,y) \in \mathcal{Z} = \mathcal{X} \times \mathcal{Y}$, where players observe a loss functions $\ell_1 \colon \mathcal{Z} \to \mathbb{R}$ and $\ell_2 \colon \mathcal{Z} \to \mathbb{R}$ that satisfy for all $(x,y) \in \mathcal{Z}$, $\ell_1(x,y) + \ell_2(x,y) = 0$. $\ell \triangleq \ell_1 = -\ell_2$ is the loss of the game.*

The compact finite-dimensional spaces of strategies $\mathcal{X}$ and $\mathcal{Y}$ are endowed with a certain distance function $d$ (which we assume Euclidean in what follows—§G.5 derives our results on arbitrary strategy manifolds). This allows to define differentiable games, amenable to first-order optimization. We make the following mild assumption over the regularity of losses and constraints (Glicksberg, 1952).

**Assumption 1.** *The parameter spaces $\mathcal{X}$ and $\mathcal{Y}$ are compact Riemannian manifolds without boundary of dimensions $d_x, d_y$ embedded in $\mathbb{R}^{D_x}, \mathbb{R}^{D_y}$ respectively. The loss $\ell$ is continuously differentiable and $L$-smooth with respect to each parameter. That is, for all $x, x' \in \mathcal{X}$ and $y, y' \in \mathcal{Y}$, $\|\nabla_x \ell(x,y) - \nabla_x \ell(x',y')\|_2 \leqslant L(d(x,x') + d(y,y'))$, $\|\nabla_y \ell(x,y) - \nabla_y \ell(x',y')\|_2 \leqslant L(d(x,x') + d(y,y'))$.*

**From pure to mixed Nash equilibria.** Assuming that both players play simultaneously, a pure Nash equilibrium point is a pair of strategies $(x^*, y^*) \in \mathcal{X} \times \mathcal{Y}$ such that, for all $(x,y) \in \mathcal{X} \times \mathcal{Y}$, $\ell(x^\star, y) \leqslant \ell(x^\star, y^\star) \leqslant \ell(x, y^\star)$. Such points do not always exist in continuous games. In contrast, mixed Nash equilibria (MNE) are guaranteed to exist (Glicksberg, 1952) under Asm. 1. Those distributions $(\mu_x^\star, \mu_y^\star) \in \mathcal{P}(\mathcal{X}) \times \mathcal{P}(\mathcal{Y})$ are global saddle points of the expected loss $\mathcal{L}(\mu_x, \mu_y) \triangleq \iint \ell(x,y)d\mu_x(x)d\mu_y(y)$. Formally, for all $\mu_x, \mu_y \in \mathcal{P}(\mathcal{X}) \times \mathcal{P}(\mathcal{Y})$,

$$\mathcal{L}(\mu_x^*, \mu_y) \leqslant \mathcal{L}(\mu_x^*, \mu_y^*) \leqslant \mathcal{L}(\mu_x, \mu_y^*). \tag{1}$$

We quantify the accuracy of an estimation $(\hat{\mu}_x, \hat{\mu}_y)$ of a MNE using the Nikaidô and Isoda (1955) error

$$\mathrm{NI}(\hat{\mu}_x, \hat{\mu}_y) = \sup_{\mu_y \in \mathcal{P}(\mathcal{Y})} \mathcal{L}(\hat{\mu}_x, \mu_y) - \inf_{\hat{\mu}_x \in \mathcal{P}(\mathcal{X})} \mathcal{L}(\mu_x, \hat{\mu}_y). \tag{2}$$

We track the evolution of this metric in our theoretical results (§4.2) and in our experiments. We obtain guarantees on finding $\varepsilon$-MNE $(\mu_x^\varepsilon, \mu_y^\varepsilon)$, i.e. distribution pairs such that $\mathrm{NI}(\mu_x^\varepsilon, \mu_y^\varepsilon) \leqslant \varepsilon$.

### 3.2 Training dynamics on discrete mixtures of strategies

We study three different dynamics for solving (1). Let us first assume that the two players play *finite* mixtures of $n$ strategies $\mu_x = \sum_{i=1}^n w_x^i \delta_{x^i} \in \mathcal{P}(\mathcal{X})$, $\mu_y = \sum_{i=1}^n w_y^i \delta_{y^i} \in \mathcal{P}(\mathcal{Y})$, where

---

**Algorithm 1** Langevin Descent-Ascent (L-DA).

---

1: **Input**: IID samples $x_0^1, \ldots, x_0^n$ from $\mu_{x,0} \in \mathcal{P}(\mathcal{X})$, IID samples $y_0^1, \ldots, y_0^n \in \mathcal{Y}$ from $\mu_{y,0} \in \mathcal{P}(\mathcal{Y})$
2: **for** $t = 0, \ldots, T$ **do**
3:      **for** $i = 1, \ldots, n$ **do**
4:         Sample $\Delta W_t^i \sim \mathcal{N}(0, I)$, $x_{t+1}^i = x_t^i - \frac{\eta}{n} \sum_{j=1}^n \nabla_x \ell(x_t^i, y_t^j) + \sqrt{2\eta\beta^{-1}} \Delta W_t^i$
5:         Sample $\Delta \bar{W}_t^i \sim \mathcal{N}(0, I)$, $y_{t+1}^i = y_t^i + \frac{\eta}{n} \sum_{j=1}^n \nabla_y \ell(x_t^j, y_t^i) + \sqrt{2\eta\beta^{-1}} \Delta \bar{W}_t^i$
6: **Return** $\mu_{x,T}^n = \frac{1}{n} \sum_{i=1}^n \delta_{x_T^i}$,    $\mu_{y,T}^n = \frac{1}{n} \sum_{i=1}^n \delta_{y_T^i}$

---

$\{x^i, y^i\}_{i \in [1:n]}$ are the positions of the strategies and $w_x^i, w_y^i \geqslant 0$ are their weights. In the simplest setting, those mixtures are assumed uniform, i.e. $w_x^i = w_y^i = 1/n$. Finding the best $2n$ strategies involve finding a saddle point of $\mathcal{L}(\mu_x, \mu_y) = \frac{1}{n^2} \sum_i \sum_j \ell(x_i, y_j)$. Starting from random independent initial strategies $x_0^i = \xi_i \sim \mu_{x,0}, y_0^i = \bar{\xi}_i \sim \mu_{y,0}$, we may hope that the gradient descent-ascent dynamics

$$\frac{dx_t^i}{dt} = -\frac{1}{n} \sum_{j=1}^n \nabla_x \ell(x_t^i, y_t^j), \quad \frac{dy_t^i}{dt} = \frac{1}{n} \sum_{j=1}^n \nabla_y \ell(x_t^j, y_t^i), \quad \forall i \in [1:n] \tag{3}$$

finds such a saddle point. Yet this may fail in simple nonconvex-nonconcave games, as illustrated in §G.2—the particle distributions collapse to a stationary point that is not a MNE.

To mitigate this convergence problem, we analyse a perturbed dynamics analogous to Langevin gradient descent. Using the same initialization as in (3), we add a small amount of noise in the gradient dynamics and obtain the stochastic differential equations

$$dX_t^i = -\frac{1}{n} \sum_{j=1}^n \nabla_x \ell(X_t^i, Y_t^j) dt + \sqrt{\frac{2}{\beta}} dW_t^i, \ dY_t^i = \frac{1}{n} \sum_{j=1}^n \nabla_y \ell(X_t^j, Y_t^i) dt + \sqrt{\frac{2}{\beta}} d\bar{W}_t^i, \tag{4}$$

where $W_t^i, \bar{W}_t^i$ are independent Brownian motions. The discretization of (4) results in Alg. 1; it is similar to Alg. 4 in Hsieh et al. (2019).

We propose a second alternative dynamics to (3), that updates both the positions and the weights of the particles, using relative updates for weights. We will show that it enjoys better convergence properties in the mean-field limit.

$$\frac{dx_t^i}{dt} = -\gamma \sum_{j=1}^n w_{y,t}^j \nabla_x \ell(x_t^i, y_t^j), \quad \frac{dw_{x,t}^i}{dt} = \alpha \left( -\sum_{j=1}^n w_{y,t}^j \ell(x_t^i, y_t^j) + K(t) \right) w_{x,t}^i \tag{5}$$

and similarly for all $y_t^i$ (flipping the sign of $\ell$). $K(t) \triangleq \sum_{k=1}^n \sum_{j=1}^n w_{y,t}^j w_{x,t}^k \ell(x_t^i, y_t^j)$ keeps $w_{x,t}$ in the simplex. We use uniform weights for initialization. When $\gamma = 0$ and $\alpha = 1$, only the weights are updated: this results in the continuous-time version of the infinite-dimensional mirror descent studied by Hsieh et al. (2019). The Euler discretization of (5) results in Alg. 2.

### 3.3 Training dynamics as gradient flows on measures

The three dynamics that we have introduced at the level of particles induces dynamics on the associated empirical probability measures. If $\{x_t^i, y_t^i\}_{i \in [1,n]}$ is a solution of (3), then $\mu_x(t) = \frac{1}{n} \sum_{i=1}^n \delta_{x_t^i}$ and $\mu_y(t) = \frac{1}{n} \sum_{i=1}^n \delta_{y_t^i}$ are solutions of the *Interacting Wasserstein Gradient Flow* (IWGF) of $\mathcal{L}$:

$$\begin{cases} \partial_t \mu_x = \nabla \cdot (\mu_x \nabla_x V_x(\mu_y, x)), & \mu_x(0) = \frac{1}{n} \sum_{i=1}^n \delta_{x_0^i}, \\ \partial_t \mu_y = -\nabla \cdot (\mu_y \nabla_y V_y(\mu_x, y)), & \mu_y(0) = \frac{1}{n} \sum_{i=1}^n \delta_{y_0^i}. \end{cases} \tag{6}$$

The derivation of (6) is provided in §G.3. We use the notation $V_x(\mu_y, x) \triangleq \frac{\delta \mathcal{L}}{\delta \mu_x}(\mu_x, \mu_y)(x) = \int \ell(x, y) d\mu_y(y)$ for the first variations of the functional $\mathcal{L}(\mu_x, \mu_y)$. Holding $\mu_y$ fixed, the evolution

---

**Algorithm 2** Wasserstein-Fisher-Rao Descent-Ascent (WFR-DA).

---

1: **Input**: IID samples $x_0^{(1)}, \ldots, x_0^{(n)}$ from $\nu_{x,0} \in \mathcal{P}(\mathcal{X})$, IID samples $y_0^{(1)}, \ldots, y_0^{(n)}$ from $\nu_{y,0} \in \mathcal{P}(\mathcal{Y})$. Initial weights: For all $i \in [1:n]$, $w_x^{(i)} = 1$, $w_y^{(i)} = 1$.

2: **for** $t = 0, \ldots, T$ **do**

3: $\quad [x_{t+1}^{(i)}]_{i=1}^n = [x_t^{(i)} - \eta \sum_{j=1}^n w_{y,t}^{(i)} \nabla_x \ell(x_t^{(i)}, y_t^{(j)})]_{i=1}^n$

4: $\quad [\hat{w}_{x,t+1}^{(i)}]_{i=1}^n = \left[ w_{x,t}^{(i)} \exp\left(-\eta' \sum_{j=1}^n w_{y,t}^{(j)} \ell(x_t^{(i)}, y_t^{(j)})\right) \right]_{i=1}^n, \quad [w_{x,t+1}^{(i)}]_{i=1}^n = [\hat{w}_{x,t+1}^{(i)}]_{i=1}^n / \sum_{j=1}^n \hat{w}_{x,t+1}^{(j)}$

5: $\quad [y_{t+1}^{(i)}]_{i=1}^n = [y_t^{(i)} + \eta \sum_{j=1}^n w_{x,t}^{(j)} \nabla_y \ell(x_t^{(j)}, y_t^{(i)})]_{i=1}^n,$

6: $\quad [\hat{w}_{y,t+1}^{(i)}]_{i=1}^n = \left[ w_{y,t}^{(i)} \exp\left(\eta' \sum_{j=1}^n w_{x,t}^{(j)} \ell(x_t^{(j)}, y_t^{(i)})\right) \right]_{i=1}^n, \quad [w_{y,t+1}^{(i)}]_{i=1}^n = [\hat{w}_{y,t+1}^{(i)}]_{i=1}^n / \sum_{j=1}^n \hat{w}_{y,t+1}^{(j)}$

7: **Return** $\bar{\nu}_{x,T}^n = \frac{1}{T+1} \sum_{t=0}^T \sum_{i=1}^n w_{x,T}^{(i)} \delta_{x_T^{(i)}}, \quad \bar{\nu}_{y,T}^n = \frac{1}{T+1} \sum_{t=0}^T \sum_{i=1}^n w_{y,T}^{(i)} \delta_{y_T^{(i)}}$

---

of $\mu_x$ is a Wasserstein gradient flow on $\mathcal{L}(\cdot, \mu_y)$ (Ambrosio et al., 2005). We interpret these PDEs in the weak sense, i.e. equality holds when integrating measures against bounded continuous functions.

The distributions $\mu_x(t) = \frac{1}{n} \sum_{i=1}^n \delta_{X_t^i}$ and $\mu_y(t) = \frac{1}{n} \sum_{i=1}^n \delta_{Y_t^i}$, where $\{X^i, Y^i\}_{i \in [1:n]}$ are solutions of (4) follows a *Entropy-Regularized Interacting Wasserstein Gradient Flow (ERIWGF)*:

$$\begin{cases} \partial_t \mu_x = \nabla_x \cdot (\mu_x \nabla_x V_x(\mu_y, x)) + \beta^{-1} \Delta_x \mu_x, & \mu_x(0) = \frac{1}{n} \sum_{i=1}^n \delta_{x_0^i} \\ \partial_t \mu_y = -\nabla_y \cdot (\mu_y \nabla_y V_y(\mu_x, y)) + \beta^{-1} \Delta_y \mu_y, & \mu_y(0) = \frac{1}{n} \sum_{i=1}^n \delta_{y_0^i} \end{cases} \tag{7}$$

The derivation of (7) is provided in Lemma 10. It is a system of coupled nonlinear Fokker-Planck equations, that are the Kolmogorov forward equations of the SDE (4). They correspond to the IWGF of the entropy-regularized loss $\mathcal{L}_\beta(\mu_x, \mu_y) \triangleq \mathcal{L}(\mu_x, \mu_y) + \beta^{-1}(H(\mu_y) - H(\mu_x))$.

Finally, if $\{x^i, y^i, w_x^i, w_y^i\}_{i \in [1:n]}$ solve (5), then $\mu_x(t) = \sum_{i=1}^n w_{x,t}^i \delta_{x_t^i}$, $\mu_y(t) = \sum_{i=1}^n w_{y,t}^i \delta_{y_t^i}$ solve the *Interacting Wasserstein-Fisher-Rao Gradient Flow (IWFRGF)* of $\mathcal{L}$:

$$\begin{cases} \partial_t \mu_x & = \gamma \nabla_x \cdot (\mu_x \nabla_x V_x(\mu_y, x)) - \alpha \mu_x (V_x(\mu_y, x) - \mathcal{L}(\mu_x, \mu_y)), \ \mu_x(0) = \sum_{i=1}^n w_{x,0}^i \delta_{x_0^i}, \\ \partial_t \mu_y & = -\gamma \nabla_y \cdot (\mu_y \nabla_y V_y(\mu_x, y)) + \alpha \mu_y (V_y(\mu_x, y) - \mathcal{L}(\mu_x, \mu_y)), \ \mu_y(0) = \sum_{i=1}^n w_{y,0}^i \delta_{y_0^i}. \end{cases} \tag{8}$$

The derivation of (8) is provided in App. A and Lemma 11. The Wasserstein-Fisher-Rao or Hellinger-Kantorovich metric (Chizat, Peyré, et al., 2015; Kondratyev et al., 2016; Gallouët and Monsaingeon, 2016) is a metric on the probability space $\mathcal{M}_+(\mathcal{X})$ induced by a lifting to the space $\mathcal{P}(\mathcal{X} \times \mathbb{R}^+)$ of the form $\nu \mapsto \mu = \int_{\mathbb{R}^+} w \, d\nu(\cdot, w)$. If we keep $\nu_y$ fixed, the first equation in (8) is a Wasserstein-Fisher-Rao gradient flow (slightly modified by the term $\alpha \mu_x \mathcal{L}(\mu_x, \mu_y)$ to constrain $\mu_x$ in $\mathcal{P}(\mathcal{X})$). The term $-\alpha \mu_x (V_x(\mu_y, x) - \mathcal{L}(\mu_x, \mu_y))$, which also arises in entropic mirror descent, allow mass to 'teleport' from bad strategies to better ones with finite cost by moving along the weight coordinate. Wasserstein-Fisher-Rao gradient flows have been used by Chizat (2019), Rotskoff, Jelassi, et al. (2019), and Liero et al. (2018) in the context of optimization.

Initialization of (6), (7) and (8) may be done with the measures $\mu_{x,0}$ and $\mu_{y,0}$ from which $\{x_0^i\}, \{y_0^i\}$ are sampled, in which case the measures $\mu_x(t)$ and $\mu_y(t)$ are not discrete and follow the *mean-field* dynamics. In §4.3 we link the dynamics starting from discrete realizations to the mean-field dynamics.

# 4 Convergence analysis

We establish convergence results for the entropy-regularized dynamics and the WFR dynamics.

## 4.1 Convergence of the entropy-regularized Wasserstein dynamics

The following theorem characterizes the stationary points of the entropy-regularized dynamics.

**Theorem 1.** *Suppose that Asm. 1 holds, that $\ell \in C^2(\mathcal{X} \times \mathcal{Y})$ and that the initial measures $\mu_{x,0}, \mu_{y,0}$ have densities in $L^1(\mathcal{X}), L^1(\mathcal{Y})$. If a solution $(\mu_x(t), \mu_y(t))$ of the ERIWGF (7) converges in time, it must converge to the point $(\hat{\mu}_x, \hat{\mu}_y)$ which is the unique fixed point of the problem*

$$\rho_x(x) = \frac{1}{Z_x} e^{-\beta \int \ell(x,y) \, d\mu_y(y)}, \quad \rho_y(y) = \frac{1}{Z_y} e^{\beta \int \ell(x,y) \, d\mu_x(x)}. \tag{9}$$

$(\hat{\mu}_x, \hat{\mu}_y)$ *is an $\varepsilon$-Nash equilibrium of the game given by $\mathcal{L}$ when $\beta \geqslant \frac{4}{\varepsilon} \log \left( 2 \frac{1-V_\delta}{V_\delta} (2K_\ell/\varepsilon - 1) \right)$, where $K_\ell := \max_{x,y} \ell(x, y) - \min_{x,y} \ell(x, y)$ is the length of the range of $\ell$, $\delta := \varepsilon/(2Lip(\ell))$ and $V_\delta$ is a lower bound on the volume of a ball of radius $\delta$ in $\mathcal{X}, \mathcal{Y}$.*

The proof is in App. C. Theorem 1 characterizes the stationary points of the ERIWGF but does not provide a guarantee of convergence in time. It implies that if the dynamics (7) converges in time, the limit will be an $\varepsilon$-Nash equilibrium of $\mathcal{L}$, with $\varepsilon = \tilde{O}(1/\beta)$ (disregarding log factors). The dynamics (7) correspond to a McKean-Vlasov process on the joint probability measure $\mu_x \times \mu_y$. While convergence to stationary solutions of such processes have been studied in the Euclidean case (Eberle et al., 2019)l, their results would only guarantee convergence for temperatures $\beta^{-1} \gtrsim Lip(\ell)$ in our setup, which is not strong enough to certify convergence to arbitrary $\varepsilon$-NE.

There is a trade-off between setting a low temperature $\beta^{-1}$, which yields an $\varepsilon$-Nash equilibrium with small $\varepsilon$ but possibly slow or no convergence, and setting a high temperature, which has the opposite effect. Linear potential Fokker-Planck equations (that we recover when both players are decoupled) indeed converge exponentially with rate $e^{-\lambda_\beta t}$ for all $\beta$, with $\lambda_\beta$ decreasing exponentially with $\beta$ for nonconvex potentials (Markowich and Villani, 1999, sec. 5). Entropic regularization also biases the dynamics towards measures with full support and hence precludes convergence to sparse equilibria even if they exist. This problem does not arise in the WFR dynamics.

## 4.2 Analysis of the Wasserstein-Fisher-Rao dynamics

Theorem 2 states that, at a certain time $t_0$, the time averaged measures of the solution $(\nu_x, \nu_y)$ of (8) are an $\varepsilon$-MNE, where $\varepsilon$ can be made arbitrarily small by adjusting the constants $\gamma, \alpha$ of the dynamics. We define $\bar{\nu}_x(t) = \frac{1}{t} \int_0^t \nu_x(s) \, ds$ and $\bar{\nu}_y(t) = \frac{1}{t} \int_0^t \nu_y(s) \, ds$, where $\nu_x$ and $\nu_y$ are solutions of (8).

**Theorem 2.** *Let $\varepsilon > 0$ arbitrary. Suppose that $\nu_{x,0}, \nu_{y,0}$ are such that their Radon-Nikodym derivatives with respect to the Borel measures of $\mathcal{X}, \mathcal{Y}$ are lower-bounded by $e^{-K'_x}, e^{-K'_y}$ respectively. For any $\delta \in (0, 1/2)$, there exists a constant $C_{\delta, \mathcal{X}, \mathcal{Y}, K'_x, K'_y} > 0$ depending on the dimensions of $\mathcal{X}, \mathcal{Y}$, their curvatures and $K'_x, K'_y$, such that if $\gamma/\alpha < 1$, $\frac{\gamma}{\alpha} \leqslant \left( \varepsilon/C_{\delta, \mathcal{X}, \mathcal{Y}, K'_x, K'_y} \right)^{\frac{2}{1-\delta}}$*

$$NI(\bar{\nu}_x(t_0), \bar{\nu}_y(t_0)) \leqslant \varepsilon \quad where \quad t_0 = (\alpha\gamma)^{-1/2}.$$

The proof (App. D) builds on the convergence properties of continuous-time mirror descent and closely follows the proof of Theorem 3.8 from Chizat (2019). We explicit the dependency of $C_{\delta, \mathcal{X}, \mathcal{Y}, K'_x, K'_y}$ on the dimensions of the manifolds and the properties of the loss $\ell$. Notice that Theorem 2 ensures convergence towards an $\varepsilon$-Nash equilibrium of the non-regularized game. Following Chizat (2019), it is possible to replace the regularity assumption on the initial measures $\nu_{x,0}, \nu_{y,0}$ by a singular initialisation, at the expense of using $O(\exp(d))$ particles. This result is not a convergence result for the measures, but rather on the value of the NI error. Notice that it involves time-averaging and a finite horizon. Similar results are common for mirror descent in convex games (Juditsky et al., 2011), albeit in the discrete-time setting.

Theorem 2 does not capture the benefits of transport, as it regards it as a perturbation of mirror descent (which corresponds to $\gamma = 0$). When targetting a small error $\varepsilon$, we need to set $\gamma \ll \alpha$ because of the bound on $\gamma/\alpha$. In this case, mirror descent is the main driver of the dynamics. However, it is seen empirically that taking much higher ratios $\gamma/\alpha$ (i.e. increasing the importance of the transport term) results in better performance. A satisfying explanation of this phenomenon is still sought after in the simpler optimization setting (Chizat, 2019).

## 4.3 Convergence to mean-field

The following theorem (proof in App. F) links the empirical measures of the systems (4), (5) to the solutions of the mean field dynamics (7) and (8) respectively. It can be seen as a law of large numbers. It shows that by Theorem 3, Alg. 1 and Alg. 2 approximate the mean-field dynamics studied in §4.1 and §4.2.

**Theorem 3.** *(i) Let $\mu_x^n = \frac{1}{n} \sum_{i=1}^n \delta_{X^{(i)}} \in \mathcal{C}([0, T], \mathcal{P}(\mathcal{X})), \mu_y^n = \frac{1}{n} \sum_{i=1}^n \delta_{Y^{(i)}} \in \mathcal{C}([0, T], \mathcal{P}(\mathcal{Y}))$ be the empirical measures of a solution of (4) up to an arbitrary time $T$. Let $\mu_x \in$*

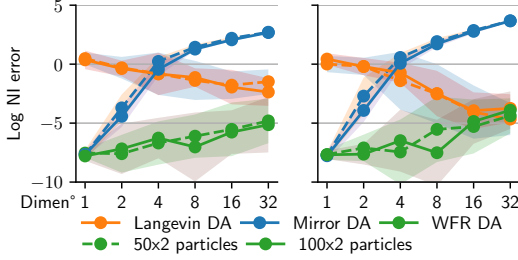

Figure 1: Nikaido-Isoida errors for L-DA, WFR-DA and mirror descent, as a function of the problem dimension, for a nonconvex loss $\ell_a$ (left) and convex loss $\ell_b$ (right). L-DA and WFR-DA outperforms mirror descent for large dimensions. Values averaged over 20 runs after 30000 iterations. Error bars show standard deviation across runs.

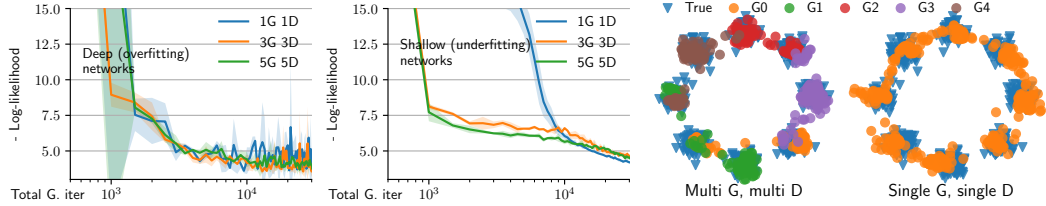

Figure 2: Training mixtures of GANs over a synthetic mixture of Gaussians in 2D. WFR-DA converges faster with models with low number of parameters, and similar performance with over-parametrized models. Mixtures naturally perform a form of clustering of the data. Errors bars show variance across 5 runs.

$\mathcal{C}([0,T],\mathcal{P}(\mathcal{X})), \mu_y \in \mathcal{C}([0,T],\mathcal{P}(\mathcal{Y}))$ *be a solution of the ERIWGF* (7) *with mean-field initial conditions* $\mu_x(0) = \mu_{x,0}, \mu_y(0) = \mu_{y,0}$. *Then,*

$$\mathbb{E}[\mathcal{W}_2^2(\mu_{x,t}^n, \mu_{x,t}) + \mathcal{W}_2^2(\mu_{y,t}^n, \mu_{y,t})] \xrightarrow{n\to\infty} 0, \quad \mathbb{E}[|NI(\mu_{x,t}^n, \mu_{y,t}^n) - NI(\mu_{x,t}, \mu_{y,t})|] \xrightarrow{n\to\infty} 0,$$

*uniformly over* $t \in [0,T]$. *NI is the Nikaido-Isoda error defined in* (2).

*(ii) Let* $\nu_x^n = \sum_{i=1}^n w_{x,t}^i \delta_{X^{(i)}} \in \mathcal{C}([0,T],\mathcal{P}(\mathcal{X})), \mu_y^n = \sum_{i=1}^n w_{y,t}^i \delta_{Y^{(i)}} \in \mathcal{C}([0,T],\mathcal{P}(\mathcal{Y}))$ *be the (projected) empirical measures of a solution of* (5) *up to an arbitrary time* $T$. *Let* $\nu_x \in \mathcal{C}([0,T],\mathcal{P}(\mathcal{X})), \nu_y \in \mathcal{C}([0,T],\mathcal{P}(\mathcal{Y}))$ *be a solution of* (8) *with mean-field initial conditions* $\mu_x(0) = \mu_{x,0}, \mu_y(0) = \mu_{y,0}$. *Then,*

$$\mathbb{E}[\mathcal{W}_2^2(\nu_{x,t}^n, \nu_{x,t}) + \mathcal{W}_2^2(\nu_{y,t}^n, \nu_{y,t})] \xrightarrow{n\to\infty} 0, \quad \mathbb{E}[|NI(\bar{\nu}_{x,t}^n, \bar{\nu}_{y,t}^n) - NI(\bar{\nu}_{x,t}, \bar{\nu}_{y,t})|] \xrightarrow{n\to\infty} 0,$$

*uniformly over* $t \in [0,T]$. $\bar{\nu}_{x,t}, \bar{\nu}_{y,t}, \bar{\nu}_{x,t}^n, \bar{\nu}_{y,t}^n$ *are the time-averaged measures, as in* Theorem 2.

## 5 Numerical Experiments

We show that WFR and Langevin dynamics outperform mirror descent in high dimension, on synthetic games. We then show the interests of using WFR-DA for training GANs. Code has been made available for reproducibility.

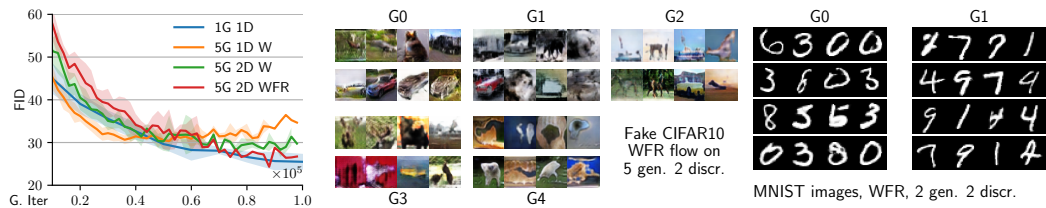

Figure 3: Training mixtures of GANs over CIFAR10. We compare the algorithm that updates the mixture weights and parameters (WFR-DA flow) with the algorithm that only updates parameters (W-DA flow). Using several discriminators and a WFR-DA flow brings more stable convergence. Each generator tends to specialize in a type of images. Errors bars show variance across 5 runs.

## 5.1 Polynomial games on spheres

We study two different games with losses $\ell_a, \ell_b : \mathcal{S}^{d-1} \times \mathcal{S}^{d-1} \to \mathbb{R}$ of the form

$$\ell_a(x, y) = x^\top A_0 x + x^\top A_1 y + y^\top A_2 y + y^\top A_3(x^2) + a_0^\top x + a_1^\top y$$
$$\ell_b(x, y) = x^\top A_0^\top A_0 x + x^\top A_1 y + y^\top A_2^\top A_2 y + a_0^\top x + a_1^\top y,$$

where $A_0, A_1, A_2, A_3, a_0, a_1$ are matrices and vectors with components sampled from a normal distribution $\mathcal{N}(0, 1)$, and $x^2$ is the vector given by component-wise multiplication of $x$. $\ell_b$ is a convex loss on the sphere, while $\ell_a$ is not. We run Langevin Descent-Ascent (updates of positions) and WFR Descent-Ascent (updates of weights and positions), and compare it with mirror descent (updates of weights).We note that the computation of the NI error (2) entails solving two optimization problems on measures, or equivalently in parameter space. We solve each of them by performing 2000 gradient ascent runs with random uniform initialization and selecting the highed minimum final value. This gives a lower bound on the NI error which is precise enough for our purposes. We perform time averaging on the weights of mirror descent and WFR-DA, but not on the positions of WFR-DA because that would incur an $O(t)$ overhead on memory.

**Results.** Wirror descent performs like WFR-DA in low dimensions, but suffers strongly from the curse of dimensionality (Fig. 1). On the other hand, algorithms that incorporate a transport term keep performing well in high dimensions. In particular, WFR-DA is consistently the algorithm with lowest NI error. Notice that the errors in the $n = 50$ and $n = 100$ plots do not differ much, confirming that we reach a mean-field regime.

## 5.2 Training GAN mixtures

We now use WFR-DA to train mixtures of generator networks. We consider the Wasserstein-GAN (Arjovsky et al., 2017) setting. We seek to approximate a distribution $\mathcal{P}_{\text{data}}$ with a distribution $\mathcal{G}_x$, defined as the push-forward of a noise distribution $\mathcal{N}(0, I)$ by a neural-network $g_x$. The discrepancy between $\mathcal{P}_{\text{data}}$ and $\mathcal{G}_x$ is estimated by a neural-network discriminator $f_y$, leading to the problem

$$\min_x \max_y \ell(x, y) \triangleq \mathbb{E}_{a \sim p_{\text{data}}}[f_y(a)] - \mathbb{E}_{\varepsilon \sim \mathcal{N}(0,I)}[f_y(g_x(\varepsilon))].$$

We lift this problem in the space of distributions over the parameters $x$ and $y$ (see §G.4), that we represent through weighted discrete distributions of $\sum_{i=1}^p w_x^{(i)} \delta_{x^{(i)}}$ and $\sum_{j=1}^q w_y^{(j)} \delta_{y^{(j)}}$. We solve

$$\min_{x^{(i)}, w_x \in \triangle_p} \max_{y^{(j)}, w_y \in \triangle_q} \sum_{i=1}^p \sum_{j=1}^q w_x^{(i)} w_y^{(j)} \ell(x^{(i)}, y^{(j)}),$$

using Alg. 2, where $\triangle^q$ is the $q$-dimensional simplex. The optimal generation strategy corresponding to an equilibrium point $(x^{(i)})_i, w_x, (y^{(j)})_j, w_y$ is then to randomly select a generator $g_{x_I}$ with $I$ sampled among $[n]$ with probability $w_x^{(i)}$, and use it to generate $g_{x_I}(\varepsilon)$, with $\varepsilon \sim \mathcal{N}(0, I)$. Training mixtures of generators has been proposed by Ghosh et al. (2018), with a tweaked discriminator loss. Our formulation only involves a lifting in the space of measures, and uses a new training algorithm.

**Results on 2D GMMs.** We first set $\mathcal{P}_{\text{data}}$ to be an 8-mode mixture of Gaussians in two dimensions. We use the original W-GAN loss, with weight cropping for the discriminators $(f_{y^{(j)}})_j$. We measure the interest of using mixtures when a single generator $g_{x^{(i)}}$ cannot fit $\mathcal{P}_{\text{data}}$ (single-layer MLP), and when it can (4-layer MLP). We report results in Fig. 2, measuring the log likelihood of $\mathcal{G}_x$ for the GMM during training. The WFR dynamic is stable even with few particles. When training under-parametrized generators, using mixtures permits faster convergence (in terms of generator updates). In the over-parametrized setting, training a single generator or a mixture of generators perform similarly. WFR-DA is thus useful to train mixtures of simple generators. In this setting, each simple generator identifies modes in the training data, doing data clustering at no cost (Fig. 2 right).

**Results on real data.** We train a mixture of ResNet generators on CIFAR10 and MNIST. We replace the position updates in Alg. 2 by extrapolated Adam steps (Gidel et al., 2019) to achieve faster convergence, and perform grid search over generator and discriminators learning rates. Convergence curves for the best learning rates are displayed in Fig. 3 right, measuring test FID (Heusel et al.,

2017). With a sufficient number of generators and discriminators ($G > 5, D > 2$), the model trains as fast as a normal GAN. WFR-DA is thus stable and efficient even with a reasonable number of particles. Using the discretized WFR versus the Wasserstein flow provides a slight improvement over updating parameters only. As with GMMs, each generator trained with WFR-DA becomes specialised in generating a fraction of the target data, thereby identifying clusters. Those could be used for unsupervised conditional generation of images.

## 6  Conclusions and future work

We have explored non-convex-non-concave, high-dimensional games from the perspective of optimal transport. As with non-convex optimization, framing the problem in terms of measures provides geometric benefits, at the expense of moving into non-Euclidean metric spaces over measures. Our theoretical results establish approximate mean-field convergence for two setups: Langevin Descent-Ascent and WFR D-A, and directly applies to GANs, for mixtures of generators and discriminators.

Despite the positive convergence guarantees our results are qualitative in nature, i.e. without rates. In the entropic case, the unfavorable tradeoff between temperature and convergence of the associated McKean-Vlasov scheme deserves further study, maybe through log-Sobolev-type inequalities (Markowich and Villani, 1999). In the WFR case, we lack a local convergence analysis explaining the benefits of transport observed empirically, perhaps leveraging sharpness Polyak-Łojasiewicz results such as those in (Chizat, 2019) or (Sanjabi et al., 2018). Finally, in our GAN formulation, each generator is associated to a single particle in a high-dimensional product space of all network parameters, which is not scalable to large population sizes that would approximate their mean-field limit. A natural question is to understand to what extent our framework could be combined with specific choices of architecture, as recently studied in (Lei et al., 2019).

## Broader impact

We study algorithms designed to find equilibria in games, provide theoretical guarantees of convergence and test their performance empirically. Among other applications, our results give insight into training algorithms for generative adversarial networks (GANs), which are useful for many relevant tasks such as image generation, image-to-image or text-to-image translation and video prediction. As always, we note that machine learning improvements like ours come in the form of "building machines to do X better". For a sufficiently malicious or ill-informed choice of X, such as surveillance or recidivism prediction, almost any progress in machine learning might indirectly lead to a negative outcome, and our work is not excluded from that.

## Funding disclosure

C. Domingo-Enrich thanks J. De Dios Pont for conversations on the subject. This work is partially supported by the Alfred P. Sloan Foundation, NSF RI-1816753, NSF CAREER CIF 1845360, NSF CHS-1901091, Samsung Electronics, and the Institute for Advanced Study. The work of C. Domingo-Enrich is partially supported by the La Caixa Fellowship. The work of A. Mensch is supported by the European Research Council (ERC project NORIA). The work of G. Rotskoff is supported by the James S. McDonnell Foundation.

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
