[Supplementary Material 1]



## A   Lifted dynamics for the Interacting Wasserstein-Fisher-Rao Gradient Flow

Recall the IWFRGF in (8), which we reproduce here for convenience.

$$\begin{cases} \partial_t \mu_x = \gamma \nabla_x \cdot (\mu_x \nabla_x V_x(\mu_y, x)) - \alpha \mu_x (V_x(\mu_y, x) - \mathcal{L}(\mu_x, \mu_y)), & \mu_x(0) = \mu_{x,0} \\ \partial_t \mu_y = -\gamma \nabla_y \cdot (\mu_y \nabla_y V_y(\mu_x, y)) + \alpha \mu_y (V_y(\mu_x, y) - \mathcal{L}(\mu_x, \mu_y)), & \mu_y(0) = \mu_{y,0} \end{cases}$$

Given $\nu_x \in \mathcal{P}(\mathcal{X} \times \mathbb{R}^+)$ define $\mu_x = \int_{\mathcal{X}} w_x \, d\nu_x(\cdot, w_x) \in \mathcal{P}(\mathcal{X})$, that is

$$\int_{\mathcal{X}} \varphi(x) \, d\mu_x(x) = \int_{\mathcal{X} \times \mathbb{R}^+} w_x \varphi(x) \, d\nu_x(x, w_x),$$

for all $\varphi \in C(\mathcal{X})$. Given $\nu_y \in \mathcal{P}(\mathcal{Y} \times \mathbb{R}^+)$, define $\mu_y = \int_{\mathcal{X}} w_y \, d\nu_y(\cdot, w_y) \in \mathcal{P}(\mathcal{Y})$ analogously. We say that $\nu_x, \nu_y$ are "lifted" measures of $\mu_x, \mu_y$, and reciprocally $\mu_x, \mu_y$ are "projected" measures of $\nu_x, \nu_y$.

By Lemma 1 below, we can view a solution of (8) as the projection of a solution of the following dynamics on the lifted domains $\mathcal{X} \times \mathbb{R}^+$ and $\mathcal{Y} \times \mathbb{R}^+$:

$$\begin{cases} \partial_t \nu_x = \nabla_{w_x,x} \cdot (\nu_x g_{\mu_y}(x, w_x)), & \nu_x(0) = \mu_{x,0} \times \delta_{w_x=1} \\ \partial_t \nu_y = -\nabla_{w_y,y} \cdot (\nu_y g_{\mu_x}(y, w_y)), & \nu_y(0) = \mu_{y,0} \times \delta_{w_y=1} \end{cases} \tag{10}$$

where

$$g_{\mu_y}(x, w_x) = (\alpha w_x(V_x(\mu_y, x) - \mathcal{L}(\mu_x, \mu_y)), \gamma \nabla_x V_x(\mu_y, x))),$$
$$g_{\mu_x}(y, w_y) = (\alpha w_y(V_y(\mu_x, x) - \mathcal{L}(\mu_x, \mu_y)), \gamma \nabla_y V_y(\mu_x, y))).$$

**Lemma 1.** *For a solution $\nu_x : [0, T] \to \mathcal{P}(\mathcal{X} \times \mathbb{R}^+), \nu_y : [0, T] \to \mathcal{P}(\mathcal{Y} \times \mathbb{R}^+)$ of (10), the projections $\mu_x, \mu_y$ are solutions of (8).*

That is, given any $\varphi_x \in \mathcal{C}^1(\mathcal{X}), \varphi_y \in \mathcal{C}^1(\mathcal{Y})$, we have

$$\frac{d}{dt} \int_{\mathcal{X}} \varphi_x(x) \, d\mu_x = -\gamma \int_{\mathcal{X}} \nabla_x \varphi_x(x) \cdot \nabla_x V_x(\mu_y, x) \, d\mu_x - \alpha \int_{\mathcal{X}} \varphi_x(x)(V_x(\mu_y, x) - \mathcal{L}(\mu_x, \mu_y)) \, d\mu_x,$$
$$\frac{d}{dt} \int_{\mathcal{Y}} \varphi_y(y) \, d\mu_y = \gamma \int_{\mathcal{Y}} \nabla_y \varphi_y(y) \cdot \nabla_y V_y(\mu_x, y)) \, d\mu_y + \alpha \int_{\mathcal{Y}} \varphi_y(y)(V_y(\mu_x, y) - \mathcal{L}(\mu_x, \mu_y)) \, d\mu_y,$$
$$\mu_x(0) = \mu_{x,0}, \quad \mu_y(0) = \mu_{y,0} \tag{11}$$

From (10) in the weak form, we obtain that given any $\psi_x \in \mathcal{C}^1(\mathcal{X} \times \mathbb{R}^+), \psi_y \in \mathcal{C}^1(\mathcal{Y} \times \mathbb{R}^+)$,

$$\begin{aligned} \frac{d}{dt} \int_{\mathcal{X} \times \mathbb{R}^+} \psi_x(x, w_x) \, d\nu_x(x, w_x) = \int_{\mathcal{X} \times \mathbb{R}^+} &-\gamma \nabla_x \psi_x(x, w_x) \cdot \nabla_x V_x(\mu_y, x) \\ &- \alpha w_x \frac{d\psi_x}{dw_x}(x, w_x)(V_x(\mu_y, x) - \mathcal{L}(\mu_x, \mu_y)) \, d\mu_x, \\ \frac{d}{dt} \int_{\mathcal{Y} \times \mathbb{R}^+} \psi_y(y, w_y) \, d\nu_y(y, w_y) = \int_{\mathcal{Y} \times \mathbb{R}^+} &\gamma \nabla_y \psi_y(y, w_y) \cdot \nabla_y V_y(\mu_x, y) \\ &+ \alpha w_y \frac{d\psi_y}{dw_y}(y, w_y)(V_y(\mu_x, y) - \mathcal{L}(\mu_x, \mu_y)) \, d\mu_y, \end{aligned} \tag{12}$$
$$\nu_x(0) = \mu_{x,0} \times \delta_{w_x=1}, \quad \nu_y(0) = \mu_{y,0} \times \delta_{w_y=1}.$$

Taking $\psi_x(x, w_x) = w_x \varphi_x(x), \psi_y(y, w_y) = w_y \varphi_y(y)$ yields

$$\begin{aligned} \frac{d}{dt} \int_{\mathcal{X} \times \mathbb{R}^+} w_x \varphi_x(x) \, d\nu_x(x, w_x) = \int_{\mathcal{X} \times \mathbb{R}^+} &-\gamma w_x \nabla_x \varphi_x(x) \cdot \nabla_x V_x(\mu_y, x) \\ &- \alpha w_x \varphi_x(x)(V_x(\mu_y, x) - \mathcal{L}(\mu_x, \mu_y)) \, d\mu_x, \\ \frac{d}{dt} \int_{\mathcal{Y} \times \mathbb{R}^+} w_y \psi_y(y, w_y) \, d\nu_y(y, w_y) = \int_{\mathcal{Y} \times \mathbb{R}^+} &\gamma w_y \nabla_y \varphi_y(y) \cdot \nabla_y V_y(\mu_x, y) \\ &+ \alpha w_y \varphi_y(y)(V_y(\mu_x, y) - \mathcal{L}(\mu_x, \mu_y)) \, d\mu_y. \end{aligned} \tag{13}$$

Notice that (13) is indeed (11).

## B  Continuity and convergence properties of the Nikaido-Isoda error

**Lemma 2.** *The Nikaido-Isoda error NI : $\mathcal{P}(\mathcal{X}) \times \mathcal{P}(\mathcal{Y}) \to \mathbb{R}$ defined in (2) is continuous when we endow $\mathcal{P}(\mathcal{X}), \mathcal{P}(\mathcal{Y})$ with the topology of weak convergence. Specifically, it is Lip($\ell$)-Lipschitz when we use the distance $\mathcal{W}_1(\mu_x, \mu'_x) + \mathcal{W}_1(\mu_y, \mu'_y)$ between $(\mu_x, \mu_y)$ and $(\mu'_x, \mu'_y)$ in $\mathcal{P}(\mathcal{X}) \times \mathcal{P}(\mathcal{Y})$.*

*Proof.* For any $\mu_y$, the function $V_x(\mu_y, \cdot) : \mathcal{X} \to \mathbb{R}$ defined as $x \mapsto \int \ell(x,y) \, d\mu_y$ is continuous and it has the same Lipschitz constant $\mathrm{Lip}(\ell)$ as $\ell$. Hence, for any $\mu_x, \mu'_x \in \mathcal{P}(\mathcal{X})$,

$$
\sup_{\mu_y \in \mathcal{P}(\mathcal{Y})} \mathcal{L}(\mu_x, \mu_y) - \sup_{\mu_y \in \mathcal{P}(\mathcal{Y})} \mathcal{L}(\mu'_x, \mu_y) = \sup_{\mu_y \in \mathcal{P}(\mathcal{Y})} \int V_x(\mu_y, x) d\mu_x - \sup_{\mu_y \in \mathcal{P}(\mathcal{Y})} \int V_x(\mu_y, x) d\mu'_x
$$

$$
\leqslant \sup_{\mu_y \in \mathcal{P}(\mathcal{Y})} \int V_x(\mu_y, x) d\mu'_x + \sup_{\mu_y \in \mathcal{P}(\mathcal{Y})} \int V_x(\mu_y, x) d(\mu_x - \mu'_x) - \sup_{\mu_y \in \mathcal{P}(\mathcal{Y})} \int V_x(\mu_y, x) d\mu'_x
$$

$$
= \sup_{\mu_y \in \mathcal{P}(\mathcal{Y})} \int V_x(\mu_y, x) d(\mu_x - \mu'_x) \leqslant \mathrm{Lip}(\ell) \mathcal{W}_1(\mu_x, \mu'_x)
$$

The same inequality interchanging the roles of $\mu_x, \mu'_x$ shows that $|\sup_{\mu_y \in \mathcal{P}(\mathcal{Y})} \mathcal{L}(\mu_x, \mu_y) - \sup_{\mu_y \in \mathcal{P}(\mathcal{Y})} \mathcal{L}(\mu'_x, \mu_y)| \leqslant \mathrm{Lip}(\ell) \mathcal{W}_1(\mu_x, \mu'_x)$ holds. An analogous reasoning for $\ell(\mu_x, \cdot) : \mathcal{Y} \to \mathbb{R}$ and the triangle inequality complete the proof. $\square$

**Lemma 3.** *Suppose that $(\mu_x^n)_{n \in \mathbb{N}}$ is a sequence of random elements valued in $\mathcal{P}(\mathcal{X})$ such that*

$$
\mathbb{E}[\mathcal{W}_2^2(\mu_x^n, \mu_x)] \xrightarrow{n \to \infty} 0,
$$

*where $\mu_x \in \mathcal{P}(X)$. Analogously, suppose that $(\mu_y^n)_{n \in \mathbb{N}}$ is a sequence of random elements valued in $\mathcal{P}(\mathcal{Y})$ such that*

$$
\mathbb{E}[\mathcal{W}_2^2(\mu_y^n, \mu_y)] \xrightarrow{n \to \infty} 0,
$$

*where $\mu_y \in \mathcal{P}(Y)$.*

*Then,*

$$
\mathbb{E}[|NI(\mu_x^n, \mu_y^n) - NI(\mu_x, \mu_y)|] \xrightarrow{n \to \infty} 0
$$

*Proof.* First,
$$
\mathbb{E}[\mathcal{W}_1(\mu_x^n, \mu_x)] \leqslant \mathbb{E}[\mathcal{W}_2(\mu_x^n, \mu_x)] \leqslant \left( \mathbb{E}[\mathcal{W}_2^2(\mu_x^n, \mu_x)] \right)^{1/2}, \tag{14}
$$
which results from two applications of the Cauchy-Schwarz inequality on the appropriate scalar products. An analogous inequality holds for $\mathbb{E}[\mathcal{W}_1(\mu_y^n, \mu_y)]$. Hence, by Lemma 2,

$$
\mathbb{E}[|\mathrm{NI}(\mu_x^n, \mu_y^n) - \mathrm{NI}(\mu_x, \mu_y)|] \leqslant \mathrm{Lip}(\ell) \mathbb{E}[\mathcal{W}_1(\mu_x^n, \mu_x) + \mathcal{W}_1(\mu_y^n, \mu_y)]
$$

$$
\leqslant \mathrm{Lip}(\ell) \left( \left( \mathbb{E}[\mathcal{W}_2^2(\mu_x^n, \mu_x)] \right)^{1/2} + \left( \mathbb{E}[\mathcal{W}_2^2(\mu_x^n, \mu_x)] \right)^{1/2} \right)
$$

$$
\leqslant \mathrm{Lip}(\ell) \sqrt{2} \left( \mathbb{E}[\mathcal{W}_2^2(\mu_x^n, \mu_x)] + \mathbb{E}[\mathcal{W}_2^2(\mu_x^n, \mu_x)] \right)^{1/2},
$$

where the second inequality uses (14) and the third inequality is another application of the Cauchy-Schwarz inequality. Since the right hand side converges to 0 by assumption, this concludes the proof. $\square$

## C  Proof of Theorem 1

We restate Theorem 1 for convenience.

**Theorem 1.** *Suppose that Asm. 1 holds, that $\ell \in C^{2,\alpha}(\mathcal{X} \times \mathcal{Y})$ for some $\alpha \in (0,1)$ and that the initial measures $\mu_{x,0}, \mu_{y,0}$ have densities in $L^1(\mathcal{X}), L^1(\mathcal{Y})$. If a solution $(\mu_x(t), \mu_y(t))$ of the ERIWGF (7) converges in time, it must converge to the point $(\hat{\mu}_x, \hat{\mu}_y)$ which is the unique fixed point of the problem*

$$
\rho_x(x) = \frac{1}{Z_x} e^{-\beta \int \ell(x,y) \, d\mu_y(y)}, \quad \rho_y(y) = \frac{1}{Z_y} e^{\beta \int \ell(x,y) \, d\mu_x(x)}.
$$

*$(\hat{\mu}_x, \hat{\mu}_y)$ is an $\varepsilon$-Nash equilibrium of the game given by $\mathcal{L}$ when $\beta \geqslant \frac{4}{\varepsilon} \log \left( 2 \frac{1 - V_\delta}{V_\delta} (2K_\ell/\varepsilon - 1) \right)$, where $K_\ell := \max_{x,y} \ell(x,y) - \min_{x,y} \ell(x,y)$ is the length of the range of $\ell$, $\delta := \varepsilon/(2Lip(\ell))$ and $V_\delta$ is a lower bound on the volume of a ball of radius $\delta$ in $\mathcal{X}, \mathcal{Y}$.*

Theorem 1 is a consequence of the following three results, which we prove separately.

**Theorem 4.** *Assume $\mathcal{X}, \mathcal{Y}$ are compact Polish metric spaces equipped with canonical Borel measures, and that $\ell$ is a continuous function on $\mathcal{X} \times \mathcal{Y}$. Let us consider the fixed point problem*

$$\begin{cases} \rho_x(x) &= \frac{1}{Z_x} e^{-\beta \int \ell(x,y) \, d\mu_y(y)}, \\ \rho_y(y) &= \frac{1}{Z_y} e^{\beta \int \ell(x,y) \, d\mu_x(x)}, \end{cases}$$

*where $Z_x$ and $Z_y$ are normalization constants and $\rho_x, \rho_y$ are the densities of $\mu_x, \mu_y$. This fixed point problem has a unique solution $(\hat{\mu}_x, \hat{\mu}_y)$ that is also the unique Nash equilibrium of the game given by $\mathcal{L}_\beta(\mu_x, \mu_y) \triangleq \mathcal{L}(\mu_x, \mu_y) + \beta^{-1}(H(\mu_y) - H(\mu_x))$.*

**Theorem 5.** *Let $K_\ell := \max_{x,y} \ell(x,y) - \min_{x,y} \ell(x,y)$ be the length of the range of $\ell$. Let $\varepsilon > 0$, $\delta := \varepsilon/(2Lip(\ell))$ and $V_\delta$ be a lower bound on the volume of a ball of radius $\delta$ in $\mathcal{X}, \mathcal{Y}$. Then the solution $(\hat{\mu}_x, \hat{\mu}_y)$ of (9) is an $\varepsilon$-Nash equilibrium of the game given by $\mathcal{L}$ when*

$$\beta \geqslant \frac{4}{\varepsilon} \log \left( 2 \frac{1 - V_\delta}{V_\delta} (2K_\ell/\varepsilon - 1) \right).$$

**Theorem 6.** *Suppose that Asm. 1 holds and $\ell \in C^{2,\alpha}(\mathcal{X} \times \mathcal{Y})$ for some $\alpha \in (0,1)$, i.e. the second derivatives of $\ell$ are $\alpha$-Hölder. Then, there exists only one stationary solution of the ERIWGF (7) and it is the solution of the fixed point problem (9).*

## C.1 Proof of Theorem 4: Preliminaries

**Definition 2** (Upper hemicontinuity). *A set-valued function $\varphi : X \to 2^Y$ is upper hemicontinuous if for every open set $W \subset Y$, the set $\{x | \varphi(x) \subset W\}$ is open.*

Alternatively, set-valued functions can be seen as correspondences $\Gamma : X \to Y$. The graph of $\Gamma$ is $\mathrm{Gr}(\Gamma) = \{(a,b) \in X \times Y | b \in \Gamma(a)\}$. If $\Gamma$ is upper hemicontinuous, then $\mathrm{Gr}(\Gamma)$ is closed. If $Y$ is compact, the converse is also true.

**Definition 3** (Kakutani map). *Let $X$ and $Y$ be topological vector spaces and $\varphi : X \to 2^Y$ be a set-valued function. If $Y$ is convex, then $\varphi$ is termed a Kakutani map if it is upper hemicontinuous and $\varphi(x)$ is non-empty, compact and convex for all $x \in X$.*

**Theorem 7** (Kakutani-Glicksberg-Fan). *Let $S$ be a non-empty, compact and convex subset of a Hausdorff locally convex topological vector space. Let $\varphi : S \to 2^S$ be a Kakutani map. Then $\varphi$ has a fixed point.*

**Definition 4** (Lower semi-continuity). *Suppose $X$ is a topological space, $x_0$ is a point in $X$ and $f : X \to \mathbb{R} \cup \{-\infty, \infty\}$ is an extended real-valued function. We say that $f$ is lower semi-continuous (l.s.c.) at $x_0$ if for every $\varepsilon > 0$ there exists a neighborhood $U$ of $x_0$ such that $f(x) \geqslant f(x_0) - \varepsilon$ for all $x$ in $U$ when $f(x_0) < +\infty$, and $f(x)$ tends to $+\infty$ as $x$ tends towards $x_0$ when $f(x_0) = +\infty$.*

We can also characterize lower-semicontinuity in terms of level sets. A function is lower semi-continuous if and only if all of its lower level sets $\{x \in X : f(x) \leqslant \alpha\}$ are closed. This property will be useful.

**Theorem 8** (Weierstrass theorem for l.s.c. functions). *Let $f : T \to (-\infty, +\infty]$ be a l.s.c. function on a compact Hausdorff topological space $T$. Then $f$ attains its infimum over $T$, i.e. there exists a minimum of $f$ in $T$.*

*Proof.* Proof. Let $\alpha_0 = \inf f(T)$. If $\alpha_0 = +\infty$, then $f$ is infinite and the assertion trivially holds. Let $\alpha_0 < +\infty$. Then, for each real $\alpha > \alpha_0$, the set $\{f \leqslant \alpha\}$ is closed and nonempty. Any finite collection of such sets has a nonempty intersection. By compactness, also the set $\bigcap_{\alpha > \alpha_0} \{f \leqslant \alpha\} = \{f \leqslant \alpha_0\} = f^{-1}(\alpha_0)$ is nonempty. (In particular, this implies that $\alpha_0$ is finite.) $\qquad\square$

**Remark 1.** *By Prokhorov's theorem, since $\mathcal{X}$ and $\mathcal{Y}$ are compact separable metric spaces, $\mathcal{P}(\mathcal{X})$ and $\mathcal{P}(\mathcal{Y})$ are compact in the topology of weak convergence.*

## C.2 Proof of Theorem 4: Existence

Lemma 4 and 5 are intermediate results, and Lemma 6 shows existence of the solution.

**Lemma 4.** *For any $\mu_y \in \mathcal{P}(\mathcal{Y})$, $\mathcal{L}_\beta(\cdot, \mu_y) : \mathcal{P}(\mathcal{X}) \to \mathbb{R}$ is lower semicontinuous, and it achieves a unique minimum in $\mathcal{P}(\mathcal{X})$. Moreover, the minimum $m_x(\mu_y)$ is absolutely continuous with respect to the Borel measure, it has full support and its density takes the form*

$$\frac{dm_x(\mu_y)}{dx}(x) = \frac{1}{Z_{\mu_y}} e^{-\beta \int L(x,y)d\mu_y}, \tag{15}$$

*where $Z_{\mu_y}$ is a normalization constant.*

*Analogously, for any $\mu_x \in \mathcal{P}(\mathcal{X})$, $-\mathcal{L}_\beta(\mu_x, \cdot) : \mathcal{P}(\mathcal{Y}) \to \mathbb{R}$ is lower semicontinuous, and it achieves a unique minimum in $\mathcal{P}(\mathcal{Y})$. The minimum $m_y(\mu_x)$ is absolutely continuous with respect to the Borel measure, it has full support and its density takes the form*

$$\frac{dm_y(\mu_x)}{dy}(y) = \frac{1}{Z_{\mu_x}} e^{\beta \int L(x,y)d\mu_x},$$

*where $Z_{\mu_x}$ is a normalization constant.*

*Proof.* We will prove the result for $\mathcal{L}_\beta(\cdot, \mu_y)$, as the other one is analogous. Let $dx$ denote the canonical Borel measure on $\mathcal{X}$, and let $\tilde{p}$ be the probability measure proportional to the canonical Borel measure, i.e. $\frac{d\tilde{p}}{dx} = \frac{1}{\text{vol}(\mathcal{X})}$. Notice that $\text{vol}(\mathcal{X})$ is by definition the value of the canonical Borel measure on the whole $\mathcal{X}$. We rewrite

$$
\begin{aligned}
\mathcal{L}_\beta(\mu_x, \mu_y) &= \iint \ell(x,y)d\mu_y d\mu_x + \beta^{-1} \int \log\left(\frac{d\mu_x}{dx}\right)d\mu_x + \beta^{-1}H(\mu_y) \\
&= \iint \ell(x,y)d\mu_y d\mu_x + \beta^{-1} \int \log\left(\frac{d\mu_x}{d\tilde{p}}\frac{d\tilde{p}}{dx}\right)d\mu_x + \beta^{-1}H(\mu_y) \\
&= \iint \left(\ell(x,y) - \beta^{-1}\log\left(\text{vol}(\mathcal{X})\right)\right)d\mu_y d\mu_x + \beta^{-1}\int \log\left(\frac{d\mu_x}{d\tilde{p}}\right)d\mu_x + \beta^{-1}H(\mu_y)
\end{aligned}
$$

Notice that the first term in the right hand side is a lower semi-continuous (in weak convergence topology) functional in $\mu_x$ when $\mu_y$ is fixed. That is because it is a linear functional in $\mu_x$ with a continuous integrand, which implies that it is continuous in the weak convergence topology. The second to last term can be seen as the relative entropy (or Kullback-Leibler divergence) between $\mu_x$ and $\tilde{p}$:

$$H_{\tilde{p}}(\mu_x) := \int \log\left(\frac{d\mu_x}{d\tilde{p}}\right)d\mu_x$$

The relative entropy $H_{\tilde{p}}(\mu_x)$ is a lower semi-continuous functional with respect to $\mu_x$ (see Theorem 1 of Posner (1975), which proves a stronger statement: joint semi-continuity with respect to both measures).

Therefore, we conclude that $\mathcal{L}_\beta(\cdot, \mu_y)$ (with $\mu_y \in \mathcal{P}(\mathcal{Y})$ fixed) is a l.s.c. functional on $\mathcal{P}(\mathcal{X})$. By Theorem 8 and using the compactness of $\mathcal{P}(\mathcal{X})$, there exists a minimum of $\mathcal{L}_\beta(\cdot, \mu_y)$ in $\mathcal{P}(\mathcal{X})$.

Denote a minimum of $\mathcal{L}_\beta(\cdot, \mu_y)$ by $\hat{\mu}_x$. $\hat{\mu}_x$ must be absolutely continuous, because otherwise $-\beta^{-1}H(\hat{\mu}_x)$ would take an infinite value. By the Euler-Lagrange equations for functionals on probability measures, a necessary condition for $\hat{\mu}_x$ to be a minimum of $\mathcal{L}_\beta(\cdot, \mu_y)$ is that the first variation $\frac{\delta\mathcal{L}_\beta(\cdot, \mu_y)}{\delta\mu_x}(\hat{\mu}_x)(x)$ must take a constant value for all $x \in \text{supp}(\hat{\mu}_x)$ and values larger or equal outside of $\text{supp}(\hat{\mu}_x)$. The intuition behind this is that otherwise a zero-mean signed measure with positive mass on the minimizers of $\frac{\delta\mathcal{L}_\beta(\cdot, \mu_y)}{\delta\mu_x}(\hat{\mu}_x)$ and negative mass on the maximizers would provide a direction of decrease of the functional. We compute the first variation at $\hat{\mu}_x$:

$$
\begin{aligned}
\frac{\delta\mathcal{L}_\beta(\cdot, \mu_y)}{\delta\mu_x}(\hat{\mu}_x)(x) &= \frac{\delta}{\delta\mu_x}\left(\int L(x,y)d\mu_y d\mu_x - \beta^{-1}H(\hat{\mu}_x) + \beta^{-1}H(\mu_y)\right) \\
&= \int L(x,y)d\mu_y + \beta^{-1}\log\left(\frac{d\hat{\mu}_x}{dx}(x)\right),
\end{aligned}
$$

We equate $\int \ell(x,y)d\mu_y + \beta^{-1}\log(\frac{d\hat{\mu}_x}{dx}(x)) = K$, $\forall x \in \text{supp}(\hat{\mu}_x)$, where $K$ is a constant. The first variation must take values larger or equal than $K$ outside of $\text{supp}(\hat{\mu}_x)$, but since $\log(\frac{d\hat{\mu}_x}{dx}(x)) = -\infty$ outside of $\text{supp}(\hat{\mu}_x)$, we obtain that $\text{supp}(\hat{\mu}_x) = \mathcal{X}$. Then, for all $x \in \mathcal{X}$,

$$\frac{d\hat{\mu}_x}{dx}(x) = e^{-\beta \int L(x,y)d\mu_y + \beta K} = \frac{1}{Z_{\mu_y}} e^{-\beta \int L(x,y)d\mu_y}$$

where $Z_{\mu_y}$ is a normalization constant obtained from imposing $\int \frac{d\hat{\mu}_x}{dx}(x)\,dx = \int 1\,d\hat{\mu}_x = 1$. Since the necessary condition for optimality specifies a unique measure and the minimum exists, we obtain that $m_x(\mu_y) = \hat{\mu}_x$ is the unique minimum. An analogous argument holds for $m_y(\hat{\mu}_x)$ □

**Lemma 5.** *Suppose that the measures $(\mu_{y,n})_{n \in \mathbb{N}}$ and $\mu_y$ are in $\mathcal{P}(\mathcal{Y})$. Recall the definition of $m_x : \mathcal{P}(\mathcal{Y}) \to \mathcal{P}(\mathcal{X})$ in equation (15). If $(\mu_{y,n})_{n \in \mathbb{N}}$ converges weakly to $\mu_y$, then $(m_x(\mu_{y,n}))_{n \in \mathbb{N}}$ converges weakly to $m_x(\mu_y)$, i.e. $m_x$ is a continuous mapping when we endow $\mathcal{P}(\mathcal{Y})$ and $\mathcal{P}(\mathcal{X})$ with their weak convergence topologies.*

*The same thing holds for $m_y$ and measures $(\mu_{x,n})_{n \in \mathbb{N}}$ and $\mu_x$ on $\mathcal{X}$.*

*Proof.* Given $x \in \mathcal{X}$, we have $\int \ell(x,y)d\mu_{y,n} \to \int \ell(x,y)d\mu_y$, because $\ell(x,\cdot)$ is a continuous bounded function on $\mathcal{Y}$. By continuity of the exponential function, we have that for all $x \in \mathcal{X}$, $e^{-\beta \int \ell(x,y)d\mu_{y,n}} \to e^{-\beta \int \ell(x,y)d\mu_y}$. Using the dominated convergence theorem,

$$\int_{\mathcal{X}} e^{-\beta \int \ell(x,y)d\mu_{y,n}} dx \to \int_{\mathcal{X}} e^{-\beta \int \ell(x,y)d\mu_y} dx$$

We need to find a dominating function. It is easy, because $\forall n \in \mathbb{N}$, $\forall x \in \mathcal{X}$, $e^{-\beta \int \ell(x,y)d\mu_{y,n}} \leqslant e^{-\beta \min_{(x,y) \in \mathcal{X} \times \mathcal{Y}} \ell(x,y)}$. And $\int_{\mathcal{X}} e^{-\beta \min_{(x,y) \in \mathcal{X} \times \mathcal{Y}} \ell(x,y)} dx = e^{-\beta \min_{(x,y) \in \mathcal{X} \times \mathcal{Y}} \ell(x,y)} \text{vol}(\mathcal{X}) < \infty$. By the Portmanteau theorem, we just need to prove that for all continuity sets $B$ of $m_x(\mu_y)$, we have $m_x(\mu_{y,n})(B) \to m_x(\mu_y)(B)$. This translates to

$$\frac{\int_B e^{-\beta \int \ell(x,y)d\mu_{y,n}} dx}{\int_{\mathcal{X}} e^{-\beta \int \ell(x,y)d\mu_{y,n}} dx} \to \frac{\int_B e^{-\beta \int \ell(x,y)d\mu_y} dx}{\int_{\mathcal{X}} e^{-\beta \int \ell(x,y)d\mu_y} dx}$$

We have proved that the denominators converge appropriately, and the numerator converges as well using the same reasoning with dominated convergence. And both the numerators and the denominators are positive and the numerator is always smaller denominator, the quotient must converge. □

**Lemma 6.** *There exists a solution of (9), which is the Nash equilibrium of the game given by $\mathcal{L}_\beta$.*

*Proof.* We use Theorem 7 on the set $\mathcal{P}(\mathcal{X}) \times \mathcal{P}(\mathcal{Y})$, with the map $m : \mathcal{P}(\mathcal{X}) \times \mathcal{P}(\mathcal{Y}) \to \mathcal{P}(\mathcal{X}) \times \mathcal{P}(\mathcal{Y})$ given by $m(\mu_x, \mu_y) = (m_x(\mu_y), m_y(\mu_x))$. The only condition to check is upper hemicontinuity of $m$. By Lemma 5 we know that $m_x, m_y$ are continuous, and since continuous functions are upper hemicontinuous as set valued functions, this concludes the argument. Indeed, we could have used Tychonoff's theorem, which is similar to Theorem 7 but for single-valued functions. □

### C.3 Proof of Theorem 4: Uniqueness

**Lemma 7.** *The solution of (9) is unique.*

*Proof.* The argument is analogous to the proof of Theorem 2 of Rosen (1965). Suppose $(\mu_{x,1}, \mu_{y,1})$ and $(\mu_{x,2}, \mu_{y,2})$ are two different solutions of (9). We use the notation $F_1(\mu_x, \mu_y) = \mathcal{L}_\beta(\mu_x, \mu_y)$, $F_2(\mu_x, \mu_y) = -\mathcal{L}_\beta(\mu_x, \mu_y)$. Hence, there exist constants $K_{x,1}, K_{y,1}, K_{x,2}, K_{y,2}$ such that

$$\frac{\delta F_1}{\delta \mu_x}(\mu_{x,1}, \mu_{y,1})(x) + K_{x,1} = 0, \frac{\delta F_2}{\delta \mu_y}(\mu_{x,1}, \mu_{y,1})(y) + K_{y,1} = 0,$$

$$\frac{\delta F_1}{\delta \mu_x}(\mu_{x,2}, \mu_{y,2})(x) + K_{x,2} = 0, \frac{\delta F_2}{\delta \mu_y}(\mu_{x,2}, \mu_{y,2})(y) + K_{y,2} = 0$$

On the one hand, we know that

$$\int \frac{\delta F_1}{\delta \mu_x}(\mu_{x,1},\mu_{y,1})(x)\, d(\mu_{x,2}-\mu_{x,1}) + \int \frac{\delta F_2}{\delta \mu_y}(\mu_{x,1},\mu_{y,1})(y)\, d(\mu_{y,2}-\mu_{y,1})$$

$$+ \int \frac{\delta F_1}{\delta \mu_x}(\mu_{x,2},\mu_{y,2})(x)\, d(\mu_{x,1}-\mu_{x,2}) + \int \frac{\delta F_2}{\delta \mu_y}(\mu_{x,2},\mu_{y,2})(y)\, d(\mu_{y,1}-\mu_{y,2}) \tag{16}$$

$$= -\int K_{x,1}\, d(\mu_{x,2}-\mu_{x,1}) - \int K_{y,1}\, d(\mu_{y,2}-\mu_{y,1})$$

$$- \int K_{x,2}\, d(\mu_{x,1}-\mu_{x,2}) - \int K_{y,2}\, d(\mu_{y,1}-\mu_{y,2}) = 0$$

We will now prove that the left hand side of (16) must be strictly larger than 0, reaching a contradiction. We can write

$$\frac{\delta F_1}{\delta \mu_x}(\mu_{x,2},\mu_{y,2})(x) - \frac{\delta F_1}{\delta \mu_x}(\mu_{x,1},\mu_{y,1})(x) = \int L(x,y)\, d(\mu_{y,2}-\mu_{y,1})$$

$$+ \beta^{-1}(\log(\mu_{x,2}(x)) - \log(\mu_{x,1}(x))),$$

$$\frac{\delta F_2}{\delta \mu_y}(\mu_{x,2},\mu_{y,2})(x) - \frac{\delta F_2}{\delta \mu_y}(\mu_{x,1},\mu_{y,1})(x) = -\int L(x,y)\, d(\mu_{x,2}-\mu_{x,1})$$

$$+ \beta^{-1}(\log(\mu_{y,2}(x)) - \log(\mu_{y,1}(x)))$$

Hence, we rewrite the left hand side of (16) as

$$\iint L(x,y)\, d(\mu_{y,2}-\mu_{y,1})d(\mu_{x,2}-\mu_{x,1}) + \beta^{-1}\int(\log(\mu_{x,2}(x)) - \log(\mu_{x,1}(x)))\, d(\mu_{x,2}-\mu_{x,1})$$

$$- \iint L(x,y)\, d(\mu_{x,2}-\mu_{x,1})d(\mu_{y,2}-\mu_{y,1}) + \beta^{-1}\int(\log(\mu_{y,2}(x)) - \log(\mu_{y,1}(x)))\, d(\mu_{y,2}-\mu_{y,1})$$

$$= \beta^{-1}(H_{\mu_{x,1}}(\mu_{x,2}) + H_{\mu_{x,2}}(\mu_{x,1}) + H_{\mu_{y,1}}(\mu_{y,2}) + H_{\mu_{y,1}}(\mu_{y,2})).$$

Since the relative entropy is always non-negative and zero only if the two measures are equal, we have reached the desired contradiction. $\qquad\square$

### C.4   Proof of Theorem 5

We will use the shorthand $V_x(x) = V_x(\hat{\mu}_y)(x) = \int \mathcal{L}(x,y)d\hat{\mu}_y$, $V_y(y) = V_y(\hat{\mu}_x)(y) = \int \mathcal{L}(x,y)d\hat{\mu}_x$. Since $\ell : \mathcal{X} \times \mathcal{Y} \to \mathbb{R}$ is a continuous function on a compact metric space, it is uniformly continuous. Hence,

$$\forall \varepsilon > 0, \exists \delta > 0 \text{ st. } \sqrt{d(x,x')^2 + d(y,y')^2} < \delta \implies |\ell(x,y) - \ell(x',y')| < \varepsilon$$

Which means that

$$d(x,x') < \delta \implies |V_x(x) - V_x(x')| = \left|\int (\ell(x,y) - \ell(x',y))dy\right| < \varepsilon$$

This proves that $V_x$ is uniformly continuous on $\mathcal{X}$ (and $V_y$ is uniformly continuous on $\mathcal{Y}$ using the same argument).

We can write the Nikaido-Isoda function of the game with loss $\mathcal{L}$ (equation (2)) evaluated at $(\hat{\mu}_x, \hat{\mu}_y)$ as

$$\mathrm{NI}(\hat{\mu}_x, \hat{\mu}_y) := \mathcal{L}(\hat{\mu}_x, \hat{\mu}_y) - \min_{\mu'_x}\{\mathcal{L}(\mu'_x, \hat{\mu}_y)\} + (-\mathcal{L}(\hat{\mu}_x, \hat{\mu}_y) + \max_{\mu'_y}\{\mathcal{L}(\hat{\mu}_x, \mu'_y)\})$$

$$= \frac{\int V_x(x)e^{-\beta V_x(x)}dx}{\int e^{-\beta V_x(x)}dx} - \min_{x \in \mathcal{C}_1} V_x(x) + \frac{-\int V_y(y)e^{\beta V_y(y)}dy}{\int e^{\beta V_y(y)}dy} + \max_{y \in \mathcal{C}_2} V_y(y) \tag{17}$$

The second equality follows from the definitions of $\mathcal{L}, V_x, V_y$. We observe that in the right-most expression the first two terms and the last two terms are analogous. Let us show the first two terms

can be made smaller than an arbitrary $\varepsilon > 0$ by taking $\beta$ large enough; the last two will be dealt with in an analogous manner. Let us define $\tilde{V}_x(x) = V_x(x) - \min_{x' \in \mathcal{C}_1} V_x(x')$.

$$\frac{\int V_x(x)e^{-\beta V_x(x)}dx}{\int e^{-\beta V_x(x)}dx} - \min_{x \in \mathcal{C}_1} V_x(x) = \frac{\int (V_x(x) - \min_{x' \in \mathcal{C}_1} V_x(x'))e^{-\beta V_x(x)}dx}{\int e^{-\beta V_x(x)}dx}$$

$$= \frac{\int \tilde{V}_x(x)e^{-\beta V_x(x)}\left(\mathbb{1}_{\{\tilde{V}_x(x) \leqslant \varepsilon/2\}} + \mathbb{1}_{\{\varepsilon/2 < \tilde{V}_x(x) \leqslant \varepsilon\}} + \mathbb{1}_{\{\varepsilon < \tilde{V}_x(x)\}}\right)dx}{\int e^{-\beta V_x(x)}\mathbb{1}_{\{\tilde{V}_x(x) \leqslant \varepsilon/2\}}dx + \int e^{-\beta V_x(x)}\mathbb{1}_{\{\varepsilon/2 < \tilde{V}_x(x) \leqslant \varepsilon\}}dx + \int e^{-\beta V_x(x)}\mathbb{1}_{\{\varepsilon < \tilde{V}_x(x)\}}dx}$$

(18)

Let us define

$$q_{\{\tilde{V}_x(x) \leqslant \varepsilon/2\}} = \int e^{-\beta V_x(x)}\mathbb{1}_{\{\tilde{V}_x(x) \leqslant \varepsilon/2\}}dx,$$

and $q_{\{\varepsilon/2 < \tilde{V}_x(x) \leqslant \varepsilon\}}$ and $q_{\{\varepsilon < \tilde{V}_x(x)\}}$ analogously.

Similarly, let

$$r_{\{\tilde{V}_x(x) \leqslant \varepsilon/2\}} = \int \tilde{V}_x(x)e^{-\beta V_x(x)}\mathbb{1}_{\{\tilde{V}_x(x) \leqslant \varepsilon/2\}}dx,$$

and $r_{\{\varepsilon/2 < \tilde{V}_x(x) \leqslant \varepsilon\}}$ and $r_{\{\varepsilon < \tilde{V}_x(x)\}}$ analogously.

Let

$$\tilde{p} = \frac{q_{\{\varepsilon/2 < \tilde{V}_x(x) \leqslant \varepsilon\}}}{q_{\{\tilde{V}_x(x) \leqslant \varepsilon/2\}} + q_{\{\varepsilon/2 < \tilde{V}_x(x) \leqslant \varepsilon\}} + q_{\{\varepsilon < \tilde{V}_x(x)\}}}$$

Then, we can rewrite the right-most expression of (18) as

$$\frac{r_{\{\tilde{V}_x(x) \leqslant \varepsilon/2\}} + r_{\{\varepsilon/2 < \tilde{V}_x(x) \leqslant \varepsilon\}} + r_{\{\varepsilon < \tilde{V}_x(x)\}}}{q_{\{\tilde{V}_x(x) \leqslant \varepsilon/2\}} + q_{\{\varepsilon/2 < \tilde{V}_x(x) \leqslant \varepsilon\}} + q_{\{\varepsilon < \tilde{V}_x(x)\}}}$$

$$= \tilde{p}\frac{r_{\{\varepsilon/2 < \tilde{V}_x(x) \leqslant \varepsilon\}}}{q_{\{\varepsilon/2 < \tilde{V}_x(x) \leqslant \varepsilon\}}} + (1 - \tilde{p})\frac{r_{\{\tilde{V}_x(x) \leqslant \varepsilon/2\}} + r_{\{\varepsilon < \tilde{V}_x(x)\}}}{q_{\{\tilde{V}_x(x) \leqslant \varepsilon/2\}} + q_{\{\varepsilon < \tilde{V}_x(x)\}}}$$

(19)

Since $\tilde{V}(x) \leqslant \varepsilon$ in the set $\{x | \varepsilon/2 < \tilde{V}_x(x) \leqslant \varepsilon\}$, $r_{\{\varepsilon/2 < \tilde{V}_x(x) \leqslant \varepsilon\}}/q_{\{\varepsilon/2 < \tilde{V}_x(x) \leqslant \varepsilon\}} \leqslant \varepsilon$.

Let $x_{\min}$ be such that $V(x_{\min}) = \min_{x \in C_1} V(x)$ (possibly not unique). By uniform continuity of $V_x$, we know there exists $\delta > 0$ (dependent only on $\varepsilon$) such that $B(x_{\min}, \delta) \subseteq \{x | \tilde{V}_x(x) \leqslant \varepsilon/2\}$. The following inequalities hold:

$$r_{\{\tilde{V}_x(x) \leqslant \varepsilon/2\}} \leqslant \frac{\varepsilon}{2}q_{\{\tilde{V}_x(x) \leqslant \varepsilon/2\}},$$

$$r_{\{\varepsilon < \tilde{V}_x(x)\}} \leqslant (\max_{x \in \mathcal{C}_1} V_x(x) - \min_{x \in \mathcal{C}_1} V_x(x))q_{\{\varepsilon < \tilde{V}_x(x)\}} \leqslant (\max_{x,y} L(x,y) - \min_{x,y} L(x,y))q_{\{\varepsilon < \tilde{V}_x(x)\}}$$

$$= K_L q_{\{\varepsilon < \tilde{V}_x(x)\}}.$$

(20)

where we define $K_\ell = \max_{x,y} \ell(x,y) - \min_{x,y} \ell(x,y)$. Using (20), we obtain

$$\frac{r_{\{\tilde{V}_x(x) \leqslant \varepsilon/2\}} + r_{\{\varepsilon < \tilde{V}_x(x)\}}}{q_{\{\tilde{V}_x(x) \leqslant \varepsilon/2\}} + q_{\{\varepsilon < \tilde{V}_x(x)\}}} \leqslant \frac{\frac{\varepsilon}{2}q_{\{\tilde{V}_x(x) \leqslant \varepsilon/2\}} + K_L q_{\{\varepsilon < \tilde{V}_x(x)\}}}{q_{\{\tilde{V}_x(x) \leqslant \varepsilon/2\}} + q_{\{\varepsilon < \tilde{V}_x(x)\}}}.$$

If the right-hand side is smaller or equal than $\varepsilon$, then equation (19) would be smaller than $\varepsilon$ and the proof would be concluded. For that to happen, we need $(K_\ell - \varepsilon)q_{\{\varepsilon < \tilde{V}_x(x)\}} \leqslant \frac{\varepsilon}{2}q_{\{\tilde{V}_x(x) \leqslant \varepsilon/2\}} \iff q_{\{\tilde{V}_x(x) \leqslant \varepsilon/2\}}/q_{\{\varepsilon < \tilde{V}_x(x)\}} \geqslant 2(K_\ell/\varepsilon - 1)$. The following bounds hold:

$$q_{\{\tilde{V}_x(x) \leqslant \varepsilon/2\}} \geqslant \mathrm{Vol}(B(x_{\min}, \delta))e^{-\beta(\min_{x \in \mathcal{C}_1} V_x(x) + \varepsilon/2)},$$

$$q_{\{\varepsilon < \tilde{V}_x(x)\}} \leqslant (1 - \mathrm{Vol}(B(x_{\min}, \delta)))e^{-\beta(\min_{x \in \mathcal{C}_1} V_x(x) + \varepsilon)}.$$

Thus, the following condition is sufficient:

$$\frac{\mathrm{Vol}(B(x_{\min}, \delta))}{1 - \mathrm{Vol}(B(x_{\min}, \delta))}e^{\beta\varepsilon/2} \geqslant 2(K_L/\varepsilon - 1).$$

Hence, if we take

$$\beta \geqslant \frac{2}{\varepsilon} \log \left( 2 \frac{1 - \mathrm{Vol}(B(x_{\min}, \delta))}{\mathrm{Vol}(B(x_{\min}, \delta))} (K_L/\varepsilon - 1) \right) \tag{21}$$

then $(\hat{\mu}_x, \hat{\mu}_y)$ is an $\varepsilon$-Nash equilibrium. Since we have only bound the first two terms in the right hand side of (17) and the other two are bounded in the same manner, the statement of the theorem results from setting $\varepsilon = \varepsilon/2$ in (21).

## C.5  Proof of Theorem 6

First, we show that any pair $\hat{\mu}_x, \hat{\mu}_y$ such that

$$\frac{d\hat{\mu}_x}{dx}(x) = \frac{1}{Z_x} e^{-\beta \int \ell(x,y)\, d\hat{\mu}_y(y)}, \qquad \frac{d\hat{\mu}_y}{dy}(y) = \frac{1}{Z_y} e^{\beta \int \ell(x,y)\, d\hat{\mu}_x(x)}$$

is a stationary solution of (7). Denoting the Radon-Nikodym derivatives $\frac{d\hat{\mu}_x}{dx}, \frac{d\hat{\mu}_y}{dy}$ by $\hat{\rho}_x, \hat{\rho}_y$, it is sufficient to see that

$$\begin{cases} 0 = \nabla_x \cdot (\hat{\rho}_x \nabla_x V_x(\mu_y, x)) + \beta^{-1} \Delta_x \hat{\rho}_x \\ 0 = -\nabla_y \cdot (\hat{\rho}_y \nabla_y V_y(\mu_x, y)) + \beta^{-1} \Delta_y \hat{\rho}_y \end{cases} \tag{22}$$

holds weakly. And

$$\nabla_x \hat{\rho}_x = \frac{1}{Z_x} e^{-\beta \int \ell(x,y)\, d\hat{\mu}_y(y)} \left( -\beta \nabla_x \int \ell(x,y)\, d\hat{\mu}_y(y) \right) = -\hat{\rho}_x \nabla_x V_x(\hat{\mu}_y, x),$$

$$\nabla_y \hat{\rho}_y = \frac{1}{Z_y} e^{\beta \int \ell(x,y)\, d\hat{\mu}_x(x)} \left( \beta \nabla_y \int \ell(x,y)\, d\hat{\mu}_x(x) \right) = \hat{\rho}_y \nabla_y V_y(\hat{\mu}_x, y),$$

implies that (22) holds.

Now we will prove the converse. Suppose that $\hat{\mu}_x, \hat{\mu}_y$ are (weak) stationary solutions of (7). That is, if $\varphi_x \in C^2(\mathcal{X}), \varphi_y \in C^2(\mathcal{Y})$ are arbitrary twice continuously differentiable functions, the following holds

$$0 = \int_{\mathcal{X}} \left( -\int_{\mathcal{Y}} \nabla_x \varphi_x(x) \cdot \nabla_x \ell(x,y)\, d\hat{\mu}_y + \beta^{-1} \Delta_x \varphi_x(x) \right)\, d\hat{\mu}_x$$
$$0 = \int_{\mathcal{Y}} \left( \int_{\mathcal{X}} -\nabla_y \varphi_y(y) \cdot \nabla_y \ell(x,y)\, d\hat{\mu}_x - \beta^{-1} \Delta_y \varphi_y(x,y) \right)\, d\hat{\mu}_y \tag{23}$$

(23) can be seen as two measure-valued stationary Fokker-Planck equations. We want to see that they have densities and that the densities satisfy the corresponding classical stationary Fokker-Planck equations (22). Works in the theory of PDEs have studied sufficient conditions for measure-valued stationary Fokker-Planck equations to correspond to weak stationary Fokker-Planck equations, and further to classical stationary Fokker-Planck equations. See page 3 of Huang et al. (2015) for a more detailed explanation on the two steps. That measure-valued stationary correspond to weak stationary solutions is shown in Theorem 2.2 of Bogachev et al., 2001. That weak stationary solutions are classical stationary solutions requires that the drift term is in $C_{\mathrm{loc}}^{1,\alpha}$ (locally $\alpha$-Hölder continuous with exponent 1), meaning that it is in $C^1$ and that its derivatives are $\alpha$-Hölder in compact sets. The result follows from the theory of Schauder estimates. Differentiating under the integral sign, the drift terms $-\int_{\mathcal{Y}} \nabla_x \ell(x,y)\, d\hat{\mu}_y, \int_{\mathcal{X}} \nabla_y \ell(x,y)\, d\hat{\mu}_x$ fulfill the condition if $\ell \in C^{2,\alpha}$.

## D  Proof of Theorem 2

Recall the expression of an *Interacting Wasserstein-Fisher-Rao Gradient Flow (IWFRGF)* in (8):

$$\begin{cases} \partial_t \mu_x & = \gamma \nabla \cdot (\mu_x \nabla_x V_x(\mu_y, x)) \\ & \quad -\alpha \mu_x (V_x(\mu_y, x) - \mathcal{L}(\mu_x, \mu_y)), \quad \mu_x(0) = \mu_{x,0} \\ \partial_t \mu_y & = -\gamma \nabla \cdot (\mu_y \nabla_y V_y(\mu_x, y)) \\ & \quad +\alpha \mu_y (V_y(\mu_x, y) - \mathcal{L}(\mu_x, \mu_y)), \quad \mu_y(0) = \mu_{y,0} \end{cases}$$

The aim is to obtain a global convergence result like the one in Theorem 3.8 of Chizat (2019). First, we will rewrite Lemma 3.10 of Chizat (2019) in our case.

**Lemma 8.** *Let $\mu_x, \mu_y$ be the solution of the IWFRGF in* (8). *Let $\mu_x^\star, \mu_y^\star$ be arbitrary measures on $\mathcal{X}, \mathcal{Y}$. Let $\bar{\mu}_x(t) = \frac{1}{t}\int_0^t \mu_x(s)\,ds$ and $\bar{\mu}_y(t) = \frac{1}{t}\int_0^t \mu_y(s)\,ds$. Let $\|\cdot\|_{BL}$ be the bounded Lipschitz norm, i.e. $\|f\|_{BL} = \|f\|_\infty + Lip(f)$. Let*

$$\mathcal{Q}_{\mu^\star,\mu_0}(\tau) = \inf_{\mu \in \mathcal{P}(\Theta)} \|\mu^\star - \mu\|_{BL}^* + \frac{1}{\tau}\mathcal{H}(\mu,\mu_0) \tag{24}$$

*with $\Theta = \mathcal{X}$ or $\mathcal{Y}$. Let*

$$B = \frac{1}{2}\left(\max_{x \in \mathcal{X}, y \in \mathcal{Y}} \ell(x,y) - \min_{x \in \mathcal{X}, y \in \mathcal{Y}} \ell(x,y)\right) + Lip(\ell) \tag{25}$$

*Then,*

$$\mathcal{L}(\bar{\mu}_x(t), \mu_y^\star) - \mathcal{L}(\mu_x^\star, \bar{\mu}_y(t)) \leqslant B\mathcal{Q}_{\mu_x^\star,\mu_{x,0}}(\alpha Bt) + B\mathcal{Q}_{\mu_y^\star,\mu_{y,0}}(\alpha Bt) + \gamma B^2 t \tag{26}$$

*Proof.* The proof is as in Lemma 3.10 of Chizat (2019), but in this case we have to do everything twice. Namely, we define the dynamics

$$\frac{d\mu_x^\varepsilon}{dt} = \gamma\nabla \cdot (\mu_x^\varepsilon \nabla V_x(\mu_y, x))$$

$$\frac{d\mu_y^\varepsilon}{dt} = -\gamma\nabla \cdot (\mu_y^\varepsilon \nabla V_y(\mu_x, y))$$

initialized at $\mu_x^\varepsilon(0) = \mu_{x,0}^\varepsilon, \mu_y^\varepsilon(0) = \mu_{y,0}^\varepsilon$ arbitrary such that $\mu_{x,0}^\varepsilon$ and $\mu_{y,0}^\varepsilon$ are absolutely continuous with respect to $\mu_{x,0}$ and $\mu_{y,0}$ respectively.

Let us show that

$$\frac{1}{\alpha}\frac{d}{dt}\mathcal{H}(\mu_x^\varepsilon, \mu_x) = \int \frac{\delta\mathcal{L}}{\delta\mu_x}(\mu_x, \mu_y)(x)\,d(\mu_x^\varepsilon - \mu_x) \tag{27}$$

where $\mathcal{H}(\mu_x^\varepsilon, \mu_x)$ is the relative entropy, i.e.

$$\frac{d}{dt}\mathcal{H}(\mu_x^\varepsilon, \mu_x) = \frac{d}{dt}\int \log(\rho_x^\varepsilon)\,d\mu_x^\varepsilon,$$

$\rho_x^\varepsilon$ being the Radon-Nikodym derivative $d\mu_x^\varepsilon/d\mu_x$.

Assume to begin with that $\mu_x^\varepsilon$ remains absolutely continuous with respect to $\mu_x$ through time. We can write

$$\frac{d}{dt}\int \varphi_x(x)\rho_x^\varepsilon(x)d\mu_x(x) = \frac{d}{dt}\int \varphi(x)d\mu_x^\varepsilon(x)$$

We can develop the left hand side into

$$\frac{d}{dt}\int \varphi_x(x)\rho_x^\varepsilon(x)d\mu_x(x) = \int -\gamma\nabla(\varphi_x(x)\rho_x^\varepsilon(x)) \cdot \nabla V_x(\mu_y, x)d\mu_x(x)$$

$$+ \int -\alpha\varphi_x(x)\rho_x^\varepsilon(x)(V_x(\mu_y, x) - \mathcal{L}(\mu_x, \mu_y))d\mu_x(x)$$

$$+ \int \varphi_x(x)\frac{\partial\rho_x^\varepsilon}{\partial t}(x)d\mu_x(x)$$

$$= \int -\gamma(\nabla\varphi_x(x)\rho_x^\varepsilon(x) + \varphi_x(x)\nabla\rho_x^\varepsilon(x)) \cdot \nabla V_x(\mu_y, x)\,d\mu_x(x)$$

$$+ \int -\alpha\varphi_x(x)\rho_x^\varepsilon(x)(V_x(\mu_y, x) - \mathcal{L}(\mu_x, \mu_y))d\mu_x(x)$$

$$+ \int \varphi_x(x)\frac{\partial\rho_x^\varepsilon}{\partial t}(x)d\mu_x(x)$$

and the right hand side into

$$\frac{d}{dt}\int \varphi(x)d\mu_x^\varepsilon(x) = \int -\gamma\nabla\varphi_x(x) \cdot \nabla V_x(\mu_y, x)d\mu_x^\varepsilon(x)$$

Note that comparing terms, we obtain

$$
\int -\gamma \varphi_x(x) \nabla \rho_x^\varepsilon(x) \cdot \nabla V_x(\mu_y, x) \, d\mu_x(x)
$$

$$
= \int \alpha \varphi_x(x) \rho_x^\varepsilon(x) (V_x(\mu_y, x) - \mathcal{L}(\mu_x, \mu_y)) - \varphi_x(x) \frac{\partial \rho_x^\varepsilon}{\partial t}(x) \, d\mu_x(x)
$$

Since $\varphi_x$ is arbitrary, it must be that

$$
-\gamma \nabla \rho_x^\varepsilon(x) \cdot \nabla V_x(\mu_y, x) = \alpha \rho_x^\varepsilon(x)(V_x(\mu_y, x) - \mathcal{L}(\mu_x, \mu_y)) - \frac{\partial}{\partial t} \rho_x^\varepsilon(x) \tag{28}
$$

holds $\mu_x$-almost everywhere. Now,

$$
\frac{d}{dt} \int \log(\rho_x^\varepsilon) \, d\mu_x^\varepsilon = -\gamma \int \nabla(\log(\rho_x^\varepsilon(x))) \cdot \nabla V_x(\mu_y, x) \, d\mu_x^\varepsilon(x)
$$

$$
= -\gamma \int \frac{1}{\rho_x^\varepsilon(x)} \nabla(\rho_x^\varepsilon(x)) \cdot \nabla V_x(\mu_y, x) \, d\mu_x^\varepsilon(x)
$$

$$
= \alpha \int (V_x(\mu_y, x) - \mathcal{L}(\mu_x, \mu_y)) \, d\mu_x^\varepsilon(x) - \int \frac{1}{\rho_x^\varepsilon(x)} \frac{\partial}{\partial t} \rho_x^\varepsilon(x) d\mu_x^\varepsilon(x)
$$

Here,

$$
\int \frac{1}{\rho_x^\varepsilon(x)} \frac{\partial}{\partial t} \rho_x^\varepsilon(x) d\mu_x^\varepsilon(x) = \int \frac{\partial}{\partial t} \rho_x^\varepsilon(x) d\mu_x(x) = 0
$$

And since

$$
\mathcal{L}(\mu_x, \mu_y) = \int \frac{\delta \mathcal{L}}{\delta \mu_x}(\mu_x, \mu_y)(x) \, d\mu_x,
$$

the first term yields (27). We assumed that $\rho_x^\varepsilon$ existed and was regular enough. To make the argument precise, we can define the density of $\mu_x^\varepsilon$ with respect to $\mu_x$ to be a solution $\rho_x^\varepsilon$ of (28), and thus specify $\mu_x^\varepsilon$.

Now, recall that $\mu_x^\star$ is an arbitrary measure in $\mathcal{P}(\mathcal{X})$. By linearity of $\mathcal{L}$ with respect to $\mu_x$,

$$
\int \frac{\delta \mathcal{L}}{\delta \mu_x}(\mu_x, \mu_y)(x) \, d(\mu_x^\varepsilon - \mu_x) = \int \frac{\delta \mathcal{L}}{\delta \mu_x}(\mu_x, \mu_y)(x) \, d(\mu_x^\star - \mu_x) + \int \frac{\delta \mathcal{L}}{\delta \mu_x}(\mu_x, \mu_y)(x) \, d(\mu_x^\varepsilon - \mu_x^\star)
$$

$$
\leqslant -(\mathcal{L}(\mu_x, \mu_y) - \mathcal{L}(\mu_x^\star, \mu_y)) + \|\frac{\delta \mathcal{L}}{\delta \mu_x}(\mu_x, \mu_y)\|_{\mathrm{BL}} \|\mu_x^\varepsilon - \mu_x^\star\|_{\mathrm{BL}}^* \tag{29}
$$

Notice that we can take $\|\frac{\delta \mathcal{L}}{\delta \mu_x}(\mu_x, \mu_y)\|_{\mathrm{BL}}$ to be smaller than $B$ (defined in (25)). If we integrate (27) and (29) from 0 to $t$ and divide by $t$, we obtain

$$
\frac{1}{t} \int_0^t \mathcal{L}(\mu_x(s), \mu_y(s)) \, ds - \frac{1}{t} \int_0^t \mathcal{L}(\mu_x^\star, \mu_y(s)) \, ds
$$

$$
\leqslant \frac{1}{\alpha t} (\mathcal{H}(\mu_{x,0}^\varepsilon, \mu_{x,0}) - \mathcal{H}(\mu_x^\varepsilon(t), \mu_x(t))) + \frac{B}{t} \int_0^t \|\mu_x^\varepsilon - \mu_x^\star\|_{\mathrm{BL}}^* \, ds \tag{30}
$$

We bound the last term on the RHS:

$$
\frac{B}{t} \int_0^t \|\mu_x^\varepsilon - \mu_x^\star\|_{\mathrm{BL}}^* \, ds \leqslant B \|\mu_{x,0}^\varepsilon - \mu_x^\star\|_{\mathrm{BL}}^* + \frac{B}{t} \int_0^t \|\mu_{x,0}^\varepsilon - \mu_x^\varepsilon\|_{\mathrm{BL}}^* \, ds \tag{31}
$$

And

$$
\|\mu_x^\varepsilon(t) - \mu_{x,0}^\varepsilon\|_{\mathrm{BL}}^* = \sup_{\|f\|_{\mathrm{BL}} \leqslant 1, f \in C^2(\mathcal{X})} \int f \, d(\mu_x^\varepsilon(t) - \mu_{x,0}^\varepsilon) = \sup_{\|f\|_{\mathrm{BL}} \leqslant 1, f \in C^2(\mathcal{X})} \int_0^t \frac{d}{ds} \int f \, d\mu_x^\varepsilon(s) \, ds
$$

$$
= \sup_{\|f\|_{\mathrm{BL}} \leqslant 1, f \in C^2(\mathcal{X})} - \int_0^t \int \gamma \nabla f(x) \cdot \nabla \frac{\delta \mathcal{L}}{\delta \mu_x}(\mu_x^\varepsilon, \mu_y)(x) \, d\mu_x^\varepsilon(s) \, ds \leqslant \int_0^t \int \gamma B \, d\mu_x^\varepsilon(s) \, ds = \gamma B t
$$

$$
\tag{32}
$$

Also, by linearity of $\mathcal{L}$ with respect to $\mu_y$,

$$-\frac{1}{t}\int_0^t \mathcal{L}(\mu_x^\star, \mu_y(s))\, ds = -\mathcal{L}(\mu_x^\star, \bar\mu_y(t)) \tag{33}$$

If we use (31), (32) and (33) and the non-negativeness of the relative entropy on (30), we obtain:

$$\frac{1}{t}\int_0^t \mathcal{L}(\mu_x(s), \mu_y(s))\, ds - \mathcal{L}(\mu_x^\star, \bar\mu_y(t)) \leqslant \frac{\mathcal{H}(\mu_{x,0}^\varepsilon, \mu_{x,0})}{4\alpha t} + B\|\mu_{x,0}^\varepsilon - \mu_x^\star\|_{\mathrm{BL}}^* + \frac{B^2\gamma}{2}t \tag{34}$$

$$-\frac{1}{t}\int_0^t \mathcal{L}(\mu_x(s), \mu_y(s))\, ds + \mathcal{L}(\bar\mu_x(t), \mu_y^\star) \leqslant \frac{\mathcal{H}(\mu_{y,0}^\varepsilon, \mu_{y,0})}{4\alpha t} + B\|\mu_{y,0}^\varepsilon - \mu_y^\star\|_{\mathrm{BL}}^* + \frac{B^2\gamma}{2}t \tag{35}$$

Equation (35) is obtained by performing the same argument switching the roles of $x$ and $y$, and $\mathcal{L}$ by $-\mathcal{L}$. By adding equations (34) and (35) and considering the definition of $\mathcal{Q}$ in (24), we obtain the inequality (26).

$\square$

Notice that by taking the supremum wrt $\mu_x^\star, \mu_y^\star$ on (26) we obtain a bound on the Nikaido-Isoda error of $(\bar\mu_x(t), \bar\mu_y(t))$ (see (2)).

Next, we will obtain a result like Lemma E.1 from Chizat (2019) in which we bound $\mathcal{Q}$. The proof is a variation of the argument in Lemma E.1 from Chizat (2019), as in our case no measures are necessarily sparse.

**Lemma 9.** *Let $\Theta$ be a Riemannian manifold of dimension $d$. Assume that $Vol(B_{\theta,\varepsilon}) \geqslant e^{-K}\varepsilon^d$ for all $\theta \in \Theta$, where the volume is defined of course in terms of the Borel measure[1] of $\Theta$. If $\rho := \frac{d\mu_0}{d\theta}$ is the Radon-Nikodym derivative of $\mu_0$ with respect to the Borel measure of $\Theta$, assume that $\rho(\theta) \geqslant e^{-K'}$ for all $\theta \in \Theta$. The function $\mathcal{Q}_{\mu^\star,\mu_0}(\tau)$ defined in (24) can be bounded by*

$$\mathcal{Q}_{\mu^\star,\mu_0}(\tau) \leqslant \frac{d}{\tau}(1 - \log d + \log \tau) + \frac{1}{\tau}(K + K')$$

*Proof.* We will choose $\mu^\varepsilon$ in order to bound the infimum. For $\theta \in \Theta, \varepsilon > 0$, let $\xi_{\theta,\varepsilon}$ be a probability measure on $\Theta$ with support on the ball $B_{\theta,\varepsilon}$ of radius $\varepsilon$ centered at $\theta$ and proportional to the Borel measure for all subsets of the ball. Let us define the measure

$$\mu^\varepsilon(A) = \int_\Theta \xi_{\theta,\varepsilon}(A)\, d\mu^\star(\theta)$$

for all Borel sets $A$ of $\mathcal{X}$. Now, we can bound $\|\mu^\varepsilon - \mu^\star\|_{\mathrm{BL}}^* \leqslant W_1(\mu^\varepsilon, \mu^\star)$. Let us consider the coupling $\gamma$ between $\mu^\varepsilon$ and $\mu^\star$ defined as:

$$\gamma(A \times B) = \int_A \xi_{\theta,\varepsilon}(B)\, d\mu^\star(\theta)$$

for $A, B$ arbitrary Borel sets of $\Theta$. Notice that $\gamma$ is indeed a coupling between $\mu^\varepsilon$ and $\mu^\star$, because $\gamma(A \times \Theta) = \mu^\star(A)$ and $\gamma(\Theta \times B) = \mu^\varepsilon(B)$. Hence,

$$W_1(\mu^\varepsilon, \mu^\star) \leqslant \int_{\Theta \times \Theta} d_\Theta(\theta, \theta')\, d\gamma(\theta, \theta') = \int_\Theta \frac{1}{\mathrm{Vol}(B_{\theta',\varepsilon})}\int_{B_{\theta',\varepsilon}} d_\Theta(\theta, \theta')\, d\theta\, d\mu^\star(\theta') \tag{36}$$

where the inner integral is with respect to the Borel measure on $\Theta$. Since $d_\Theta(\theta, \theta') \leqslant \varepsilon$ for all $\theta \in B_{\theta',\varepsilon}$, we conclude from that (36) that $W_1(\mu^\varepsilon, \mu^\star) \leqslant \varepsilon$.

Next, let us bound the relative entropy term. Define $\rho_\varepsilon$ as the Radon-Nikodym derivative of $\mu^\varepsilon$ with respect to the Borel measure of $\Theta$, i.e.

$$\rho_\varepsilon(\theta) := \frac{d\mu^\varepsilon}{d\theta}(\theta) = \int_\Theta \frac{1}{\mathrm{Vol}(B_{\theta',\varepsilon})}\mathbb{1}_{B_{\theta',\varepsilon}}(\theta)\, d\mu^\star(\theta').$$

Also, recall that $\rho := \frac{d\mu_0}{d\theta}$. Then, we write

$$\mathcal{H}(\mu^\varepsilon, \mu_0) = \int_\Theta \log \frac{\rho_\varepsilon}{\rho} d\mu^\varepsilon = \int_\Theta \log(\rho_\varepsilon)\rho_\varepsilon d\theta - \int_\Theta \log(\rho)\rho_\varepsilon d\theta. \tag{37}$$

On the one hand, we use the convexity of the function $x \to x \log x$:

$$\rho_\varepsilon(\theta) \log \rho_\varepsilon(\theta) = \left(\int_\Theta \frac{1}{\text{Vol}(B_{\theta',\varepsilon})} \mathbb{1}_{B_{\theta',\varepsilon}}(\theta)\, d\mu^\star(\theta')\right) \log \left(\int_\Theta \frac{1}{\text{Vol}(B_{\theta',\varepsilon})} \mathbb{1}_{B_{\theta',\varepsilon}}(\theta)\, d\mu^\star(\theta')\right)$$

$$\leqslant \int_\Theta \left(\frac{1}{\text{Vol}(B_{\theta',\varepsilon})} \mathbb{1}_{B_{\theta',\varepsilon}}(\theta)\right) \log \left(\frac{1}{\text{Vol}(B_{\theta',\varepsilon})} \mathbb{1}_{B_{\theta',\varepsilon}}(\theta)\right) d\mu^\star(\theta').$$

We use Fubini's theorem:

$$\int_\Theta \rho_\varepsilon(\theta) \log \rho_\varepsilon(\theta)\, d\theta \leqslant \int_\Theta \int_\Theta \left(\frac{1}{\text{Vol}(B_{\theta',\varepsilon})} \mathbb{1}_{B_{\theta',\varepsilon}}(\theta)\right) \log \left(\frac{1}{\text{Vol}(B_{\theta',\varepsilon})} \mathbb{1}_{B_{\theta',\varepsilon}}(\theta)\right)\, d\theta\, d\mu^\star(\theta')$$

$$= \int_\Theta \frac{1}{\text{Vol}(B_{\theta',\varepsilon})} \int_{B_{\theta',\varepsilon}} -\log\left(\text{Vol}(B_{\theta',\varepsilon})\right)\, d\theta\, d\mu^\star(\theta') = -\int_\Theta \log\left(\text{Vol}(B_{\theta',\varepsilon})\right) d\mu^\star(\theta')$$

$$\leqslant -d\log\varepsilon + K$$
$$\tag{38}$$

where $d$ is the dimension of $\Theta$ and $K$ is a constant such that $\text{Vol}(B_{\theta',\varepsilon}) \geqslant e^{-K}\varepsilon^d$ for all $\theta' \in \Theta$.

On the other hand,

$$-\int_\Theta \log(\rho(\theta))\rho_\varepsilon(\theta)\, d\theta = \int_\Theta \frac{1}{\text{Vol}(B_{\theta',\varepsilon})} \int_{\text{Vol}(B_{\theta',\varepsilon})} -\log(\rho(\theta))\, d\theta\, d\mu^\star(\theta')$$

$$\leqslant \int_\Theta \frac{1}{\text{Vol}(B_{\theta',\varepsilon})} \int_{\text{Vol}(B_{\theta',\varepsilon})} K'\, d\theta\, d\mu^\star(\theta') = K'$$
$$\tag{39}$$

where $K'$ is defined such that $\rho(\theta) \geqslant e^{-K'}$ for all $\theta \in \Theta$.

By plugging (38) and (39) into (37) we obtain:

$$\|\mu^\star - \mu^\varepsilon\|^*_{\text{BL}} + \frac{1}{\tau}\mathcal{H}(\mu^\varepsilon, \mu_0) \leqslant \varepsilon + \frac{1}{\tau}(-d\log\varepsilon + K + K').$$

If we optimize the bound with respect to $\varepsilon$ we obtain the final result. $\qquad\square$

**Theorem 2.** *Let $\varepsilon > 0$ arbitrary. Suppose that $\mu_{x,0}, \mu_{y,0}$ are such that their Radon-Nikodym derivatives with respect to the Borel measures of $\mathcal{X}, \mathcal{Y}$ are lower-bounded by $e^{-K'_x}, e^{-K'_y}$ respectively. For any $\delta \in (0, 1/2)$, there exists a constant $C_{\delta, \mathcal{X}, \mathcal{Y}, K'_x, K'_y} > 0$ depending on the dimensions of $\mathcal{X}, \mathcal{Y}$, their curvatures and $K'_x, K'_y$, such that if $\gamma/\alpha < 1$ and*

$$\frac{\gamma}{\alpha} \leqslant \left(\frac{\varepsilon}{C_{\delta, \mathcal{X}, \mathcal{Y}, K'_x, K'_y}}\right)^{\frac{2}{1-\delta}} \tag{40}$$

*Then, at $t_0 = (\alpha\gamma)^{-1/2}$ we have*

$$NI(\bar\mu_x(t_0), \bar\mu_y(t_0)) := \sup_{\mu^\star_x, \mu^\star_y} \mathcal{L}(\bar\mu_x(t_0), \mu^\star_y) - \mathcal{L}(\mu^\star_x, \bar\mu_y(t_0)) \leqslant \varepsilon$$

*Proof.* We plug the bound of Theorem 9 into the result of Theorem 8, obtaining

$$\mathcal{L}(\bar\mu_x(t), \mu^\star_y) - \mathcal{L}(\mu^\star_x, \bar\mu_y(t)) \leqslant \frac{d_x}{\alpha t}(1 - \log d_x + \log(\alpha Bt))$$

$$+ \frac{d_y}{\alpha t}(1 - \log d_y + \log(\alpha Bt))$$

$$+ \frac{1}{\alpha t}(K_x + K'_x + K_y + K'_y) + \gamma B^2 t$$

Now, we set $t = (\alpha\gamma)^{-1/2}$, and thus the right hand side becomes

$$\sqrt{\frac{\gamma}{\alpha}}\left(d_x\left(1 - \log\frac{d_x}{B} + \log\sqrt{\frac{\alpha}{\gamma}}\right) + d_y\left(1 - \log\frac{d_y}{B} + \log\sqrt{\frac{\alpha}{\gamma}}\right) + K_x + K'_x + K_y + K'_y + B^2\right)$$

(41)

Let $\varepsilon > 0$ arbitrary. We want (41) to be lower or equal than $\varepsilon$. For any $\delta$ such that $0 < \delta < 1/2$, there exists $C_\delta$ such that $\log(x) \leqslant C_\delta x^\delta$. This yields

$$\sqrt{\frac{\gamma}{\alpha}}\left(d_x\left(1 - \log\frac{d_x}{B} + C_\delta\left(\frac{\alpha}{\gamma}\right)^{-\delta/2}\right) + d_y\left(1 - \log\frac{d_y}{B} + C_\delta\left(\frac{\alpha}{\gamma}\right)^{-\delta/2}\right)\right)$$
$$+ \sqrt{\frac{\gamma}{\alpha}}\left(K_x + K'_x + K_y + K'_y + B^2\right)$$

(42)

If we set $\gamma < \alpha$, $(\gamma/\alpha)^{-\delta/2} > 1$ then (42) is upper-bounded by

$$\left(\frac{\gamma}{\alpha}\right)^{\frac{1-\delta}{2}}\left(d_x(1 - \log\frac{d_x}{B} + C_\delta) + d_y(1 - \log\frac{d_y}{B} + C_\delta) + K_x + K'_x + K_y + K'_y + B^2\right)$$

If we bound this by $\varepsilon$, we obtain the bound in (40).  $\square$

**Corollary 1.** *Let $(\mathcal{X}_{d_x}, \mathcal{Y}_{d_y}, l_{d_x,d_y})_{d_x\in\mathbb{N}, d_y\in\mathbb{N}}$ be a family indexed by $\mathbb{N}^2$. Assume that $\mu_{x,0}, \mu_{y,0}$ are set to be the Borel measures in $\mathcal{X}_{d_x}, \mathcal{Y}_{d_y}$, that $\mathcal{X}_{d_x}, \mathcal{Y}_{d_y}$ are locally isometric to the $d_x, d_y$-dimensional Euclidean spaces, and that the volumes of $\mathcal{X}_{d_x}, \mathcal{Y}_{d_y}$ grow no faster than exponentially on the dimensions $d_x, d_y$. Assume that $l_{d_x,d_y}$ are such that $B$ is constant. Then, we can rewrite (40) as*

$$\frac{\gamma}{\alpha} \leqslant O\left(\left(\frac{\varepsilon}{(d_x + d_y)\log(B) + d_x\log(d_x) + d_y\log(d_y) + B^2}\right)^{\frac{2}{1-\delta}}\right)$$

*Proof.* The volume of $n$-dimensional ball of radius $r$ in $n$-dimensional Euclidean space is

$$V_n(r) = \frac{\pi^{n/2}}{\Gamma(\frac{n}{2} + 1)}R^n,$$

and hence, if $\mathcal{X}, \mathcal{Y}$ are locally isometric to the $d_x$ and $d_y$-dimensional Euclidean spaces we can take

$$K_x = \log\Gamma\left(\frac{d_x}{2} + 1\right) - \frac{d_x}{2}\log(\pi) \leqslant \left(\frac{d_x}{2} + 1\right)\log\left(\frac{d_x}{2} + 1\right) - \frac{d_x}{2}\log(\pi) \leqslant O(d_x\log d_x)$$

$$K_y = \log\Gamma(\frac{d_y}{2} + 1) - \frac{n}{2}\log(\pi) \leqslant O(d_x\log d_x)$$

If the volumes of $\mathcal{X}, \mathcal{Y}$ grow no faster than an exponential of the dimensions $d_x, d_y$ and we take $\mu_{x,0}, \mu_{y,0}$ to be the Borel measures, we can take $K'_x = \log(\mathrm{Vol}(\mathcal{X})), K'_y = \log(\mathrm{Vol}(\mathcal{Y}))$ to be constant with respect to the dimensions $d_x, d_y$.  $\square$

# E  Proof of Theorem 3(i)

## E.1  Preliminaries

Throughout the section we will use the techniques shown in §G.5 to deal with SDEs on manifolds. Effectively, this means that for SDEs we have additional drift terms $\hat{\mathbf{h}}_x$ or $\hat{\mathbf{h}}_x$ induced by the geometry of the manifold, and that we must project the variations of the Brownian motion onto the tangent space.

Define the processes $\mathbf{X}^n = (X^1, \ldots, X^n)$ and $\mathbf{Y}^n = (Y^1, \ldots, Y^n)$ such that for all $i \in \{1, \ldots, n\}$,

$$dX_t^i = \left( -\frac{1}{n} \sum_{j=1}^n \nabla_x \ell(X_t^i, Y_t^j) + \hat{\mathbf{h}}_x(X_t^i) \right) dt + \sqrt{2\beta^{-1}} \, \mathrm{Proj}_{T_{X_t^i} \mathcal{X}}(dW_t^i), \quad X_0^{n,i} = \xi^i \sim \mu_{x,0}$$

$$dY_t^i = \left( \frac{1}{n} \sum_{j=1}^n \nabla_y \ell(X_t^j, Y_t^i) + \hat{\mathbf{h}}_y(Y_t^i) \right) dt + \sqrt{2\beta^{-1}} \, \mathrm{Proj}_{T_{Y_t^i} \mathcal{Y}}(d\bar{W}_t^i), \quad Y_0^{n,i} = \bar{\xi}^i \sim \mu_{y,0}$$

(43)

where $\mathbf{W}_t = (W_t^1, \ldots, W_t^n)$, and $\bar{\mathbf{W}}_t = (\bar{W}_t^1, \ldots, \bar{W}_t^n)$ are Brownian motions on $\mathbb{R}^{nD_x}$ and $\mathbb{R}^{nD_y}$ respectively. Notice that $\mathbf{X}_t$ is valued in $\mathcal{X}^n \subseteq \mathbb{R}^{nD_x}$ and $\mathbf{Y}_t$ is valued in $\mathcal{Y}^n \subseteq \mathbb{R}^{nD_y}$. (43) can be seen as a system of $2n$ interacting particles in which each particle of one player interacts with all the particles of the other one. It also corresponds to noisy continuous-time mirror descent on parameter spaces for an augmented game in which there are $n$ replicas of each player, choosing $\frac{1}{2} \| \cdot \|_2^2$ for the mirror map.

Now, define $\tilde{\mathbf{X}} = (\tilde{X}^1, \ldots, \tilde{X}^n)$ and $\tilde{\mathbf{Y}} = (\tilde{Y}^1, \ldots, \tilde{Y}^n)$ for all $i \in \{1, \ldots, n\}$ let

$$d\tilde{X}_t^i = \left( -\int_{\mathcal{Y}} \nabla_x \ell(\tilde{X}_t^i, y) \, d\mu_{y,t} + \hat{\mathbf{h}}_x(\tilde{X}_t^i) \right) dt + \sqrt{2\beta^{-1}} \, \mathrm{Proj}_{T_{\tilde{X}_t^i} \mathcal{X}}(dW_t^i),$$

$$d\tilde{Y}_t^i = \left( \int_{\mathcal{X}} \nabla_y \ell(x, \tilde{Y}_t^i) \, d\mu_{x,t} + \hat{\mathbf{h}}_y(\tilde{Y}_t^i) \right) dt + \sqrt{2\beta^{-1}} \, \mathrm{Proj}_{T_{\tilde{Y}_t^i} \mathcal{Y}}(d\bar{W}_t^i),$$

(44)

$$\tilde{X}_0^i = \xi^i \sim \mu_{x,0}, \quad \mu_{y,t} = \mathrm{Law}(\tilde{Y}_t^i), \quad \tilde{Y}_0^i = \bar{\xi}^i \sim \mu_{y,0}, \quad \mu_{x,t} = \mathrm{Law}(\tilde{X}_t^i)$$

**Lemma 10** (Forward Kolmogorov equation). *The laws $(\mu_x)_{t \in [0,T]}, (\mu_y)_{t \in [0,T]}$ of a solution $\tilde{X}, \tilde{Y}$ of (44) with $n = 1$ (seen as elements of $\mathcal{C}([0,T], \mathcal{P}(\mathcal{X})), \mathcal{C}([0,T], \mathcal{P}(\mathcal{Y})))$ are a solution of (45).*

$$\begin{cases} \partial_t \mu_x = \nabla_x \cdot (\mu_x \nabla_x V_x(\mu_y, x)) + \beta^{-1} \Delta_x \mu_x, & \mu_x(0) = \mu_{x,0} \\ \partial_t \mu_y = -\nabla_y \cdot (\mu_y \nabla_y V_y(\mu_x, y)) + \beta^{-1} \Delta_y \mu_y, & \mu_y(0) = \mu_{y,0} \end{cases}$$

(45)

*Proof.* We sketch the derivation for the forward Kolmogorov equation on manifolds. First, we define the semigroups

$$P_t^x \varphi_x(x) = \mathbb{E}[\varphi_x(\tilde{X}_t) | \tilde{X}_0 = x], \quad P_t^y \varphi_y(y) = \mathbb{E}[\varphi_y(\tilde{Y}_t) | \tilde{Y}_0 = y],$$

where $\tilde{X}, \tilde{Y}$ are solutions of (44) with $n = 1$. We obtain that if $\mathcal{L}_t^x, \mathcal{L}_t^y$ are the infinitesimal generators (i.e., $\mathcal{L}_t^x \varphi_x(x) = \lim_{t \to 0^+} \frac{1}{t} (P_t^x \varphi_x(x) - \varphi_x(x))$), the backward Kolmogorov equations $\frac{d}{dt} P_t^x \varphi_x(x) = \mathcal{L}_t^x P_t^x \varphi_x(x), \frac{d}{dt} P_t^y \varphi_y(y) = \mathcal{L}_t^y P_t^y \varphi_y(y)$ hold for $\varphi_x, \varphi_y$ in the domains of the generators. Since $\mathcal{L}_t^x$ and $P_t^x$ commute for these choices of $\varphi_x$, we have $\frac{d}{dt} P_t^x \varphi_x(x) = P_t^x \mathcal{L}_t^x \varphi_x(x), \frac{d}{dt} P_t^y \varphi_y(y) = P_t^y \mathcal{L}_t^y \varphi_y(y)$. By integrating these two equations over the initial measures $\mu_{x,0}, \mu_{y,0}$, we get

$$\frac{d}{dt} \int \varphi_x(x) \, d\mu_{x,t} = \int \mathcal{L}_t^x \varphi_x(x) \, d\mu_{x,t}, \quad \frac{d}{dt} \int \varphi_y(y) \, d\mu_{y,t} = \int \mathcal{L}_t^y \varphi_y(y) \, d\mu_{y,t}.$$

We can write an explicit form for $\mathcal{L}_t^x P_t^x \varphi_x(x)$ by using Itô's lemma on (44):

$$\mathcal{L}_t^x \varphi_x(x) = \left( \int_{\mathcal{Y}} \nabla_x \ell(x, y) \, d\mu_{y,s} \, ds - \hat{\mathbf{h}}_x(x) \right) \nabla_x \varphi_x(x) + \beta^{-1} \mathrm{Tr} \left( \left( \mathrm{Proj}_{T_x \mathcal{X}} \right)^\top H \varphi_x(x) \, \mathrm{Proj}_{T_x \mathcal{X}} \right),$$

where we use $\mathrm{Proj}_{T_{\tilde{X}_t^i} \mathcal{X}}$ to denote its matrix in the canonical basis.

Let $\{\xi_k\}$ be a partition of unity for $\mathcal{X}$ (i.e. a set of functions such that $\sum_k \xi_k(x) = 1$) in which each $\xi_k$ is regular enough and supported on a patch of $\mathcal{X}$. We can write

$$\frac{d}{dt} \int_{\mathcal{X}} \varphi_x(x) \, d\mu_{x,t}(x) = \frac{d}{dt} \int_{\mathcal{X}} \varphi_x(x) \, d\mu_{x,t}(x) = \sum_k \frac{d}{dt} \int_{\mathcal{X}} \xi_k(x) \varphi_x(x) \, d\mu_{x,t}(x)$$

$$= \sum_k \int \mathcal{L}_t^x (\xi_k(x) \varphi_x(x)) \, d\mu_{x,t}$$

Now, let $\tilde{\varphi}_x^k(x) = \xi_k(x)\varphi_x(x)$.

$$\int_{\mathcal{X}} \mathcal{L}_t^x \tilde{\varphi}_x^k(x)\, d\mu_{x,t}$$
$$= \int_{\mathcal{X}} \left( \nabla_x V_x(\mu_{y,s}, x) - \hat{\mathbf{h}}_x(x) \right) \nabla_x \tilde{\varphi}_x^k(x) + \beta^{-1} \mathrm{Tr}\left( \left( \mathrm{Proj}_{T_x \mathcal{X}} \right)^\top H \tilde{\varphi}_x^k(x)\, \mathrm{Proj}_{T_x \mathcal{X}} \right)\, d\mu_{x,t}$$

Notice that this equation is analogous to (66). We reverse the argument made in §G.5. Using the fact that the support of $\tilde{\varphi}_x^k(x)$ is contained on some patch of $\mathcal{X}$ given by the mapping $\psi_k : U_{\mathbb{R}^d} \subseteq \mathbb{R}^d \to U \subseteq \mathcal{X} \subseteq \mathbb{R}^D$, the corresponding Fokker-Planck on $U_{\mathbb{R}^d}$ is

$$\frac{d}{dt} \int_{U_{\mathbb{R}^d}} \tilde{\varphi}_x^k(\psi_k(q))\, d(\psi_k^{-1})_* \mu_{x,t}(q)$$
$$= \int_{U_{\mathbb{R}^d}} \nabla V_x(\mu_{y,s}, \psi_k(q)) \cdot \nabla \tilde{\varphi}_x^k(\psi_k(q)) + \beta^{-1} \Delta \tilde{\varphi}_x^k(\psi_k(q))\, d(\psi_k^{-1})_* \mu_{x,t}(q),$$

where the gradients and the Laplacian are in the metric inherited from the embedding (as in §G.5). The pushforward definition implies

$$\frac{d}{dt} \int_{\mathcal{X}} \tilde{\varphi}_x^k(x)\, d\mu_{x,t}(x) = \int_{U_{\mathbb{R}^d}} \nabla V_x(\mu_{y,s}, x) \cdot \nabla \tilde{\varphi}_x^k(x) + \beta^{-1} \Delta \tilde{\varphi}_x^k(x)\, d\mu_{x,t}(x),$$

By substituting $\tilde{\varphi}_x^k(x) = \xi_k(x)\varphi_x(x)$, summing for all $k$ and using $\sum_k \xi_k(x) = 1$, we obtain:

$$\frac{d}{dt} \int_{\mathcal{X}} \varphi_x(x)\, d\mu_{x,t}(x) = \int_{\mathcal{X}} \nabla_x V_x(\mu_{y,s}, x) \cdot \nabla_x \varphi_x(x) + \beta^{-1} \Delta_x \varphi_x(x)\, d\mu_{x,t}(x)$$

which is the same as the first equation in (7). The second equation is obtained analogously. $\qquad\square$

Let $\mu_x^n = \frac{1}{n} \sum_{i=1}^n \delta_{X^i}$ be a $\mathcal{P}(\mathcal{C}([0,T], \mathcal{X}))$-valued random element that corresponds to the empirical measure of a solution $\mathbf{X}^n$ of (43). Analogously, let $\mu_y^n = \frac{1}{n} \sum_{i=1}^n \delta_{Y^i}$ be a $\mathcal{P}(\mathcal{C}([0,T], \mathcal{Y}))$-valued random element corresponding to the empirical measure of $\mathbf{Y}^n$.

Define the 2-Wasserstein distance on $\mathcal{P}(\mathcal{C}([0,T], \mathcal{X}))$ as

$$\mathcal{W}_2^2(\mu, \nu) := \inf_{\pi \in \Pi(\mu,\nu)} \int_{C([0,T],\mathcal{X})^2} d(x,y)^2\, d\pi(x,y) \tag{46}$$

where $d(x,y) = \sup_{t \in [0,T]} d_{\mathcal{X}}(x(t), y(t))$. Define it analogously on $\mathcal{P}(\mathcal{C}([0,T], \mathcal{Y}))$.

We state a stronger version of the law of large numbers in the first statement of Theorem 3(i).

**Theorem 9.** *There exists a solution of the coupled McKean-Vlasov SDEs (44). Pathwise uniqueness and uniqueness in law hold. Let $\mu_x \in \mathcal{P}(\mathcal{C}([0,T], \mathcal{X})), \mu_y \in \mathcal{P}(\mathcal{C}([0,T], \mathcal{Y}))$ be the unique laws of the solutions for $n = 1$ (all pairs have the same solutions). Then,*

$$\mathbb{E}[\mathcal{W}_2^2(\mu_x^n, \mu_x) + \mathcal{W}_2^2(\mu_y^n, \mu_y)] \xrightarrow{n \to \infty} 0$$

Let us comment on why Theorem 9 implies the first statement in Theorem 3(i). We make use of the mapping $\mathcal{P}(\mathcal{C}([0,T], \mathcal{X})) \ni \mu \mapsto (\mu_t)_{t \in [0,T]} \in \mathcal{C}([0,T], \mathcal{P}(\mathcal{X}))$ into the time marginals. By the definition (46), $\sup_{t \in [0,t]} \mathcal{W}_2^2(\mu_{x,t}^n, \mu_{x,t}) \leqslant \mathcal{W}_2^2(\mu_x^n, \mu_x)$ and the same holds for $\mu_y^n, \mu_y$. At this point, Lemma 10 states that $(\mu_x)_{t \in [0,T]}, (\mu_y)_{t \in [0,T]}$ is a solution of the mean-field ERIWGF (45) and concludes the argument. The proof of Theorem 9 uses a propagation of chaos argument, originally due to Sznitman (1991) in the context of interacting particle systems. Our argument follows Theorem 3.3 of Lacker (2018).

## E.2  Existence and uniqueness

We prove existence and uniqueness of the system given by

$$\tilde{X}_t = \int_0^t \left( -\int_{\mathcal{Y}} \nabla_x \ell(\tilde{X}_s, y)\, d\mu_{y,s}\, ds + \hat{\mathbf{h}}_x(\tilde{X}_s) \right) ds + \sqrt{2\beta^{-1}} \int_0^t \mathrm{Proj}_{T_{\tilde{X}_s} \mathcal{X}}(dW_s),$$
$$\tilde{Y}_t = \int_0^t \left( \int_{\mathcal{X}} \nabla_y \ell(x, \tilde{Y}_s)\, d\mu_{x,s} + \hat{\mathbf{h}}_y(Y_s^{n,i}) \right) ds + \sqrt{2\beta^{-1}} \int_0^t \mathrm{Proj}_{T_{\tilde{Y}_s} \mathcal{Y}}(d\bar{W}_s), \tag{47}$$
$$\mu_{x,t} = \mathrm{Law}(\tilde{X}_t^n), \quad \mu_{y,t} = \mathrm{Law}(\tilde{Y}_t^n), \quad \tilde{X}_0 = \xi \sim \mu_{x,0}, \quad \tilde{Y}_0 = \bar{\xi} \sim \mu_{y,0}.$$

Path-wise uniqueness means that given $W, \bar{W}, \xi, \bar{\xi}$, two solutions are equal almost surely. Uniqueness in law means that regardless of the Brownian motion and the initialization random variables chosen (as long as they are $\mu_{x,0}$ and $\mu_{y,0}$-distributed), the law of the solution is unique. We prove that both hold for (47).

We have that for all $x, x' \in \mathcal{X}, \mu, \nu \in \mathcal{P}(\mathcal{Y})$,

$$\left| \int \nabla_x \ell(x, y) \, d\mu - \int \nabla_x \ell(x', y) \, d\nu \right| \leqslant L(d(x, x') + \mathcal{W}_2(\mu, \nu)) \tag{48}$$

This is obtained by adding and subtracting the term $\int \nabla_x \ell(x'y) \, d\mu$, by using the triangle inequality and the inequality $\mathcal{W}_1(\mu, \nu)) \leqslant \mathcal{W}_2(\mu, \nu))$ (which is proven using the Cauchy-Schwarz inequality). Hence,

$$\left| \int \nabla_x \ell(x, y) \, d\mu - \int \nabla_x \ell(x', y) \, d\nu \right|^2 \leqslant 2L^2(d(x, x')^2 + \mathcal{W}_2^2(\mu, \nu)) \tag{49}$$

On the other hand, using the regularity of the manifold, there exists $\mathcal{L}_\mathcal{X}$ such that

$$|\hat{\mathbf{h}}_x(x) - \hat{\mathbf{h}}_x(x')| \leqslant L_\mathcal{X} d(x, x'),$$
$$|\mathrm{Proj}_{T_x \mathcal{X}} - \mathrm{Proj}_{T_{x'} \mathcal{X}}| \leqslant L_\mathcal{X} d(x, x')$$

where $\mathrm{Proj}_{T_x \mathcal{X}}$ denotes its matrix in the canonical basis and the norm in the second line is the Frobenius norm. Also, let $\|x - x'\|$ be the Euclidean norm of $\mathcal{X}$ in $\mathbb{R}^{D_x}$ (the Euclidean space where $\mathcal{X}$ is embedded) and let $K_\mathcal{X} > 1$ be such that $d(x, x') \leqslant K_\mathcal{X} \|x - x'\|$.

Let $\mu_y, \nu_y \in \mathcal{P}(\mathcal{C}([0, T], \mathcal{X}))$ and let $X^{\mu_y}, X^{\nu_y}$ be the solutions of the first equation of (47) when we plug $\mu_y$ ($\nu_y$ resp.) as the measure for the other player. $X^{\mu_y}$ and $X^{\nu_y}$ exist and are unique by the classical theory of SDEs (see Chapter 18 of Kallenberg (2002)). Following the procedure in Theorem 3.3 of Lacker (2018), we obtain

$$
\begin{aligned}
\mathbb{E}[\|X^{\mu_y} - X^{\nu_y}\|_t^2] &\leqslant 3t\mathbb{E}\left[ \int_0^t \left| \int \nabla_x \ell(X^{\mu_y}, y) \, d\mu_{y,r} - \int \nabla_x \ell(X^{\nu_y}, y) \, d\nu_{y,r} \right|^2 dr \right] \\
&+ 3t\mathbb{E}\left[ \int_0^t |\hat{\mathbf{h}}_x(X^{\mu_y}) - \hat{\mathbf{h}}_x(X^{\nu_y})|^2 \, dr \right] \\
&+ 12\mathbb{E}\left[ \int_0^t |\mathrm{Proj}_{T_x \mathcal{X}} - \mathrm{Proj}_{T_{x'} \mathcal{X}}|^2 \, dr \right] \\
&\leqslant 3(3t + 4)\tilde{L}^2 \mathbb{E}\left[ \int_0^t (\|X^{\mu_y} - X^{\nu_y}\|_r^2 + \mathcal{W}_2^2(\mu_{y,r}, \nu_{y,r})) \, dr \right],
\end{aligned}
\tag{50}
$$

where $\tilde{L}^2 = (L^2 + L_\mathcal{X}^2)K_\mathcal{X}^2$. Using Fubini's theorem and Gronwall's inequality, we obtain

$$\mathbb{E}[\|X^{\mu_y} - X^{\nu_y}\|_t^2] \leqslant 3(3T + 4)\tilde{L}^2 \exp(3(3T + 4)\tilde{L}^2) \int_0^t \mathcal{W}_2^2(\mu_{y,r}, \nu_{y,r})) \, dr \tag{51}$$

Let $C_T := 3(3T + 4)\tilde{L}^2 \exp(3(3T + 4)\tilde{L}^2)$. For $\mu, \nu \in \mathcal{P}(C([0, T], \mathcal{X}))$, define

$$\mathcal{W}_{2,t}^2(\mu, \nu) := \inf_{\pi \in \Pi(\mu, \nu)} \int_{C([0,T],\mathcal{X})^2} \sup_{r \in [0,t]} d(x(r), y(r)) \, \pi(dx, dy)$$

Hence, (51) and the bound $\mathcal{W}_2^2(\mu_{y,r}, \nu_{y,r}) \leqslant \mathcal{W}_{2,r}^2(\mu_y, \nu_y)$ yield

$$\mathbb{E}[\|X^{\mu_y} - X^{\nu_y}\|_t^2] \leqslant C_T \int_0^t \mathcal{W}_{2,r}^2(\mu_y, \nu_y) \, dr$$

Reasoning analogously for the other player, we obtain

$$\mathbb{E}[\|X^{\mu_y} - X^{\nu_y}\|_t^2 + \|Y^{\mu_x} - Y^{\nu_x}\|_t^2] \leqslant C_T \int_0^t \mathcal{W}_{2,r}^2(\mu_y, \nu_y) \, dr + C_T \int_0^t \mathcal{W}_{2,r}^2(\mu_x, \nu_x) \, dr$$

Given $\mu_y \in \mathcal{P}(C([0,T], \mathcal{Y}))$, define $\Phi_x(\mu_y) = \text{Law}(X^{\mu_y}) \in \mathcal{P}(C([0,T], \mathcal{X}))$, and define $\Phi_y$ analogously. Notice that $\mathcal{W}_{2,t}^2(\Phi_x(\mu_y), \Phi_x(\nu_y)) \leqslant \mathbb{E}[\|X^{\mu_y} - X^{\nu_y}\|_t^2], \mathcal{W}_{2,t}^2(\Phi_y(\mu_x), \Phi_y(\nu_x)) \leqslant \mathbb{E}[\|X^{\mu_x} - X^{\nu_x}\|_t^2]$. Hence, we obtain

$$\mathcal{W}_{2,t}^2(\Phi_x(\mu_y), \Phi_x(\nu_y)) + \mathcal{W}_{2,t}^2(\Phi_y(\mu_x), \Phi_y(\nu_x)) \leqslant C_T \int_0^t \mathcal{W}_{2,r}^2(\mu_y, \nu_y) + \mathcal{W}_{2,r}^2(\mu_x, \nu_x) \, dr$$

Observe that $\mathcal{W}_{2,t}^2(\mu_x, \nu_x) + \mathcal{W}_{2,t}^2(\mu_y, \nu_y)$ is the square of a distance between $(\mu_x, \mu_y)$ and $(\nu_x, \nu_y)$ on $\mathcal{P}(C([0,T], \mathcal{X})) \times \mathcal{P}(C([0,T], \mathcal{Y}))$. Hence, we can apply the Piccard iteration argument to obtain the existence result and another application of Gronwall's inequality yields pathwise uniqueness.

Uniqueness in law (i.e., regardless of the specific Brownian motions and initialization random variables) follows from the typical uniqueness in law result for SDEs (see Chapter 18 of Kallenberg (2002) for example). The idea is that when we solve the SDEs with $W', \bar{W}', \xi', \bar{\xi}'$ plugging in the drift the laws of a solution for $W, \bar{W}, \xi, \bar{\xi}$, the solution has the same laws by uniqueness in law of SDEs. Hence, that new solution solves the coupled McKean-Vlasov for $W', \bar{W}', \xi', \bar{\xi}'$.

### E.3 Propagation of chaos

Let $\mu_x^n = \frac{1}{n}\sum_{i=1}^n \delta_{X^i}, \mu_y^n = \frac{1}{n}\sum_{i=1}^n \delta_{Y^i}$. Using the argument from existence and uniqueness on the $i$-th components of $\mathbf{X}, \tilde{\mathbf{X}}$,

$$\mathbb{E}[\|X^i - \tilde{X}^i\|_t^2] \leqslant 3(3T+4)\tilde{L}^2\mathbb{E}\left[\int_0^t (\|X^i - \tilde{X}^i\|_r^2 + \mathcal{W}_2^2(\mu_{y,r}^n, \mu_{y,r})) \, dr\right]$$

Arguing as before, we obtain

$$\mathbb{E}[\|X^i - \tilde{X}^i\|_t^2] \leqslant C_T\mathbb{E}\left[\int_0^t \mathcal{W}_{2,r}^2(\mu_y^n, \mu_y) \, dr\right]$$

Let $\nu_x^n = \frac{1}{n}\sum_{i=1}^n \delta_{\tilde{X}^i}$ be the empirical measure of the mean field processes in (44). Notice that $\frac{1}{n}\sum_{i=1}^n \delta_{(X^i, \tilde{X}^i)}$ is a coupling between $\nu_x^n$ and $\mu_x^n$, and so

$$\mathcal{W}_{2,t}^2(\mu_x^n, \nu_x^n) \leqslant \frac{1}{n}\sum_{i=1}^n \|X^i - \tilde{X}^i\|_t^2$$

Thus, we obtain

$$\mathbb{E}[\mathcal{W}_{2,t}^2(\mu_x^n, \nu_x^n)] \leqslant C_T\mathbb{E}\left[\int_0^t \mathcal{W}_{2,r}^2(\mu_y^n, \mu_y) \, dr\right]$$

We use the triangle inequality

$$\mathbb{E}[\mathcal{W}_{2,t}^2(\mu_x^n, \mu_x)] \leqslant 2\mathbb{E}[\mathcal{W}_{2,t}^2(\mu_x^n, \nu_x^n)] + 2\mathbb{E}[\mathcal{W}_{2,t}^2(\nu_x^n, \mu_x)]$$
$$\leqslant 2C_T\mathbb{E}\left[\int_0^t \mathcal{W}_{2,r}^2(\mu_y^n, \mu_y) \, dr\right] + 2\mathbb{E}[\mathcal{W}_{2,t}^2(\nu_x^n, \mu_x)]$$

At this point we follow an analogous procedure for the other player and we end up with

$$\mathbb{E}[\mathcal{W}_{2,t}^2(\mu_x^n, \mu_x) + \mathcal{W}_{2,t}^2(\mu_y^n, \mu_y)] \leqslant 2C_T\mathbb{E}\left[\int_0^t \mathcal{W}_{2,r}^2(\mu_y^n, \mu_y) + \mathcal{W}_{2,r}^2(\mu_x^n, \mu_x) \, dr\right]$$
$$+ 2\mathbb{E}[\mathcal{W}_{2,t}^2(\nu_x^n, \mu_x) + \mathcal{W}_{2,t}^2(\nu_y^n, \mu_y)]$$

We use Fubini's theorem and Gronwall's inequality again.

$$\mathbb{E}[\mathcal{W}_{2,t}^2(\mu_x^n, \mu_x) + \mathcal{W}_{2,t}^2(\mu_y^n, \mu_y)] \leqslant 2\exp(2C_T T)\mathbb{E}[\mathcal{W}_{2,t}^2(\nu_x^n, \mu_x) + \mathcal{W}_{2,t}^2(\nu_y^n, \mu_y)]$$

If we set $t = T$ we get

$$\mathbb{E}[\mathcal{W}_2^2(\mu_x^n, \mu_x) + \mathcal{W}_2^2(\mu_y^n, \mu_y)] \leqslant 2\exp(2C_T T)\mathbb{E}[\mathcal{W}_2^2(\nu_x^n, \mu_x) + \mathcal{W}_2^2(\nu_y^n, \mu_y)]$$

and the factor $\mathbb{E}[\mathcal{W}_2^2(\nu_x^n, \mu_x) + \mathcal{W}_2^2(\nu_y^n, \mu_y)]$ goes to 0 as $n \to \infty$ by the law of large numbers (see Corollary 2.14 of (Lacker, 2018)).

### E.4 Convergence of the Nikaido-Isoda error

**Corollary 2.** *For $t \in [0, T]$, if $\mu_{x,t}^n, \mu_{x,t}, \mu_{y,t}^n, \mu_{y,t}$ are the marginals of $\mu_x^n, \mu_x, \mu_y^n, \mu_y$ at time $t$, we have*

$$\mathbb{E}[|NI(\mu_{x,t}^n, \mu_{y,t}^n) - NI(\mu_{x,t}, \mu_{y,t})|] \xrightarrow{n \to \infty} 0$$

*Proof.* See Lemma 3. □

## F  Proof of Theorem 3(ii)

### F.1  Preliminaries

Define the processes $\mathbf{X} = (X^1, \ldots, X^n), \mathbf{w}_x = (w_x^1, \ldots, w_x^n)$ and $\mathbf{Y} = (Y^1, \ldots, Y^n), \mathbf{w}_y = (w_y^1, \ldots, w_y^n)$ such that for all $i \in \{1, \ldots, n\}$

$$\frac{dX_t^i}{dt} = -\gamma \frac{1}{n} \sum_{j=1}^n w_{y,t}^j \nabla_x \ell(X_t^i, Y_t^j), \quad X_0^i = \xi^i \sim \mu_{x,0}$$

$$\frac{dw_{x,t}^i}{dt} = \alpha \left( -\frac{1}{n} \sum_{j=1}^n w_{y,t}^j \ell(X_t^i, Y_t^j) + \frac{1}{n^2} \sum_{k=1}^n \sum_{j=1}^n w_{y,t}^j w_{x,t}^k \ell(X_t^i, Y_t^j) \right) w_{x,t}^i, \quad w_{x,0}^i = 1$$

$$\frac{dY_t^i}{dt} = \gamma \frac{1}{n} \sum_{j=1}^n w_{x,t}^j \nabla_y \ell(X_t^j, Y_t^i), \quad Y_0^i = \bar{\xi}^i \sim \mu_{y,0} \tag{52}$$

$$\frac{dw_{y,t}^i}{dt} = \alpha \left( \frac{1}{n} \sum_{j=1}^n w_{x,t}^j \ell(X_t^i, Y_t^j) - \frac{1}{n^2} \sum_{k=1}^n \sum_{j=1}^n w_{y,t}^j w_{x,t}^k \ell(X_t^i, Y_t^j) \right) w_{x,t}^i, \quad w_{y,0}^i = 1$$

Let $\nu_{x,t}^n = \frac{1}{n} \sum_{i=1}^n \delta_{(X_t^i, w_{x,t}^i)} \in \mathbb{P}(\mathcal{X} \times \mathbb{R}^+), \nu_{y,t}^n = \frac{1}{n} \sum_{i=1}^n \delta_{(Y_t^i, r_{y,t}^{n,i})} \in \mathbb{P}(\mathcal{Y} \times \mathbb{R}^+)$. Let $\mu_{x,t}^n = \frac{1}{n} \sum_{i=1}^n w_{x,t}^i \delta_{X_t^i} \in \mathbb{P}(\mathcal{X}), \mu_{y,t}^n = \frac{1}{n} \sum_{i=1}^n w_{y,t}^i \delta_{Y_t^i} \in \mathbb{P}(\mathcal{Y})$ be the projections of $\nu_{x,t}^n, \nu_{y,t}^n$. Notice that we have changed the notation with respect to the main text, multiplying $w_x^i$ by $n$: now $w_{x,0}^i = 1$ and $\sum_i w_{x,t}^i = n, \forall t \geqslant 0$ instead of $w_{x,0}^i = 1/n$ and $\sum_i w_{x,t}^i = 1, \forall t \geqslant 0$.

Let $h_x, h_y$ be the projection operators, i.e. $h_x \nu_x = \int_{\mathcal{R}^+} w_x \nu_x(\cdot, w_x)$. We also define the mean field processes $\tilde{\mathbf{X}}, \tilde{\mathbf{Y}}, \tilde{\mathbf{w}}_x, \tilde{\mathbf{w}}_y$ given component-wise by

$$\frac{d\tilde{X}_t^i}{dt} = -\gamma \nabla_x \int \ell(\tilde{X}_t^i, y) d\mu_{y,t}, \quad \tilde{X}_0^i = \xi^i \sim \mu_{x,0}$$

$$\frac{d\tilde{w}_{x,t}^i}{dt} = \alpha \left( -\int \ell(\tilde{X}_t^i, y) d\mu_{y,t} + \mathcal{L}(\mu_{x,t}, \mu_{y,t}) \right) \tilde{w}_{x,t}^i, \quad \tilde{w}_{x,0}^i = 1$$

$$\frac{d\tilde{Y}_t^i}{dt} = \gamma \nabla_y \int \ell(x, \tilde{Y}_t^i) d\mu_{x,t}, \quad \tilde{Y}_0^i = \bar{\xi}^i \sim \mu_{y,0} \tag{53}$$

$$\frac{d\tilde{w}_{y,t}^i}{dt} = \alpha \left( \int \ell(x, \tilde{Y}_t^i) d\mu_{x,t} - \mathcal{L}(\mu_{x,t}, \mu_{y,t}) \right) \tilde{w}_{x,t}^i, \quad \tilde{w}_{y,0}^i = 1$$

$$\mu_{x,t} = h_x \text{Law}(\tilde{X}_t^i, \tilde{w}_{x,t}^i), \quad \mu_{y,t} = h_y \text{Law}(\tilde{Y}_t^i, \tilde{w}_{y,t}^i)$$

for $i$ between 1 and $n$.

**Lemma 11** (Forward Kolmogorov equation). *If $\tilde{X}, \tilde{w}_x, \tilde{Y}, \tilde{w}_y$ is a solution of (53) with $n = 1$, then its laws $\nu_x, \nu_y$ fulfill (10).*

*Proof.* Let $\psi_x : \mathcal{X} \times \mathbb{R}^+ \to \mathbb{R}$. Plug the laws $\nu_x, \nu_y$ of the solution $(\tilde{X}, \tilde{w}_x), (\tilde{Y}, \tilde{w}_y)$ into the ODE (53). Let $\Phi_{x,t} = (X_{x,t}^\Phi, w_{x,t}^\Phi) : (\mathcal{X} \times \mathbb{R}^+) \to (\mathcal{X} \times \mathbb{R}^+)$ denote the flow that maps an initial condition

of the ODE (53) to the corresponding solution at time $t$. Then, we can write $\nu_{x,t} = (\Phi_{x,t})_* \nu_{x,0}$, where $(\Phi_{x,t})_*$ is the pushforward. Hence,

$$\frac{d}{dt} \int_{\mathcal{X} \times \mathbb{R}^+} \psi_x(x, w_x) \, d\nu_{x,t}(x, w_x)$$

$$= \frac{d}{dt} \int_{\mathcal{X} \times \mathbb{R}^+} \psi_x(\Phi_{x,t}(x, w_x)) \, d\nu_{x,0}(x, w_x)$$

$$= \int_{\mathcal{X} \times \mathbb{R}^+} \left( \nabla_x \psi_x(\Phi_{x,t}(x, w_x)), \frac{d\psi_x}{dw_x}(\Phi_{x,t}(x, w_x)) \right) \cdot \frac{d}{dt} \Phi_{x,t}(x, w_x) \, d\nu_{x,0}(x, w_x)$$

$$= \int_{\mathcal{X} \times \mathbb{R}^+} \nabla_x \psi_x(\Phi_{x,t}(x, w_x)) \cdot (-\gamma \nabla_x V_x(h_y \nu_{y,t}, X_{x,t}^{\Phi}))$$

$$+ \frac{d\psi_x}{dw_x}(\Phi_{x,t}(x, w_x))\alpha(-V_x(h_y \nu_{y,t}, X_{x,t}^{\Phi}) + \mathcal{L}(h_x \nu_{x,t}, h_y \mu_{y,t})) \, d\nu_{x,0}(x, w_x)$$

And we can identify the right hand side as the weak form of (10), shown in (12). The argument for $\nu_y$ is analogous. $\quad\square$

We state a stronger version of the law of large numbers in the first statement of Theorem 3(ii).

**Theorem 10.** *There exists a solution of the coupled SDEs* (53). *Pathwise uniqueness and uniqueness in law hold. Let* $\nu_x \in \mathcal{P}(\mathcal{C}([0,T], \mathcal{X} \times \mathbb{R}^+)), \nu_y \in \mathcal{P}(\mathcal{C}([0,T], \mathcal{Y} \times \mathbb{R}^+))$ *be the unique laws of the solutions for* $n = 1$ *(all pairs have the same solutions). Then,*

$$\mathbb{E}[\mathcal{W}_2^2(\nu_x^n, \nu_x) + \mathcal{W}_2^2(\nu_y^n, \nu_y)] \xrightarrow{n \to \infty} 0$$

Theorem 10 is the law of large numbers for the WFR dynamics, and its proof follows the same argument of Theorem 9. The reason Theorem 10 implies Theorem 3(ii) is analogous to the reason for which Theorem 9 implies Theorem 3(i), with the additional step that $\mathcal{W}_2^2(\mu_{x,t}^n, \mu_{x,t}) = \mathcal{W}_2^2(h_x \nu_{x,t}^n, h_x \nu_{x,t}) \leqslant e^{4MT} \mathcal{W}_2^2(\nu_{x,t}^n, \nu_{x,t})$, and this inequality is shown in (55).

## F.2 Existence and uniqueness

We choose to do an argument close to Sznitman (1991) (see Lacker (2018)), which yields convergence of the expectation of the square of the 2-Wasserstein distances between the empirical and the mean field measures.

First, to prove existence and uniqueness of the solution $(\mu_{x,t}, \mu_{y,t})$ in the time interval $[0, T]$ for arbitrary $T$, we can use the same argument as in the App. E. Now, instead of (47) we have

$$\tilde{X}_t = \xi - \gamma \int_0^t \int_{\mathcal{Y}} \nabla_x \ell(\tilde{X}_s, y) \, d\mu_{y,s} \, ds,$$

$$\tilde{w}_{x,t} = 1 + \alpha \int_0^t \left( -\int \ell(\tilde{X}_t, y) d\mu_{y,t} + \mathcal{L}(\mu_{x,t}, \mu_{y,t}) \right) \tilde{w}_{x,s} \, ds,$$

$$\tilde{Y}_t = \bar{\xi} + \gamma \int_0^t \int_{\mathcal{X}} \nabla_y \ell(x, \tilde{Y}_s) \, d\mu_{x,s} \, ds,$$

$$\tilde{w}_{y,t} = 1 + \alpha \int_0^t \left( \int \ell(x, \tilde{Y}_t) d\mu_{x,t} - \mathcal{L}(\mu_{x,t}, \mu_{y,t}) \right) \tilde{w}_{y,s} \, ds,$$

$$\mu_{x,t} = h_x \text{Law}(\tilde{X}_t, \tilde{w}_{x,t}), \quad \mu_{y,t} = h_y \text{Law}(\tilde{Y}_t, \tilde{w}_{y,t}),$$

where $\xi$ and $\bar{\xi}$ are arbitrary random variables with laws $\mu_{x,0}, \mu_{y,0}$ respectively. For $x, x' \in \mathcal{X}$, $r, r' \in \mathbb{R}^+$, $\mu_x, \mu_x' \in \mathcal{P}(\mathcal{X})$, $\mu_y, \mu_y' \in \mathcal{P}(\mathcal{Y})$, notice that using an argument similar to (48) the following bound holds

$$\left| \left( -\int \ell(x, y) d\mu_y + \mathcal{L}(\mu_x, \mu_y) \right) w - \left( -\int \ell(x', y) d\mu_y' + \mathcal{L}(\mu_x', \mu_y') \right) w' \right|$$

$$\leqslant 2M|w - w'| + |w'|\tilde{L}(|x - x'| + 3\mathcal{W}_1(\nu, \mu)) \leqslant 2M|w - w'| + |w'|\tilde{L}(|x - x'| + 3\mathcal{W}_2(\mu_y, \mu_y'))$$

$$\implies \left| \left( -\int \ell(x,y)d\mu_y + \mathcal{L}(\mu_x,\mu_y) \right) r - \left( -\int \ell(x',y)d\mu'_y + \mathcal{L}(\mu'_x,\mu'_y) \right) r' \right|^2$$

$$\leqslant 12M^2|w - w'|^2 + 3|w'|^2\tilde{L}^2(|x - x'|^2 + 9\mathcal{W}_2^2(\mu_y,\mu'_y))$$

Recall that $M$ is a bound on the absolute value of $\ell$ and $\tilde{L}$ is the Lipschitz constant of the loss $\ell$. A simple application of Gronwall's inequality shows $|\tilde{w}_{x,t}|$ is bounded by $e^{2MT}$ for all $t \in [0, T]$. Hence, we can write

$$\mathbb{E}[\|X^{\mu_y} - X^{\mu'_y}\|_t^2 + \|w_x^{\mu_y} - w_x^{\mu'_y}\|_t^2] \leqslant \gamma^2 t\mathbb{E}\left[ \int_0^t \left| \nabla_x \int \ell(X_s^{\mu_y},y)d\mu_{y,s} - \nabla_x \int \ell(X_s^{\mu'_y},y)d\mu'_{y,s} \right|^2 ds \right]$$

$$+\alpha^2 t\mathbb{E}\left[ \int_0^t \left| \left( -\int \ell(X_s^{\mu_y},y)d\mu_y + \mathcal{L}(\mu_x,\mu_y) \right) w_x^{\mu_y} - \left( -\int \ell(X_s^{\mu'_y},y)d\mu'_y + \mathcal{L}(\mu'_x,\mu'_y) \right) w_x^{\mu'_y} \right|^2 ds \right]$$

$$\leqslant Kt\mathbb{E}\left[ \int_0^t \|X^{\mu_y} - X^{\mu'_y}\|_s^2 + \|w^{\mu_y} - w^{\mu'_y}\|_s^2 \, ds \right] + K't\mathbb{E}\left[ \int_0^t \mathcal{W}_2^2(\mu_{y,s},\mu'_{y,s}) \, ds \right],$$

where $K = \max\{12\alpha^2 M^2, 2L^2\gamma^2 + 3\tilde{L}^2 e^{4MT}\alpha^2\}$, $K' = 2L^2\gamma^2 + 27\tilde{L}^2 e^{4MT}\alpha^2$. Notice that we have used (49) as well. This equation is analogous to equation (50), and upon application of Fubini's theorem and Gronwall's inequality it yields

$$\mathbb{E}[\|X^{\mu_y} - X^{\mu'_y}\|_t^2 + \|w_x^{\mu_y} - w_x^{\mu'_y}\|_t^2] \leqslant TK' \exp(TK)\mathbb{E}\left[ \int_0^t \mathcal{W}_2^2(\mu_{y,s},\mu'_{y,s}) \, ds \right] \qquad (54)$$

Now we will prove that

$$\mathcal{W}_2^2(h_x\nu_x, h_x\nu'_x) \leqslant e^{4MT}\mathcal{W}_2^2(\nu_x,\nu'_x), \qquad (55)$$

where $\nu_x, \nu'_x \in \mathcal{P}(\mathcal{X} \times [0, e^{2MT}])$. Define the homogeneous projection operator $\tilde{h} : \mathcal{P}((\mathcal{X} \times \mathbb{R}^+)^2) \to \mathcal{P}(\mathcal{X}^2)$ as $\forall f \in C(\mathcal{X}^2)$,

$$\int_{\mathcal{X}^2} f(x,y) \, d(\tilde{h}\pi)(x,y) = \int_{(\mathcal{X} \times [0,e^{2MT}])^2} w_x w_y f(x,y) \, d\pi(x,w_x,y,w_y), \ \forall \pi \in \mathcal{P}((\mathcal{X} \times \mathbb{R}^+)^2).$$

Let $\pi$ be a coupling between $h_x\nu_x, h_x\nu'_x$. Then $\tilde{h}\pi$ is a coupling between $h_x\nu_x, h_x\nu'_x$ and

$$\int_{\mathcal{X}^2} \|x - y\|^2 \, d(\tilde{h}\pi)(x,y) = \int_{(\mathcal{X} \times [0,e^{2MT}])^2} w_x w_y\|x - y\|^2 \, d\pi(x,w_x,y,w_y)$$

$$\leqslant e^{4MT}\int_{(\mathcal{X} \times [0,e^{2MT}])^2} \|x - y\|^2 \, d\pi(x,w_x,y,w_y)$$

$$\leqslant e^{4MT}\int_{(\mathcal{X} \times [0,e^{2MT}])^2} \|x - y\|^2 + |w_x - w_y|^2 \, d\pi'(x,w_x,y,w_y)$$

Taking the infimum with respect to $\pi$ on both sides we obtain the desired inequality.

Let $\nu_{x,t} = \mathrm{Law}(X_t^{\mu_y}, w_{x,t}^{\mu_y}), \nu'_{x,t} = \mathrm{Law}(X_t^{\mu'_y}, w_{x,t}^{\mu'_y})$ and recall that $\mu_{x,t} = h_x\nu_{x,t}, \mu'_{x,t} = h_x\nu'_{x,t}$. Given $\nu_y \in \mathcal{P}(C([0,T], \mathcal{Y} \times \mathbb{R}^+))$, define $\Phi_x(\nu_y) = \mathrm{Law}(X^{\nu_y}, w_x^{\nu_y}) \in \mathcal{P}(C([0,T], \mathcal{X}))$ where we abuse the notation and use $(X^{\nu_y}, w_x^{\nu_y})$ to refer to $(X^{\mu_y}, w_x^{\mu_y})$. Notice also that

$$\mathcal{W}_{2,t}^2(\Phi_x(\nu_y), \Phi_x(\nu'_y)) \leqslant \mathbb{E}\left[ \sup_{s \in [0,t]} \|X_s^{\mu_y} - X_s^{\mu'_y}\|^2 + \|w_{x,s}^{\mu_y} - w_{x,s}^{\mu'_y}\|^2 \right]$$

$$\leqslant \mathbb{E}[\|X^{\mu_y} - X^{\mu'_y}\|_t^2 + \|w_x^{\mu_y} - w_x^{\mu'_y}\|_t^2] \qquad (56)$$

We use (55) and (56) on (54) to conclude

$$\mathcal{W}_{2,t}^2(\Phi_x(\nu_y), \Phi_x(\nu'_y)) \leqslant TK' \exp(TK)\mathbb{E}\left[ \int_0^t \mathcal{W}_{2,s}^2(\nu_y, \nu'_y) \, ds \right]$$

The rest of the argument is sketched in App. E.

### F.3 Propagation of chaos

Following the reasoning in the existence and uniqueness proof, we can write

$$\mathbb{E}[\|X^i - \tilde{X}^i\|_t^2 + \|w_x^i - \tilde{w}_x^i\|_t^2]$$
$$\leqslant Kt\mathbb{E}\left[\int_0^t \|X^i - \tilde{X}^i\|_s^2 + \|w_x^i - \tilde{w}_x^i\|_s^2 \, ds\right] + K't\mathbb{E}\left[\int_0^t \mathcal{W}_2^2(\mu_{y,s}^n, \mu_{y,s}) \, ds\right],$$

Hence, we obtain

$$\mathbb{E}[\|X^i - \tilde{X}^i\|_t^2 + \|w_x^i - \tilde{w}_x^i\|_t^2] \leqslant TK' \exp(TK)\mathbb{E}\left[\int_0^t \mathcal{W}_2^2(\mu_{y,s}^n, \mu_{y,s}) \, ds\right]$$

Let $\tilde{\nu}_{x,t}^n = \frac{1}{n}\sum_{i=1}^n \delta_{(\tilde{X}_t^i, \tilde{w}_t^i)} \in \mathbb{P}(\mathcal{X} \times \mathbb{R}^+)$ be the marginal at time $t$ of the empirical measure of (52). As in App. E,

$$\mathcal{W}_{2,t}^2(\nu_x^n, \tilde{\nu}_x^n) \leqslant \frac{1}{n}\sum_{i=1}^n \sup_{s \in [0,t]} \|X_s^i - \tilde{X}_s^i\|^2 + |w_{x,s}^i - \tilde{w}_{x,s}^i|^2 \leqslant \frac{1}{n}\sum_{i=1}^n \|X^i - \tilde{X}^i\|_t^2 + \|w_x^i - \tilde{w}_x^i\|_t^2$$

which yields

$$\mathbb{E}[\mathcal{W}_{2,t}^2(\nu_x^n, \tilde{\nu}_x^n)] \leqslant TK' \exp(TK)\mathbb{E}\left[\int_0^t \mathcal{W}_2^2(\mu_{y,s}^n, \mu_{y,s}) \, ds\right]$$
$$\leqslant TK' \exp((K+4M)T)\mathbb{E}\left[\int_0^t \mathcal{W}_{2,s}^2(\nu_y^n, \nu_y) \, ds\right]$$

The second inequality above follows from inequality (55) $\mathcal{W}_2^2(\nu_{y,s}^n, \nu_{y,s}) \leqslant \mathcal{W}_{2,s}^2(\nu_y^n, \nu_y)$. Now we use the triangle inequality as in App. E:

$$\mathbb{E}[\mathcal{W}_{2,t}^2(\nu_x^n, \nu_x)] \leqslant 2\mathbb{E}[\mathcal{W}_{2,t}^2(\nu_x^n, \tilde{\nu}_x^n)] + 2\mathbb{E}[\mathcal{W}_{2,t}^2(\tilde{\nu}_x^n, \nu_x)]$$
$$\leqslant 2TK' \exp((K+4M)T)\mathbb{E}\left[\int_0^t \mathcal{W}_{2,s}^2(\nu_y^n, \nu_y) \, ds\right] + 2\mathbb{E}[\mathcal{W}_{2,t}^2(\tilde{\nu}_x^n, \nu_x)]$$

If we denote $C := 2TK' \exp((K+4M)T)$ and we make the same developments for the other player, we obtain

$$\mathbb{E}[\mathcal{W}_{2,t}^2(\nu_x^n, \nu_x) + \mathcal{W}_{2,t}^2(\nu_y^n, \nu_y)] \leqslant C\mathbb{E}\left[\int_0^t \mathcal{W}_{2,s}^2(\nu_y^n, \nu_y) + \mathcal{W}_{2,s}^2(\nu_x^n, \nu_x) \, ds\right]$$
$$+ 2\mathbb{E}[\mathcal{W}_{2,t}^2(\tilde{\nu}_x^n, \nu_x) + \mathcal{W}_{2,t}^2(\tilde{\nu}_y^n, \nu_y)]$$

From this point on, the proof works as in App. E.

### F.4 Convergence of the Nikaido-Isoda error

**Corollary 3.** *For $t \in [0, T]$, let $\bar{\mu}_{x,t}^n = \frac{1}{t}\int_0^t h_x \nu_{x,r}^n \, dr, \bar{\mu}_{x,t} = \frac{1}{t}\int_0^t h_x \nu_{x,r} \, dr$ and define $\bar{\mu}_{y,t}^n, \bar{\mu}_{y,t}$ analogously. Then,*

$$\mathbb{E}[|NI(\bar{\mu}_{x,t}^n, \bar{\mu}_{y,t}^n) - NI(\bar{\mu}_{x,t}, \bar{\mu}_{y,t})|] \xrightarrow{n \to \infty} 0$$

*Proof.* Notice that since the integral over time and the homogeneous projection commute, we have $\bar{\mu}_{x,t}^n = h_x(\frac{1}{t}\int_0^t \nu_{x,r}^n \, dr), \bar{\mu}_{x,t} = h_x(\frac{1}{t}\int_0^t \nu_{x,r} \, dr)$. Since $\frac{1}{t}\int_0^t \nu_{x,r}^n \, dr$ and $\frac{1}{t}\int_0^t \nu_{x,r} \, dr$ belong to $\mathcal{P}(\mathcal{X} \times [0, e^{2MT}])$, (55) implies

$$\mathcal{W}_2^2\left(h_x\left(\frac{1}{t}\int_0^t \nu_{x,r}^n \, dr\right), h_x\left(\frac{1}{t}\int_0^t \nu_{x,r} \, dr\right)\right) \leqslant e^{4MT}\mathcal{W}_2^2\left(\frac{1}{t}\int_0^t \nu_{x,r}^n \, dr, \frac{1}{t}\int_0^t \nu_{x,r} \, dr\right)$$

Notice that $\mathcal{W}_2^2(\frac{1}{t}\int_0^t \nu_{x,r}^n \, dr, \frac{1}{t}\int_0^t \nu_{x,r} \, dr) \leqslant \frac{1}{t}\int_0^t \mathcal{W}_2^2(\nu_{x,r}^n, \nu_{x,r}) \, dr$. Indeed,

$$
\begin{aligned}
\mathcal{W}_2^2 \left( \frac{1}{t}\int_0^t \nu_{x,r}^n \, dr, \frac{1}{t}\int_0^t \nu_{x,r} \, dr \right) &= \max_{\varphi \in \Psi_c(\mathcal{X})} \frac{1}{t}\int_0^t \int \varphi \, d\nu_{x,r}^n \, dr + \frac{1}{t}\int_0^t \int \varphi^c \, d\nu_{x,r}^n \, dr \\
&\leqslant \frac{1}{t}\int_0^t \left( \max_{\varphi \in \Psi_c(\mathcal{X})} \int \varphi \, d\nu_{x,r}^n + \int \varphi^c \, d\nu_{x,r}^n \right) \, dr \\
&= \frac{1}{t}\int_0^t \mathcal{W}_2^2(\nu_{x,r}^n, \nu_{x,r}) \, dr
\end{aligned}
$$

Hence, using the inequality $\mathcal{W}_2^2(\nu_{x,r}^n, \nu_{x,r}) \leqslant \mathcal{W}_2^2(\nu_x^n, \nu_x)$:

$$
\begin{aligned}
\mathbb{E}\left[ \mathcal{W}_2^2 \left( h_x \left( \frac{1}{t}\int_0^t \nu_{x,r}^n \, dr \right), h_x \left( \frac{1}{t}\int_0^t \nu_{x,r} \, dr \right) \right) \right] &\leqslant e^{4MT} \mathbb{E}\left[ \frac{1}{t}\int_0^t \mathcal{W}_2^2(\nu_{x,r}^n, \nu_{x,r}) \, dr \right] \\
&\leqslant e^{4MT} \mathbb{E}[\mathcal{W}_2^2(\nu_x^n, \nu_x)]
\end{aligned}
$$

Since the right hand side goes to zero as $n \to \infty$ by Theorem 10, we conclude by applying Lemma 3. $\qquad\square$

## F.5 Hint of the infinitesimal generator approach

Let $\varphi_x : \mathcal{X} \to \mathbb{R}, \varphi_y : \mathcal{Y} \to \mathbb{R}$ be arbitrary continuously differentiable functions, i.e. $\varphi_x \in C^1(\mathcal{X}, \mathbb{R}), \varphi_y \in C^1(\mathcal{Y}, \mathbb{R})$. Let us define the operators $\mathcal{L}_{x,t}^{(n)} : C^1(\mathcal{X}, \mathbb{R}) \to C^0(\mathcal{X}, \mathbb{R}), \mathcal{L}_{y,t}^{(n)} : C^1(\mathcal{Y}, \mathbb{R}) \to C^0(\mathcal{Y}, \mathbb{R})$ as

$$
\begin{aligned}
\mathcal{L}_{x,t}^{(n)} \varphi_x(x) &= -\gamma \nabla_x \int \ell(x,y) d\mu_{y,t}^n \cdot \nabla_x \varphi_x(x) + \alpha \left( -\int \ell(x,y) d\mu_{y,t}^n + \mathcal{L}(\mu_{x,t}^n, \mu_{y,t}^n) \right) \\
\mathcal{L}_{y,t}^{(n)} \varphi_y(y) &= \gamma \nabla_y \int \ell(x,y) d\mu_{x,t}^n \cdot \nabla_y \varphi_y(x) + \alpha \left( \int \ell(x,y) d\mu_{x,t}^n - \mathcal{L}(\mu_{x,t}^n, \mu_{y,t}^n) \right)
\end{aligned}
\tag{57}
$$

Notice that from (52) and (57), we have

$$
\begin{aligned}
\frac{d}{dt}\int_{\mathcal{X}} \varphi_x(x) \, d\mu_{x,t}^n(x) &= \frac{d}{dt}\int_{\mathcal{X}\times\mathbb{R}^+} w_x \varphi_x(x) \, d\nu_{x,t}^n(x, w_x) = \frac{d}{dt}\sum_{i=1}^n w_{x,t}^i \varphi_x(X_t^i) \\
&= \sum_{i=1}^n \frac{dw_{x,t}^i}{dt}\varphi_x(X_t^i) + \sum_{i=1}^n w_{x,t}^i \nabla_x \varphi_x(X_t^i) \cdot \frac{dX_t^i}{dt} \\
&= \int_{\mathcal{X}\times\mathbb{R}^+} w_x \mathcal{L}_{x,t}^{(n)} \varphi_x(x) \, d\nu_{x,t}^n(x, w_x) = \int_{\mathcal{X}} \mathcal{L}_{x,t}^{(n)} \varphi_x(x) \, d\mu_{x,t}^n(x)
\end{aligned}
\tag{58}
$$

The analogous equation holds for $\mu_{y,t}^n$:

$$
\frac{d}{dt}\int_{\mathcal{Y}} \varphi_y(y) \, d\mu_{y,t}^n(y) = \int_{\mathcal{Y}} \mathcal{L}_{y,t}^{(n)} \varphi_y(y) \, d\mu_{y,t}^n(y)
\tag{59}
$$

Formally taking the limit $n \to \infty$ on (58) and (59) yields

$$
\begin{aligned}
\frac{d}{dt}\int_{\mathcal{X}} \varphi_x(x) \, d\mu_{x,t}(x) &= \int_{\mathcal{X}} \mathcal{L}_{x,t} \varphi_x(x) \, d\mu_{x,t}(x) \\
\frac{d}{dt}\int_{\mathcal{Y}} \varphi_y(y) \, d\mu_{y,t}(y) &= \int_{\mathcal{Y}} \mathcal{L}_{y,t} \varphi_y(y) \, d\mu_{y,t}(y),
\end{aligned}
$$

where

$$
\begin{aligned}
\mathcal{L}_{x,t} \varphi_x(x) &= -\gamma \nabla_x \int \ell(x,y) d\mu_{y,t} \cdot \nabla_x \varphi_x(x) + \alpha \left( -\int \ell(x,y) d\mu_{y,t} + \mathcal{L}(\mu_{x,t}, \mu_{y,t}) \right) \\
\mathcal{L}_{y,t} \varphi_y(y) &= \gamma \nabla_y \int \ell(x,y) d\mu_{x,t} \cdot \nabla_y \varphi_y(x) + \alpha \left( \int \ell(x,y) d\mu_{x,t} - \mathcal{L}(\mu_{x,t}, \mu_{y,t}) \right)
\end{aligned}
$$

and $\mu_{x,0}, \mu_{y,0}$ are set as in (52).

To make the limit $n \to \infty$ rigorous, an argument analogous to Theorem 2.6 of Chizat and Bach (2018) would result in almost sure convergence of the 2-Wasserstein distances between the empirical and the mean field measures. In our case almost sure convergence of the squared distance implies convergence of the expectation of the squared distance through dominated convergence, and hence the almost sure convergence result is stronger. Nonetheless, such an argument would require proving uniqueness of the mean field measure PDE through some notion of geodesic convexity, which is not clear in our case.

## G  Auxiliary material

### G.1  $\varepsilon$-Nash equilibria and the Nikaido-Isoda error

Recall that an $\varepsilon$-NE $(\mu_x, \mu_y)$ satisfies $\forall \mu_x^* \in \mathcal{P}(\mathcal{X})$, $\mathcal{L}(\mu_x, \mu_y) \leqslant \mathcal{L}(\mu_x^*, \mu_y) + \varepsilon$ and $\forall \mu_y^* \in \mathcal{P}(\mathcal{Y})$, $\mathcal{L}(\mu_x, \mu_y) \geqslant \mathcal{L}(\mu_x, \mu_y^*) - \varepsilon$. That is, each player can improve its value by at most $\varepsilon$ by deviating from the equilibrium strategy, supposing that the other player is kept fixed.

Recall the Nikaido-Isoda error defined in (2). This equation can be rewritten as:

$$\mathrm{NI}(\mu_x, \mu_y) = \sup_{\mu_y^* \in \mathcal{P}(\mathcal{Y})} \mathcal{L}(\mu_x, \mu_y^*) - \mathcal{L}(\mu_x, \mu_y) + \mathcal{L}(\mu_x, \mu_y) - \inf_{\mu_x^* \in \mathcal{P}(\mathcal{X})} \mathcal{L}(\mu_x^*, \mu_y) .$$

The terms $\sup_{\mu_y^* \in \mathcal{P}(\mathcal{Y})} \mathcal{L}(\mu_x, \mu_y^*) - \mathcal{L}(\mu_x, \mu_y) > 0$ measure how much player $y$ can improve its value by deviating from $\mu_y$ while $\mu_x$ stays fixed. Analogously, the terms $\mathcal{L}(\mu_x, \mu_y) - \inf_{\mu_x^* \in \mathcal{P}(\mathcal{X})} \mathcal{L}(\mu_x^*, \mu_y) > 0$ measure how much player $x$ can improve its value by deviating from $\mu_x$ while $\mu_y$ stays fixed.

Notice that

$$\forall \mu_x^* \in \mathcal{P}(\mathcal{X}), \ \mathcal{L}(\mu_x, \mu_y) \leqslant \mathcal{L}(\mu_x^*, \mu_y) + \varepsilon \iff \mathcal{L}(\mu_x, \mu_y) - \inf_{\mu_x^* \in \mathcal{P}(\mathcal{X})} \mathcal{L}(\mu_x^*, \mu_y) \leqslant \varepsilon$$

$$\forall \mu_y^* \in \mathcal{P}(\mathcal{Y}), \ \mathcal{L}(\mu_x, \mu_y) \geqslant \mathcal{L}(\mu_x, \mu_y^*) - \varepsilon \iff \sup_{\mu_y^* \in \mathcal{P}(\mathcal{Y})} \mathcal{L}(\mu_x, \mu_y^*) - \mathcal{L}(\mu_x, \mu_y) \leqslant \varepsilon$$

Thus, an $\varepsilon$-Nash equilibrium $(\mu_x, \mu_y)$ fulfills $\mathrm{NI}(\mu_x, \mu_y) \leqslant 2\varepsilon$, and any pair $(\mu_x, \mu_y)$ such that $\mathrm{NI}(\mu_x, \mu_y) \leqslant \varepsilon$ is an $\varepsilon$-Nash equilibrium.

### G.2  Example: failure of the Interacting Wasserstein Gradient Flow

Let us consider the polynomial $f(x) = 5x^4 + 10x^2 - 2x$, which is an asymmetric double well as shown in Fig. 4.

Figure 4: Plot of the function $f(x) = 5x^4 + 10x^2 - 2x$.

Let us define the loss $\ell : \mathbb{R} \times \mathbb{R} \to \mathbb{R}$ as $\ell(x, y) = f(x) - f(y)$. That is, the two players are non-interacting and hence we obtain $V_x(x, \mu_y) = f(x) + K$, $V_y(y, \mu_x) = -f(y) + K'$. This means that the IWGF in equation (6) becomes two independent Wasserstein Gradient Flows

$$\partial_t \mu_x = \nabla \cdot (\mu_x f'(x)), \quad \mu_x(0) = \mu_{x,0},$$
$$\partial_t \mu_y = -\nabla \cdot (\mu_y f'(y)), \quad \mu_y(0) = \mu_{y,0}.$$

The particle flows in (3) become

$$\frac{dx_i}{dt} = -f'(x_i), \quad \frac{dy_i}{dt} = f'(y_i).$$

That is, the particles of player $x$ follow the gradient flow of $f$ and the particles of player $y$ follow the gradient flow of $-f$. It is clear from Fig. 4 that if the initializations $x_{0,i}, y_{0,i}$ are on the left of the barrier, they will not end up in the global minimum $f$ (resp., the global maximum of $-f$). And in this case, the pair of measures supported on the global minimum of $f$ is the only (pure) Nash equilibrium.

The game given by $\ell$ does not fall exactly in the framework that we describe in this work because $\ell$ is not defined on compact spaces. However, it is easy to construct very similar continuously differentiable functions on compact spaces that display the same behavior.

### G.3 Link between Interacting Wasserstein Gradient Flow and interacting particle gradient flows

Recall (3):

$$\frac{dx_i}{dt} = -\frac{1}{n}\sum_{j=1}^{n}\nabla_x\ell(x_i, y_j), \quad \frac{dy_i}{dt} = \frac{1}{n}\sum_{j=1}^{n}\nabla_x\ell(x_j, y_i).$$

Let $\Phi_t = (\Phi_{x,t}, \Phi_{y,t}) : \mathcal{X}^n \times \mathcal{Y}^n \to \mathcal{X}^n \times \mathcal{Y}^n$ be the flow mapping initial conditions $\mathbf{X}_0 = (x_{i,0})_{i\in[1:n]}, \mathbf{Y}_0 = (y_{i,0})_{i\in[1:n]}$ to the solution of (3). Let $\mu_{x,t}^n = \frac{1}{n}\sum_{i=1}^{n}\delta_{\Phi_{x,t}^{(i)}(\mathbf{X}_0,\mathbf{Y}_0)}, \mu_{y,t}^n = \frac{1}{n}\sum_{i=1}^{n}\delta_{\Phi_{y,t}^{(i)}(\mathbf{X}_0,\mathbf{Y}_0)}$. For all $\psi_x \in \mathcal{C}(\mathcal{X})$,

$$\frac{d}{dt}\int_{\mathcal{X}}\psi_x(x)\,d\mu_{x,t}^n(x) = \frac{1}{n}\sum_{i=1}^{n}\frac{d}{dt}\psi_x(\Phi_{x,t}^{(i)}(\mathbf{X}_0,\mathbf{Y}_0))$$

$$= \frac{1}{n}\sum_{i=1}^{n}\nabla_x\psi_x(\Phi_{x,t}^{(i)}(\mathbf{X}_0,\mathbf{Y}_0)) \cdot \left(-\frac{1}{n}\sum_{j=1}^{n}\nabla_x\ell(\Phi_{x,t}^{(i)}(\mathbf{X}_0,\mathbf{Y}_0), \Phi_{y,t}^{(j)}(\mathbf{X}_0,\mathbf{Y}_0))\right)$$

$$= -\frac{1}{n}\sum_{i=1}^{n}\nabla_x\psi_x(\Phi_{x,t}^{(i)}(\mathbf{X}_0,\mathbf{Y}_0)) \cdot \nabla_x V_x(\mu_{y,t}^n, \Phi_{x,t}^{(i)}(\mathbf{X}_0,\mathbf{Y}_0))$$

$$= -\int_{\mathcal{X}}\nabla_x\psi_x(x) \cdot \nabla_x V_x(\mu_{y,t}^n, x)\,d\mu_{x,t}^n(x),$$

which is the first line of (6). The second line follows analogously.

### G.4 Minimax problems and Stackelberg equilibria

Several machine learning problems, including GANs, are framed as a minimax problem

$$\min_{x\in\mathcal{X}}\max_{y\in\mathcal{Y}}\ell(x, y).$$

A minimax point (also known as a Stackelberg equilibrium or sequential equilibrium) is a pair $(\tilde{x}, \tilde{y})$ at which the minimum and maximum of the problem are attained, i.e.

$$\begin{cases}\min_{x\in\mathcal{X}}\max_{y\in\mathcal{Y}}\ell(x, y) = \max_{y\in\mathcal{Y}}\ell(\tilde{x}, y)\\ \max_{y\in\mathcal{Y}}\ell(\tilde{x}, y) = \ell(\tilde{x}, \tilde{y})\end{cases}.$$

We consider the lifted version of the minimax problem (G.4) in the space of probability measures.

$$\min_{\mu_x\in\mathcal{P}(\mathcal{X})}\max_{\mu_y\in\mathcal{P}(\mathcal{Y})}\mathcal{L}(\mu_x, \mu_y). \tag{60}$$

By the generalized Von Neumann's minimax theorem, a Nash equilibrium of the game given by $\mathcal{L}$ is a solution of the lifted minimax problem (60) (see Lemma 12 in the case $\varepsilon = 0$).

The converse is not true: minimax points (solutions of (60)) are not necessarily mixed Nash equilibria even in the case where the loss function is convex-concave. An example is $\mathcal{L} : \mathbb{R} \times \mathbb{R} \to \mathbb{R}$ given by $\mathcal{L}(\mu_x, \mu_y) = \iint(x^2 + 2xy)\,d\mu_x\,d\mu_y$. Let $\mathcal{M}$ be the set of measures $\mu \in \mathcal{P}(\mathbb{R})$ such that $\int x\,d\mu = 0$. Notice that any pair $(\delta_0, \mu_y)$ with $\mu_y \in \mathcal{P}(\mathbb{R})$ is a minimax point. That is because

$$\max_{\mu_y\in\mathcal{P}(\mathbb{R})}\mathcal{L}(\mu_x, \mu_y) = \begin{cases}+\infty & \text{if } \mu_x \notin \mathcal{M}\\ \text{positive} & \text{if } \mu_x \in \mathcal{M} \setminus \{\delta_0\}\\ 0 & \text{if } \mu_x = \delta_0,\end{cases}$$

and hence $\delta_0 = \mathrm{argmin}_{\mu_x \in \mathcal{P}(\mathbb{R})} \max_{\mu_y \in \mathcal{P}(\mathbb{R})} \mathcal{L}(\mu_x, \mu_y)$. But if $\mu_x = \delta_0$, we have $\mathrm{argmax}_{\mu_y \in \mathcal{P}(\mathbb{R})} \mathcal{L}(\mu_x, \mu_y) = \mathcal{P}(\mathbb{R})$, because for all measures $\mu_y \in \mathcal{P}(\mathbb{R})$, $\mathcal{L}(\delta_0, \mu_y) = 0$. However, for $\mu_y \notin \mathcal{M}$, $\mathcal{L}(\mu_x, \mu_y)$ as a function of $\mu_x$ does not have a minimum at $\delta_0$, but at $\delta_{-\int y \, d\mu_y}$. Hence, the only mixed Nash equilibria are of the form $(\delta_0, \mu_y)$, with $\mu_y \in \mathcal{M}$.

The intuition behind the counterexample is that minimax points only require the minimizing player to be non-exploitable, but the maximizing player is only subject to a weaker condition.

We define a $\varepsilon$-minimax point (or $\varepsilon$-Stackelberg equilibrium) of an objective $\mathcal{L}(\mu_x, \mu_y)$ as a couple $(\tilde{\mu}_x, \tilde{\mu}_y)$ such that

$$
\begin{cases}
\min_{\mu_x \in \mathcal{P}(\mathcal{X})} \max_{\mu_y \in \mathcal{P}(\mathcal{Y})} \mathcal{L}(\mu_x, \mu_y) \geqslant \max_{\mu_y \in \mathcal{P}(\mathcal{Y})} \mathcal{L}(\tilde{\mu}_x, \mu_y) - \varepsilon \\
\max_{\mu_y \in \mathcal{P}(\mathcal{Y})} \mathcal{L}(\tilde{\mu}_x, \mu_y) \leqslant \mathcal{L}(\tilde{\mu}_x, \tilde{\mu}_y) + \varepsilon
\end{cases}.
$$

**Lemma 12.** *An $\varepsilon$-Nash equilibrium is a $2\varepsilon$-minimax point, and it holds that*

$$
\min_{\mu_x \in \mathcal{P}(\mathcal{X})} \max_{\mu_y \in \mathcal{P}(\mathcal{Y})} \mathcal{L}(\mu_x, \mu_y) - \varepsilon \leqslant \mathcal{L}(\hat{\mu}_x, \hat{\mu}_y) \leqslant \max_{\mu_y \in \mathcal{P}(\mathcal{Y})} \min_{\mu_x \in \mathcal{P}(\mathcal{X})} \mathcal{L}(\mu_x, \hat{\mu}_y) + \varepsilon
$$

*Proof.* Let $(\hat{\mu}_x, \hat{\mu}_y)$ be an $\varepsilon$-Nash equilibrium. Notice that $\max_{\mu_y \in \mathcal{P}(\mathcal{Y})} \min_{\mu_x \in \mathcal{P}(\mathcal{X})} \mathcal{L}(\tilde{\mu}_x, \mu_y) \leqslant \min_{\mu_x \in \mathcal{P}(\mathcal{X})} \max_{\mu_y \in \mathcal{P}(\mathcal{Y})} \mathcal{L}(\tilde{\mu}_x, \mu_y)$. Also,

$$
\begin{aligned}
\min_{\mu_x \in \mathcal{P}(\mathcal{X})} \max_{\mu_y \in \mathcal{P}(\mathcal{Y})} \mathcal{L}(\mu_x, \mu_y) &\leqslant \max_{\mu_y \in \mathcal{P}(\mathcal{Y})} \mathcal{L}(\hat{\mu}_x, \mu_y) \leqslant \mathcal{L}(\hat{\mu}_x, \hat{\mu}_y) + \varepsilon \leqslant \min_{\mu_x \in \mathcal{P}(\mathcal{X})} \mathcal{L}(\mu_x, \hat{\mu}_y) + 2\varepsilon \\
&\leqslant \max_{\mu_y \in \mathcal{P}(\mathcal{Y})} \min_{\mu_x \in \mathcal{P}(\mathcal{X})} \mathcal{L}(\mu_x, \hat{\mu}_y) + 2\varepsilon
\end{aligned}
\tag{61}
$$

and this yields the chain of inequalities in the statement of the theorem. The condition $\max_{\mu_y \in \mathcal{P}(\mathcal{Y})} \mathcal{L}(\tilde{\mu}_x, \mu_y) \leqslant \mathcal{L}(\tilde{\mu}_x, \tilde{\mu}_y) + \varepsilon$ of the definition of $\varepsilon$-minimax point follows directly from the definition of an $\varepsilon$-Nash equilibrium. Using part of (61), we get

$$
\max_{\mu_y \in \mathcal{P}(\mathcal{Y})} \mathcal{L}(\hat{\mu}_x, \mu_y) - 2\varepsilon \leqslant \max_{\mu_y \in \mathcal{P}(\mathcal{Y})} \min_{\mu_x \in \mathcal{P}(\mathcal{X})} \mathcal{L}(\mu_x, \hat{\mu}_y) \leqslant \min_{\mu_x \in \mathcal{P}(\mathcal{X})} \max_{\mu_y \in \mathcal{P}(\mathcal{Y})} \mathcal{L}(\tilde{\mu}_x, \mu_y),
$$

which is the first condition of a $2\varepsilon$-minimax. $\qquad\square$

Lemma 12 provides the link between approximate Nash equilibria and approximate Stackelberg equilibria, and it allows to translate our convergence results into minimax problems such as GANs.

### G.5 Itô SDEs on Riemannian manifolds: a parametric approach

We provide a brief summary on how to deal with SDEs on Riemannian manifolds and their corresponding Fokker-Planck equations (see Chapter 8 of Chirikjian (2009)). While ODEs have a straightforward translation into manifolds, the same is not true for SDEs. Recall that the definitions of the gradient and divergence for Riemannian manifolds are

$$
\nabla \cdot X = |g|^{-1/2} \partial_i (|g|^{1/2} X^i), \quad (\nabla f)^i = g^{ij} \partial_j f,
$$

where $g_{ij}$ is the metric tensor, $g^{ij} = (g_{ij})^{-1}$ and $|g| = \det(g_{ij})$. We use the Einstein convention for summing repeated indices.

The parametric approach to SDEs in manifolds is to define the SDE for the variables $\mathbf{q} = (q_1, \cdots, q_d)$ of a patch of the manifold:

$$
d\mathbf{q} = \mathbf{h}(\mathbf{q}, t) dt + H(\mathbf{q}, t) d\mathbf{w}.
\tag{62}
$$

The corresponding forward Kolmogorov equation is

$$
\frac{\partial f}{\partial t} + |g|^{-1/2} \sum_{i=1}^{d} \frac{\partial}{\partial q_i} \left( |g|^{1/2} h_i f \right) = \frac{1}{2} |g|^{-1/2} \sum_{i,j=1}^{d} \frac{\partial^2}{\partial q_i \partial q_j} \left( |g|^{1/2} \sum_{k=1}^{D} H_{ik} H_{kj}^\top f \right),
\tag{63}
$$

which is to be understood in the weak form.

Assume that the manifold $\mathcal{M}$ embedded in $\mathbb{R}^D$. If $\varphi : \mathcal{U}_{\mathbb{R}^d} \subseteq \mathbb{R}^d \to \mathcal{U} \subseteq \mathcal{M} \subseteq \mathbb{R}^D$ is the mapping corresponding to the patch $\mathcal{U}$ and (62) is defined on $\mathcal{U}_{\mathbb{R}^d}$, let us set $H(\mathbf{q}) = (D\varphi(\mathbf{q}))^{-1}$. In this case, $\sum_k H_{ik} H_{kj}^\top = \sum_k (D\varphi)_{ik}^{-1}((D\varphi)_{kj}^{-1})^\top = g^{ij}(\mathbf{q})$. Hence, the right hand side of (63) becomes

$$\frac{1}{2}|g|^{-1/2} \sum_{i,j=1}^d \frac{\partial^2}{\partial q_i \partial q_j}\left(|g|^{1/2} g^{ij} f\right)$$

$$= |g|^{-1/2} \sum_{i=1}^d \frac{\partial}{\partial q_i}\left(|g|^{1/2}\tilde{h}_i f\right) + \frac{1}{2}|g|^{-1/2} \sum_{i,j=1}^d \frac{\partial}{\partial q_i}\left(|g|^{1/2} g^{ij} \frac{\partial}{\partial q_j} f\right)$$

$$= |g|^{-1/2} \sum_{i=1}^d \frac{\partial}{\partial q_i}\left(|g|^{1/2}\tilde{h}_i f\right) + \frac{1}{2}|g|^{-1/2} \sum_{i,j=1}^d \frac{\partial}{\partial q_i}\left(|g|^{1/2} g^{ij} \frac{\partial}{\partial q_j} f\right)$$

$$= \nabla \cdot (\tilde{\mathbf{h}} f) + \frac{1}{2}\nabla \cdot \nabla f$$

where

$$\tilde{h}_i(\mathbf{q}) = \frac{1}{2}\sum_{j=1}^d \left(|g(\mathbf{q})|^{-1/2} g^{ij}(\mathbf{q}) \frac{\partial |G(\mathbf{q})|^{1/2}}{\partial q_j} + \frac{\partial g^{ij}(\mathbf{q})}{\partial q_j}\right)$$

Hence, we can rewrite (63) as

$$\frac{\partial f}{\partial t} = \nabla \cdot ((-\mathbf{h} + \tilde{\mathbf{h}})f) + \frac{1}{2}\nabla \cdot \nabla f$$

For this equation to be a Fokker-Planck equation with potential $E$ (i.e. with a Gibbs equilibrium solution), we need $-\mathbf{h} + \tilde{\mathbf{h}} = \nabla E$, which implies $\mathbf{h} = -\nabla E + \tilde{\mathbf{h}}$.

We can convert an SDE in parametric form like (62) into an SDE on $\mathbb{R}^D$ by using Ito's lemma on $X = \varphi(\mathbf{q})$:

$$dX_i = d\varphi_i(\mathbf{q}) = \left(D\varphi_i(\mathbf{q})\mathbf{h}(\mathbf{q}) + \frac{1}{2}\mathrm{Tr}(H(\mathbf{q},t)^\top (H\varphi_i)(\mathbf{q})H(\mathbf{q},t))\right) dt + D\varphi_i(\mathbf{q})H(\mathbf{q},t)d\mathbf{w} \quad (64)$$

If we set $H(\mathbf{q}) = (D\varphi(\mathbf{q}))^{-1}$ as before, $D\varphi(\mathbf{q})H(\mathbf{q},t)$ is the projection onto the tangent space of the manifold, i.e. $D\varphi(\mathbf{q})H(\mathbf{q},t)v = \mathrm{Proj}_{T_{\varphi(\mathbf{q})}M}v$, $\forall v \in \mathbb{R}^D$. In the case $\mathbf{h} = \nabla E + \tilde{\mathbf{h}}$, $D\varphi_i(\mathbf{q})\mathbf{h}(\mathbf{q}) = D\varphi_i(\mathbf{q})\nabla E(\mathbf{q}) + D\varphi_i(\mathbf{q})\tilde{\mathbf{h}}(\mathbf{q})$. It is very convenient to abuse the notation and denote $D\varphi(\mathbf{q})\nabla E(\mathbf{q})$ by $\nabla E(\varphi(\mathbf{q}))$. We also use $\hat{\mathbf{h}}(\varphi(\mathbf{q})) := D\varphi(\mathbf{q})\tilde{\mathbf{h}}(\mathbf{q}) + \frac{1}{2}\mathrm{Tr}(((D\varphi(\mathbf{q}))^{-1})^\top (H\varphi)(\mathbf{q})(D\varphi(\mathbf{q}))^{-1})$. Both definitions are well-defined because the variables are invariant by changes of coordinates. Hence, under these assumptions (64) becomes

$$dX = (-\nabla E(X) + \hat{\mathbf{h}}(X))\, dt + \mathrm{Proj}_{T_X M}(d\mathbf{w}) \quad (65)$$

In short that means that we can treat SDEs on embedded manifolds as SDEs on the ambient space by projecting the Brownian motions to the tangent space and adding a drift term $\hat{\mathbf{h}}$ that depends on the geometry of the manifold. Notice that for ODEs on manifolds the additional drift term does not appear and (65) reads simply $dX = \nabla E(X)dt$.

Notice that the forward Kolmogorov equation for (65) on $\mathbb{R}^D$ reads

$$\frac{d}{dt}\int f(x)\, d\mu_t(x) = \int (\nabla E(x) - \hat{\mathbf{h}}(x)) \cdot \nabla_x f(x) + \frac{1}{2}\mathrm{Tr}((\mathrm{Proj}_{T_x M})^\top Hf(x)\mathrm{Proj}_{T_x M})\, d\mu_t(x),$$

$$(66)$$

for an arbitrary $f$.

## Footnotes

[1] The metric of the manifold gives a natural choice of a Borel (volume) measure, the one given by integrating the canonical volume form.


[Supplementary Material 2 · quantitative_synthetic_shallow.pdf]



Shallow (underfitting) networks

- Legend:
  - 1G 1D
  - 3G 3D
  - 5G 5D

Y-axis: - Log-likelihood (15.0, 12.5, 10.0, 7.5, 5.0)

X-axis: Total G. iter ($10^3$, $10^4$)

[Supplementary Material 3]



Figure: Plot of $-$Log-likelihood versus Total G. iter. Legend: 1G 1D, 3G 3D, 5G 5D. Text annotation: "Deep (overfitting) networks".