[Reviews · NeurIPS 2020]

Review 1

Summary and Contributions: In this paper, the authors consider the problem of computing Nash equilibria for two-player zero-sum continuous games. In particular, the focus is on games with the loss function does not have the requisite convex-concave structure, and therefore, computing pure strategy Nash is computationally hard. Allowing mixed strategy Nash equilibria is similar to lifting non-convex problems into measure space to convexify them. Mixed Nash equilibria exist in the space of measures; therefore, the mixed strategy equilibria can be computed in polynomial time using mirror descent. That said, mirror descent is very inefficient for high dimensional problems. The approach taken in this paper is to parametrize mixed strategies as a weighted combination of delta functions supported on specific points (strategies), and then compute the optimal set of weights and positions. The problem is non-convex in this new parametrization, and so one has to develop new methods to establish the converge to the global optimum. The authors propose two algorithms for solving this problem -- one that uses noisy gradient descent to escape local minima (Alg 1) and another that uses gradient descent to update the position variables, and a multiplicative weight update for the weights (Alg 2). They show that these algorithms are plausible by first considering convergence properties of the the dynamics induced on the space of measures, and then establishing a law of large numbers to show that the finite particle algorithms are close to the continuum limit. They show the computationally efficiency of their method on a set of synthetic problems, and on GANs, both with synthetic and real life data.

Strengths: This is an interesting paper in that it proposes a non-convex approach to solving a problem that is convex -- the mixed Nash equilibrium or equivalently pure strategy Nash equilibrium over the space of measures is convex. The claim is that this non-convex problem converges faster than the canonical convex optimization approach, i.e. the mirror descent algorithm, for high dimensional problems. The authors show how to map the finite particle algorithms to corresponding dynamical system over the space of measures, and then establish that the dynamical system converges to an approximate Nash equilibrium in finite time. The dynamical system has connections to optimal transport. The proposed methodology is novel, and the possibility of interesting results in the future. This paper is clearly of interest to the NeurIPS community -- both from the perspective of the problem and the proposed methodology.

Weaknesses: There are several issues with this paper: 1) Why are mixed strategies of interest in the GAN setting where the goal is to compute a parameter setting for the forward NN g and the discriminator network f? The mixed strategy approach will compute a distribution over the parameters of the forward and discriminator networks -- how is one to interpret this? More generally, if the game of interest is truly non-convex (i.e. does not satisfy the proper convexity/concavity properties) relaxing the game to allow for mixed strategies convexifies the problem. But one is either left with the problem of generating a solution for the original game, or if one is okay with mixed strategies, then the game is convex to begin with. So, it is not clear what the authors mean when they say that they are addressing non-convex games? 2) Alg 2 appears to reduce to gradient descent in the x space, and multiplicative weight update in the w space. This may be more than just a passing resemblance since the multiplicative weight algorithm is also related to entropy smoothing -- the same approach used here. The authors should explore this connection more fully. 3) Given comment 2, is it really necessary to go down the route of posing the problem over measure spaces? Can one not analyze the finite particle problem directly. Put another way, what is the advantage of introducing the measure theoretic machinery if in the end one is analyzing a finite particle system? In particular, when, by the author's own admission, the convergence results for these methods are very weak -- finite time convergence and no rates, and also slow convergence because the entropic regularization dominates over the drift term. 4) In the very beginning of the introduction, the authors appear to be confused between multi-objective and multi-agent optimization. These are two, very different, problems. This paper is about multi-agent problems and not multi-objective problems.

Correctness: The claims and methods, to the best of my knowledge, appear to be correct.

Clarity: Reasonably well written -- not withstanding the confusion between multi-agent and multi-objective problem as detailed above. That said, the paper is more of a list of results, with no intuition as to why the proposed methodology is appropriate.

Relation to Prior Work: The paper approaches problem from a GAN/two person game perspective. However, there is a literature on regret minimization over experts and online optimization that also results in similar algorithms. The authors should discuss how their work fits with that literature.

Reproducibility: No

Additional Feedback: I read through the other reviews (to understand the issues raised there) and also the author responses. The responses to the comments are quite generic. For example, in response to the comment seeking a connection to previous work on experts, the authors responded by referring to previous work on the Nash equilibrium problems! Similarly, in response to the comment about scaling the algorithm to larger datasets, I read the response as stating that algorithm is not likely to scale well. I continue to think that the direction proposed in this paper is interesting, however, the authors need to address the issues raised in the reviews before I would be inclined to move my review from "accept" to something stronger.


Review 2

Summary and Contributions: This paper focuses on two player zero-sum continuous games and provide convergence guarantee for Wasserstein gradient descent-ascend dynamical systems in looking for mixed Nash equilibrium. It is an interesting submission in the sense that there are very few work in the literature dealing with manifold games from algorithmic perspective.

Strengths: This paper provides a concrete framework for two-player zero-sum games using all necessary techniques i.e. Fokker-Planck as the foundation, Langiven descent-ascent and Wasserstein descent-ascent as two schemes of discretization. Although natural to come up with these approach, but there are some nontrivial generalization from classic potential games to manifold zero-sum games in proving uniqueness of the solution.

Weaknesses: My main concern is the approximation of integrals for each step of update. It does not seem to be clear to me that there can be a good threshold for sample complexity in Langevin descent ascent at step 0. This seems difficult but it is a very critical issue. Another problem is that the authors did not provide explanation on the assumption of X,Y to be closed manifolds. This excludes even the case of Euclidean spaces, and a reasonable amount of applications questions focuses on manifolds with boundary, simplex, polytopes. Is boundaryless a must condition to ensure the results or just to annihilate boundary integral? If the technique can be used to manifold with boundary, there might be more application potential.

Correctness: The proofs are very detailed and carefully written.

Clarity: Yes, well written and the main idea is easy to follow.

Relation to Prior Work: Yes.

Reproducibility: Yes

Additional Feedback: For the proof of uniqueness, theorem 4, is that possible to apply the current technique to potential games with independent playing, which is a direct analogy of Rosen 1965, for certain locally uniqueness in P(M_1)\times ...\times P(M_n)?


Review 3

Summary and Contributions: This paper focuses on the two-player zero-sum continuous games, and reframe it from the OT perspective. The optimization strategy is investigated after parametrizing mixed strategies as mixtures of particles. Theoretical properties such as the convergence are proved. Experiments on synthetic data show the proposed method can be incorporated with the GAN training.

Strengths: This paper analyzes the two-player zero-sum games from a novel perspective. Although the mirror descent could not solve the Mixed Nash equilibria in high dimensions, the authors turn to the OT measure, which leads to geometric benefits. Theoretical results establish approximate mean-field convergence. Empirical results show the proposed WFR-DA is stable and efficient.

Weaknesses: 1. For the experiments, the authors should apply the method on large-scale datasets, and test with different dimensions (model size). The complexity of the proposed method should also be analyzed. 2. It's better to show more applications of the proposed analysis. For example, training "robust models".

Correctness: It seems the claims and method are correct. Do not check the proofs in detail.

Clarity: Since there are a lot of theoretical proofs in the paper, the authors can summarize the main claims of the theorems and thread them for better understanding.

Relation to Prior Work: The related methods are cited and discussed in the paper.

Reproducibility: Yes

Additional Feedback: This paper provides a novel analysis of the two-player zero-sum games and proposes strategies to solve it. There are two main concerns: 1. The high dimension is one of the main problems of the vanilla approach. The authors should clarify the dimension (e.g., to which scale) and show some comparisons on the efficiency/stableness with the previous methods. 2. Parametrizing mixed strategies as mixtures of particlesis is indeed a novel way to solve the problem. However, the main advantage of such a view should be emphasized and stressed in the GAN experiments since there are many choices to train such models.

[Author Response · NeurIPS 2020]

We thank the reviewers for their insightful comments. The three reviewers agree that the paper provides a novel approach to two-player zero-sum games with non-convex losses. In particular, **R1** states that the paper is clearly of interest for the Neurips community and could lead to interesting results in the future. We address their helpful comments below.

**Reviewer 1.** *1)* The mixed strategy approach involves computing distributions over the parameter spaces of the generator and the discriminator, and the resulting game on mixed strategies is indeed convex. The claim that we address non-convex games is to be understood in the sense that the original losses are non-convex and we study algorithms to solve the lifted convex problem with better performance than mirror descent (i.e. fixing parameters and updating weights), through the use of transport. s *Interpretation of mixed equilibria:* The resulting "mixed generator" is a mixture of distributions, each of them defined by a single generator.

*2)* As pointed out, Alg. 2 can be seen as performing gradient descent on the x parameters and multiplicative weights on the w parameters. At the level of measures, the multiplicative weights algorithm is the Fisher-Rao gradient flow, which is the gradient flow on the space of measures endowed with the Fisher-Rao (or Hellinger-Kakutani) metric. Analogously, gradient descent corresponds to the Wasserstein gradient flow on measures. When combining both algorithms, the dynamics can be seen as a gradient flow in the Wasserstein-Fisher-Rao metric, which is, loosely speaking, computed as the sum of the W an F-R metrics. See Preliminaries of Gallouët and Monsaingeon [2016] for more details (on the optimization case, analogous for games).

*3 and prior work comment)* Balandat et al. [2016] put forward an alternative way to tackle the problem of finding mixed Nash equilibria in compact strategy spaces which avoids dealing with dynamics in measure spaces. They use dual averaging (related to mirror descent) and show regret bounds. In particular, they show that the average regret tends to zero as $t \to \infty$ (Hannan consistency), and Hannan consistency implies that the empirical measures of each player converge to a mixed Nash equilibrium. However, this approach does not yield rates (contrary to us) and lies far from the gradient-based approaches frequently used in ML, which are more closely related to the measure-theoretic approach we take. We will include a more thorough comparison with this alternative work.

*4)* As pointed out, the paper is on multi-agent optimization (two agents) and not multi-objective. This will be corrected.

**Reviewer 3.** *Sample complexity for measure approximation:* We prove that the particle dynamics converge to the measure dynamics as the number of particles goes to infinity using a propagation of chaos argument. Although we do not provide quantitative rates, this convergence is in general dimensionally cursed and exponential in time. These are common drawbacks of the mean-field approach which were also encountered in the mean field analysis of neural networks literature (Mei et al. [2018], Rotskoff & Vanden-Eijnden [2018], Chizat & Bach [2018]). In practice, the number of particles needed to obtain good performance is much lower than the theoretical bound, and lower than the number needed for mirror descent ascent (Figure 1).

*Boundaryless (and compactness) assumptions:* In our theoretical analysis we assume that the parameter spaces $\mathcal{X}, \mathcal{Y}$ are compact Riemannian manifolds without boundary. The compactness and boundaryless assumptions preclude direct application of the theory to typical ML settings such as GANs. While the compactness assumption is necessary for MNE to exist, we introduce the boundaryless assumption to simplify theoretical arguments involving gradient descent and Langevin dynamics (gradient descent on spaces with boundary requires projecting after each step). However, we believe that the results could be extended to manifolds with boundary using the same ideas.

*Comment on the proof of uniqueness of Thm 4:* This proof is based on the argument of Rosen, 1965, which proves uniqueness of strictly convex games. In our case, strict convexity-concavity of the losses follows from the strict convexity of the differential entropy. As long as we can ensure strict convexity, a similar argument should allow us to prove uniqueness.

**Reviewer 5.** *- the authors should apply the method on large-scale datasets.* Evaluating our approach on larger datasets than CIFAR10 would entail training generative models on e.g. ImageNet, which is known to be very costly. We unfortunately lacked resources for this at the time of submission. *Complexity of the approach:* We compare the number of generator updates in Figure 2 and 3, and show that training mixtures is not significantly slower than training a single generator, with an additional clustering effect. Using many discriminators may slow down convergence.

*Further applications.* We thank the reviewer for his suggestions. Robust training is indeed an interesting source of 2-player games, that has been studied in the light of mixed equilibria in e.g. Pinot et al. [2020]. Our transport algorithm could be used to train robust mixtures of classifiers, although we leave this for a more applied future work.

[Meta-Review · NeurIPS 2020]

Three referees support acceptance for the contributions. I agree with the reviewers that the analysis can help understand algorithms in game. Although there is a gap between theory and practice due to mean-field approximation, it may be a first step towards analyzing the difficult problem. Thus I recommend acceptance.